# Relating sex-bias in human cortical and hippocampal microstructure to sex hormones

Svenja Küchenhoff [1,2,3,4] ✉, Şeyma Bayrak[1,2], Rachel G. Zsido[5,6], Amin Saberi [1,2,3], Boris C. Bernhardt [7], Susanne Weis [1,3], H. Lina Schaare [1,2], Julia Sacher[8], Simon Eickhoff[1,3] & Sofie L. Valk [1,2,3] ✉

Determining sex-bias in brain structure is of great societal interest to improve diagnostics and treatment of brain-related disorders. So far, studies on sex-bias in brain structure predominantly focus on macro-scale measures, and often ignore factors determining this bias. Here we study sex-bias in cortical and hippocampal microstructure in relation to sex hormones. Investigating quantitative intracortical profiling in-vivo using the T1w/T2w ratio in 1093 healthy females and males of the cross-sectional Human Connectome Project young adult sample, we find that regional cortical and hippocampal microstructure differs between males and females and that the effect size of this sex-bias varies depending on self-reported hormonal status in females. Microstructural sex-bias and expression of sex hormone genes, based on an independent post-mortem sample, are spatially coupled. Lastly, sex-bias is most pronounced in paralimbic areas, with low laminar complexity, which are predicted to be most plastic based on their cytoarchitectural properties. Albeit correlative, our study underscores the importance of incorporating sex hormone variables into the investigation of brain structure and plasticity.

Determining sex and gender differences in brain structure is of great societal interest to ultimately improve diagnostics and treatment of brain-related disorders. While macro-scale morphometrical sex differences are well documented, intracortical microstructural differences between sexes have not yet been characterized. To understand the source of systematic structural variations and their implications, it is crucial to further contextualize observed sex-differences, going beyond a sex binary. Underlining the overly simplified nature of a binary system, sex differences are determined by a complex combination of societal and epigenetic factors[1,2], sex chromosomes[2,3], and gonadal hormones[4–7]. Of these, activational sex hormone levels have

a particularly strong and dynamic effect on influencing sex-specific phenotypes[8–12]. In an effort to bridge traditional neuroanatomy and neuroimaging, we here investigate sex differences in intracortical microstructure in-vivo based on the ratio of T1- over T2 weighted (T1w/T2w) magnetic-resonance-imaging (MRI) intensities, and how these sex differences can be systematically linked to gonadal hormones specifically.

Human brain structure is most commonly characterized in-vivo by determining the macro-scale morphometry of the cortex. Analyses of volume or thickness variations based on the inner and outer cortical boundaries, however, are blind to microstructural variations within the

[1]Institute of Neuroscience and Medicine (INM-7: Brain and Behavior), Research Centre Jülich, Jülich, Germany. [2]Otto Hahn Group Cognitive Neurogenetics, Max Planck Institute for Human Cognitive and Brain Sciences, Leipzig, Germany. [3]Institute of Systems Neuroscience, Heinrich Heine University Düsseldorf, Düsseldorf, Germany. [4]Wellcome Centre for Integrative Neuroimaging, University of Oxford, Oxford, UK. [5]Cognitive Neuroendocrinology, Max Planck Institute for Human Cognitive and Brain Sciences, Leipzig, Germany. [6]Department of Psychiatry, Massachusetts General Hospital, Harvard Medical School, Boston, MA, USA. [7]MICA-Lab, Montreal Neurological Institute, Montreal, QC, Canada. [8]Centre for Integrative Women's Health and Gender Medicine, Medical Faculty & University Hospital Leipzig, Leipzig, Germany. ✉e-mail: svenja.kuechenhoff@ndcn.ox.ac.uk; s.valk@fz-juelich.de

cortical sheath. Microstructural changes within the cortical sheath are traditionally examined post mortem using cell-staining procedures[13–15]. On this micro-level, the human cortex is organized into several cell layers. The amount and prominence of each layer, as well as the sharpness of their boundaries, varies across the cortex, such that cortical areas can be classified into different types according to their laminar elaboration[14,16,17]. These variations in cortical types are systematically linked to the inherent property of plasticity[16,18], such that simpler laminar structures (e.g. paralimbic structures) are hypothesized to be more plastic than highly elaborate areas (e.g. primary visual cortex)[18,19]. Among others, one explanatory factor for this covariation of laminar differentiation with plasticity is the amount of intracortical myelin, which inhibits plasticity in the brain[20–25]. Intracortical myelin content correlates with laminar differentiation so that more elaborate laminar architecture is characterized by higher intracortical myelin content and higher stability[18,26]. Lastly, gradients of microstructural variation running along major axes of organization in the cortex support variation in brain function[27–29]. Multiple neuroanatomical accounts have illustrated the intrinsic link between microstructural properties, inherent brain organization principles, and brain function[30–32]. Thus, examining variations in: i) microstructural tissue properties, ii) cortical lamination and iii) the microstructural inter-regional organization in-vivo will yield a more refined understanding of sex differences in brain structure.

Gonadal hormone receptors are expressed in neurons as well as glial cells, which allows gonadal sex hormones to influence myelination[33], dendritic spine morphology and density[10,34], and cell metabolism[35], via ion channels, second-messenger systems, and gene expression[36]. The pattern of gonadal hormone synthesis products, as well as sex hormone receptors in the brain, differ by region, changes over the lifetime, and diverges between self-reported males and females (please note that in this manuscript, the terms female and male sex refers to a combination of self-reported gender and the report of having menstruated in one's life. The authors appreciate the complexity of biological sex and the influence of gender on biology, and do not postulate a sex binary.)[5,37,38]. The influence of gonadal hormones on brain structure is strongest during critical phases in development, such as early perinatal development and puberty, where the expression of gonadal sex hormones triggers the emergence of sexually divergent traits[8]. However, they continue to modulate brain structure throughout adulthood[12,34]. In ovulating individuals, brain structure varies on a short time-scale of days to weeks alongside sex steroid fluctuations during the menstrual cycle[39–42]. Progesterone levels increase up to 80-fold, and estradiol levels up to 8-fold over a period of 25-34 days[43,44]. Additionally, brain structure can also be modulated by exogenous sex hormones that influence sex-hormone profiles in the medium term of weeks to months, such as the commonly prescribed oral contraceptive pill (OC)[45].

The links between gonadal hormones and morphometric changes in brain structure on a short, medium, and long timescale are well documented. For example, macroscale structural changes on the short time scale (weeks) covary with the menstrual cycle, where paralimbic brain structures in particular adjust their structure to fluctuations of estrogen and progesterone[46–50]. On a scale of months, the use of OC in comparison to naturally cycling (NC) females has been shown to decrease gray matter of the amygdala and the parahippocampal gyrus[51], and the cortical thickness of the prefrontal cortex[45,52]. Furthermore, the intense hormone level changes during pregnancy go along with volumetric changes in medial temporal and medial prefrontal areas relevant for social cognition[53]. Sex-bias in gray matter volume in adults that developed over years from the onset of diverging sex hormonal profiles in puberty, are partly explained by circulating testosterone, progesterone, and 17β-estradiol levels[54,55]. Together, these studies help to identify brain areas that are biased by sex and influenced by sex hormones; however, they cannot show which microstructural features underpin these macro-level differences. In fact, morphometrical sex differences don't necessarily overlap. For example, while males are characterized by overall higher gray matter volume, females generally have higher gray matter density, and sex differences in cortical thickness are apparent in development, but become less pronounced in adulthood[56]. Similarly, microstructural effects don't seem to have a direct one-to-one match with macro-level anatomy. For example, quantitative brain-wide mapping of cell type distributions revealed lower cell density in volumetrically larger brain regions in male mice in comparison to females[57]. There has not yet been a characterization of human cortical microstructure sex differences in vivo, and it remains elusive if sex hormones might play a role in these variations. This study will thus aid in developing a more nuanced understanding of these anatomical variations.

To target this question, in the present study, we investigated sex-bias in microstructural variations and characterized to what extent sex hormones might be linked to the identified effects. We studied cortical microstructure with quantitative profiling of intracortical properties based on the MRI T1w/T2w ratio. More precisely, we analyzed (i) an average measure of regional intracortical tissue properties, (ii) an in vivo proxy of the local weighting of upper vs. lower cortical layers through intra-cortical profiles, and (iii) a measure of the relative distribution of microstructural organization across the cortex. We leveraged $N = 1093$ T1w/T2w MRI scans from the HCP young adult dataset and quantified these three local and global properties of individual intracortical microstructure across the cortex. We then contrasted these microstructural measures between females and males, tested how these sex-differences vary if systematically comparing males with females of particular hormonal profiles (approximated by self-reported menstrual cycle phase at the day of the scan and OC use), and quantified how these effects overlap with transcriptomic maps of sex-hormone related genes. Lastly, we linked the observed effects to a model from traditional cytoarchitectural neuroscience, the Structural Model[16,31], which predicts elevated plasticity for areas characterized by less elaborate laminar differentiation.

## Results
### Characterization of the three intracortical T1wT2w profile measures across the whole sample Fig. 1)
We analyzed the microstructural data of $N = 1093$ subjects ($n = 594$ females) from the HCP1200 young adult dataset[58], that was projected onto the cortical surface and parcellated into 400 Schaefer parcels (Fig. 1A)[59]. We additionally demonstrate that our results are not sensitive to our selected parcellation[60] (supplementary Fig. 1). Three different local and global measures of intracortical microstructure were used, which focus on different quantitative aspects of the microstructural properties (Fig. 1C). The microstructural profile mean (i) represents the local mean T1w/T2w signal intensity across the cortical gray matter tissue, the microstructural profile skewness (ii) denotes the local dominance of T1w/T2w intensity in superficial layers compared to deeper layers, and the principal microstructural gradient (iii) reflects an organizational axis of T1w/T2w intensity covariation along the cortex. To gain insight on the endocrinological effects of the hippocampal microstructure, we further projected T1w/T2w intensities on an unfolded hippocampal formation that was automatically delineated using the HippUnfold toolbox[61,62] (Fig. 1D). We included the hippocampal data of $n = 867$ individuals, for whom data in sufficient quality was available. All measures had previously been validated with histological work[15,63,64]. The group-averaged maps of the three cortical microstructure measures (Fig. 1C) and the T1w/T2w signal intensity pattern of hippocampus (Fig. 1 D) derived from the present sample broadly overlap with previous microstructural mappings of the human cortex[17,26,61,64–66].

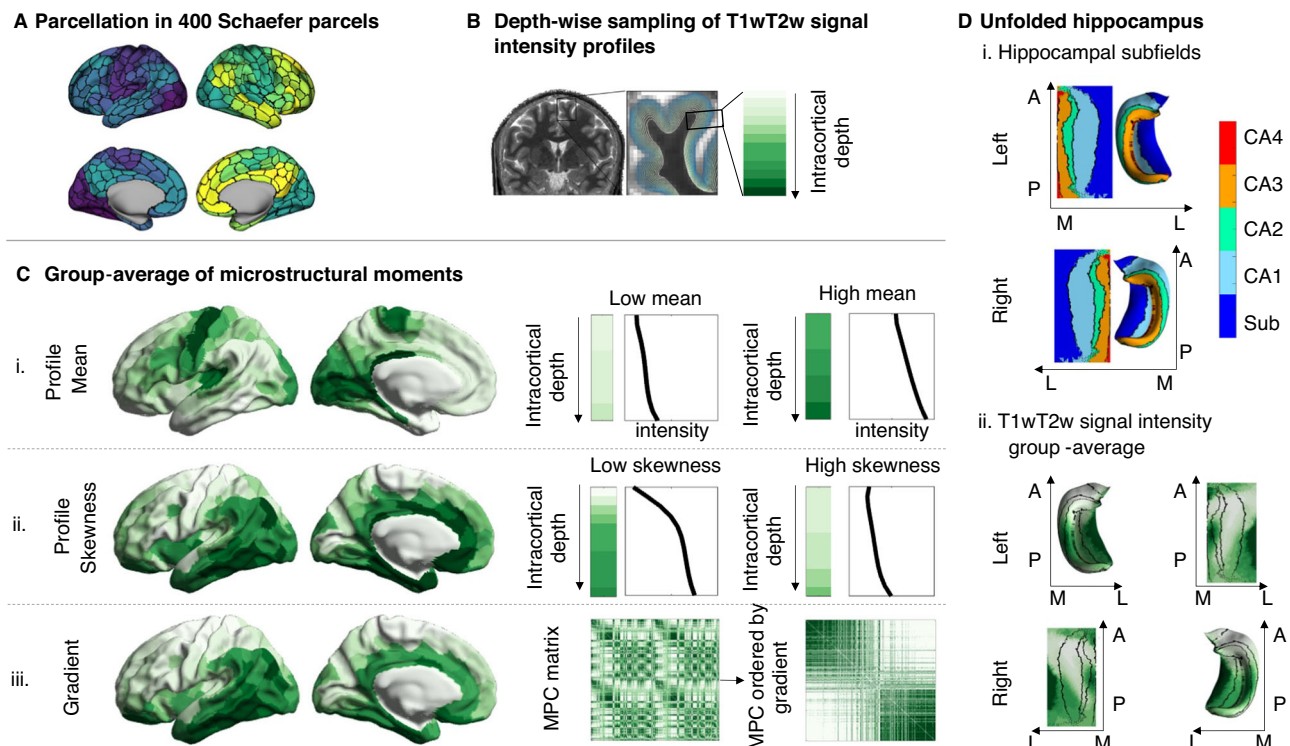

**Fig. 1 | Intracortical T1w/T2w signal intensity profiling. A** Parcellation scheme. **B** Intracortical sampling to build microstructural profiles. Twelve equivolumetric surfaces are put between cortical surface and white matter boundary of a single subject, yielding 12 sample points at different intracortical depths. **C** Left: Group averages ($N = 1093$) of microstructural measures (i-iii), plotted on the cortical surface. Right: examples for parcels with a high and low profile mean (i) and skewness (ii), per intracortical sample point, respectively. Microstructural profile covariance matrix (MPC) (iii) based on correlations of microstructural profiles between pairs of parcels. In the right MPC, parcels are ordered according to their microstructural differentiation, using the principal component derived from diffusion embedding. **D** Map of the hippocampal subfields after extraction and unfolding of the hippocampus (i), and the group-average T1w/T2w signal intensity (ii) for the left and right hippocampus. T1w/T2w the ratio of T1- over T2 weighted magnetic resonance imaging (MRI), MPC microstructural profile covariance matrix, CA Cornu ammonis, Sub Subiculum, A anterior, P posterior, M medial, L lateral. Source data are provided as a Source Data file.

## Intracortical microstructural organization differs between males and females Fig. 2)

To extract sex from our dataset, we categorized as female everyone who self-reported their gender as female and indicated they are or have been menstruating in their lives. To identify differences between males and females in each of our three intracortical microstructural measures, we first modeled each measure as a function of intracranial volume, age and sex, and then computed Benjamini-Hochberg false discovery rate (FDR) -corrected two-tailed t-tests for our contrast of interest (females > males) for each parcel separately ($q < 0.05$)[67]. To be able to compare effects between microstructural measures, we then transformed t-values to Cohen's d effect size values, where a positive Cohen's d represents parcels that had significantly higher microstructural measure values in females, and negative Cohen's d represents significantly higher values in males in the respective measure.

Comparing the microstructural profile mean between males and females, we found that males on average had a higher T1w/T2w profile mean across the whole cortex (cortex-wide average $mean_{males} = 1.7929$, $SD_{males} = 0.1191$; cortex-wide average $mean_{females} = 1.7498$; $SD_{females} = 0.0979$). These differences were particularly pronounced bilaterally in parietal, primary sensory motor areas, and unilaterally in left superior temporal and frontal areas (mean effect size of parcels that were significantly higher for males after FDR correction: d = −0.3214, $SD_{all\ neg\ parcels} = 0.1043$, Fig. 2A). The T1w/T2w profile mean of the entorhinal cortex was slightly higher in females, but this difference did not survive FDR correction. As a region bordering the entorhinal cortex, this pattern extended to the subiculum and the CA1 in the hippocampus, which also showed a higher T1w/T2w profile mean for females (mean positive effect size of all FDR-corrected areas

in the hippocampus d = .2559, $SD_{all\ pos\ parcels} = 0.0397$, Fig. 2B). In the most medial part of the hippocampus this pattern reversed such that the hippocampal subfields CA2 and CA3 had a higher profile mean in males than in females (mean negative effect size of all FDR-corrected areas d = −0.3315, $SD_{all\ neg\ parcels} = 0.1054$). The effects were slightly more pronounced in the right than in the left hippocampus.

Sex differences in T1w/T2w profile skewness were predominantly characterized by higher skewness in females, with an average of Cohen's d = 0.2438 across all parcels that had higher skewness values for females than for males ($SD_{all\ pos\ parcels} = 0.0700$). These differences represent a more equal ratio of T1w/T2w signal intensity in superficial to deep cortical layers in females in comparison to males. The observed differences were predominantly located in transmodal areas, including the anterior cingulate cortex, insular areas, and the prefrontal cortex. Only the left medial occipital cortex presented the opposite pattern, such that the ratio of T1w/T2w signal intensity was more uniform in males, while the lower profile skewness values in females represented a stronger dominance of signal intensity in deeper layers in this area (mean effect size of all FDR-corrected negative effects d = −0.3261, $SD_{all\ neg\ parcels} = 0.0424$). Cortex-wide patterns in mean and skewness sex differences showed a negative spatial correlation (r = −0.412, $p_{spin} < 0.01$). Further regional assessment of the association between mean and skewness showed that these measures had negative relationships in higher association regions, whereas they had a positive relationship in anterior insula and mid/anterior cingulate and temporal pole (Supplementary Fig. 2). This association, however, was mainly driven by females, where the average correlation between each parcel of baseline mean and skewness was r = −0.1156, while the average correlation between each parcel of baseline mean

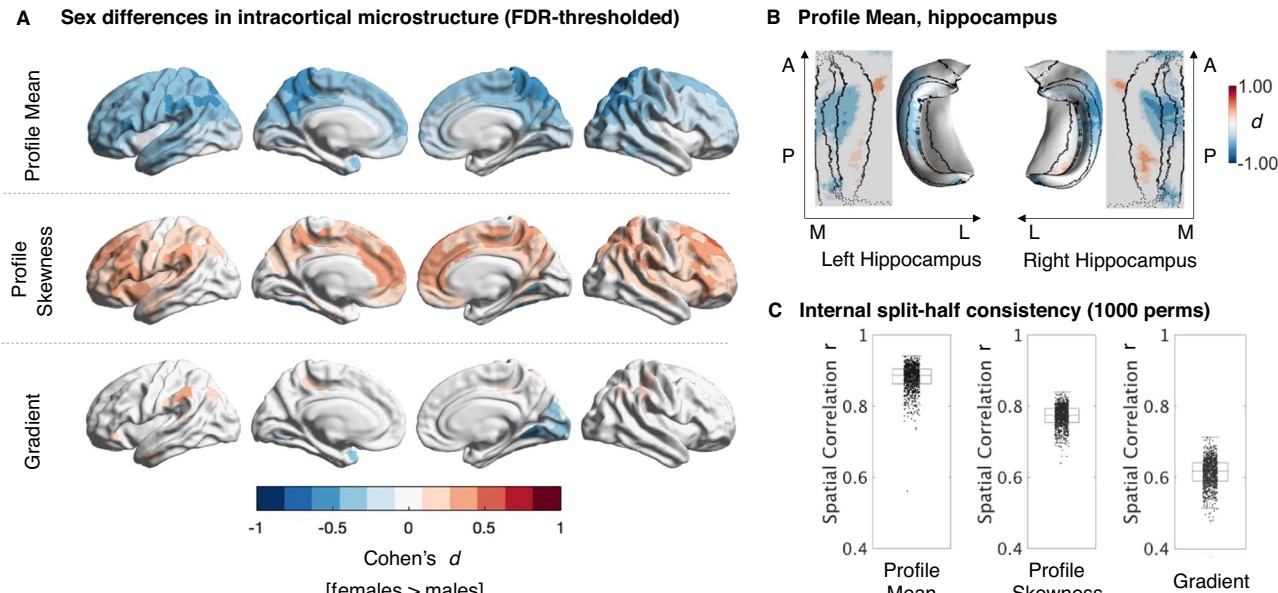

**Fig. 2 | Sex differences in the brain's microstructural organization. A** FDR-thresholded Cohen's d maps showing significant sex differences (females ($n = 594$) > males ($n = 499$)) in intracortical microstructure: T1w/T2w based intracortical profile mean, profile skewness, and the microstructural gradient. Red colors represent microstructural values higher for females, blue represent values higher for males. **B** 2D and 3D FDR thresholded Cohen's d maps of the unfolded hippocampus showing significant differences in T1w/T2w mean between females ($n = 500$) and males ($n = 367$ male). **C** Boxplots represent Pearson's r-values between unthresholded t-statistics resulting from two respective split-halves of the sample ($n = 1000$ permutations) comparing profile mean, profile skewness, and microstructural gradient between females and males, indicating the reliability of each measure. The median is shown as the central mark, the box indicates the 25th and 75th percentile; whiskers include all values not considered outliers (1.5*IQR from the quartile). Source data are provided as a Source Data file.

and skewness for males was $r = -0.0451$. This was mainly due to positive associations between mean and skewness in temporal and cingulate areas for males, but not females (Supplementary Fig. 2).

Lastly, evaluating sex differences in the microstructural gradient, we found a significant shift towards both the lower and the upper extremes of the microstructural gradient for females in comparison to males (Fig. 2A). In females, areas in red have a higher microstructural covariance with the gradient's upper anchor in fugal (limbic and temporal) areas than males, and blue areas have a higher microstructural similarity with the gradient's lower anchor in sensory-motor and primary sensory areas in comparison to males (Fig. 2A). Females' left medial occipital areas as well as the right temporal pole were found to be more microstructurally similar to the sensory anchor of the microstructural gradient relative to males (mean effect size of all FDR-corrected negative effects $d = -0.2931$, $SD_{all\ neg\ parcels} = 0.0866$). The bilateral supramarginal gyrus, parts of the inferior parietal cortex, and right anterior cingulate cortex were more similar to the upper anchor in fugal areas for females than for males (mean effect size of all FDR-corrected positive effects $d = 0.2355$; $SD_{all\ pos\ parcels} = 0.0441$).

We repeated all analyses also controlling cortical thickness as well as for family structure to account for potential confounds of twins in the dataset. Neither changed the original results (Supplementary Fig. 3). To receive a more nuanced understanding of the relationship between the morphological measure of cortical thickness and our microstructural measures, we additionally computed correlations between effect maps and found that only sex differences in the microstructural mean were negatively related to sex differences in cortical thickness, however the relationship was not significant if correcting for spatial autocorrelation with spin tests ($r = -0.36$, $p_{spin} = 0.092$, supplementary Fig. 4).

We then tested the consistency of our results by quantifying the split-half-reliability for microstructural mean profiles, skewness, and MPC gradient results[68]. We repeated our analysis in 1000 independent split-halves of our dataset (with equal ratios of males and females), and determined the mean spatial correlation for each of the three

measures between the independent halves, respectively. The mean spatial correlation between the t-statistic maps of split-halves for profile mean was $r = 0.8802$ across the 1000 permutations, the 5% 95% CI ranging between $r = 0.8156$ and $0.9227$ across all 1000 tests. The hippocampal values were similarly high, with a mean of $r = 0.8093$ (CI [5%, 95%] = 0.7481–0.8549) for the left, and a mean of $r = 0.8327$ (CI [5%, 95%] = 0.7818–0.8716) for the right hemisphere (supplementary Fig. 5). For profile skewness, split-halves spatial overlapped with a Pearson's $r = 0.7718$ between t-statistic maps (CI [5%, 95%] = 0.7173–0.8111). Only sex-differences in the gradient were less reliable, with a mean correlation of $r = 0.6136$ of t-statistic maps between split halves, ranging between $r = 0.5445$ and $r = 0.6698$. Overall, within this large cohort of healthy adults, observed sex differences in intracortical microstructure were thus highly to moderately reproducible.

### Effect size of sex-bias in intracortical microstructure varies as a function of approximated sex hormone concentration (Fig. 3)

We hypothesized that sex hormones play a substantial role in shaping cortical microstructure. Hence, we expected that differences in menses-related hormonal profiles would influence the effect size of the previously reported sex-differences. To test variation across hormonal status, we built five female subgroups that were determined by proxies for their current estrogen and progesterone concentration using a normative model of cyclic variations as well as by OC intake. We repeated the previous male vs. female contrasts five times, each time considering only those subgroups of females that were characterized by a certain hormonal profile: females who regularly took OC ($n = 170$), females who reported to be around their menstruation at the day of the scan (low estrogen, $n = 100$); females who reported to be around their ovulation (high estrogen, $n = 184$); females who reported to be in their follicular phase (low progesterone, $n = 171$) and females who reported to be in their luteal phase (high progesterone, $n = 113$) (Fig. 3A). Extending evidence about neuroendocrine plasticity effects on the short and medium term, we found that the observed differences between males and females varied as a function of menstrual cycle

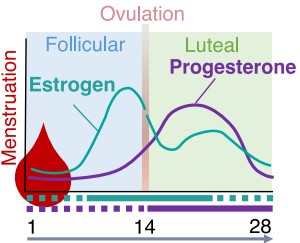

**A. Hormonal profiles during the menstrual cycle**

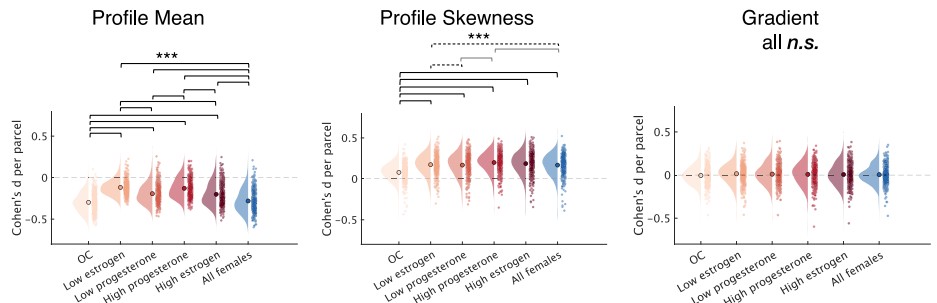

**B. Cohen's *d* per parcel, varying sex differences in microstructural measures by hormonal profile**

**C. Spatial maps of sex differences in profile mean and skewness if only females of certain hormonal profiles are considered**

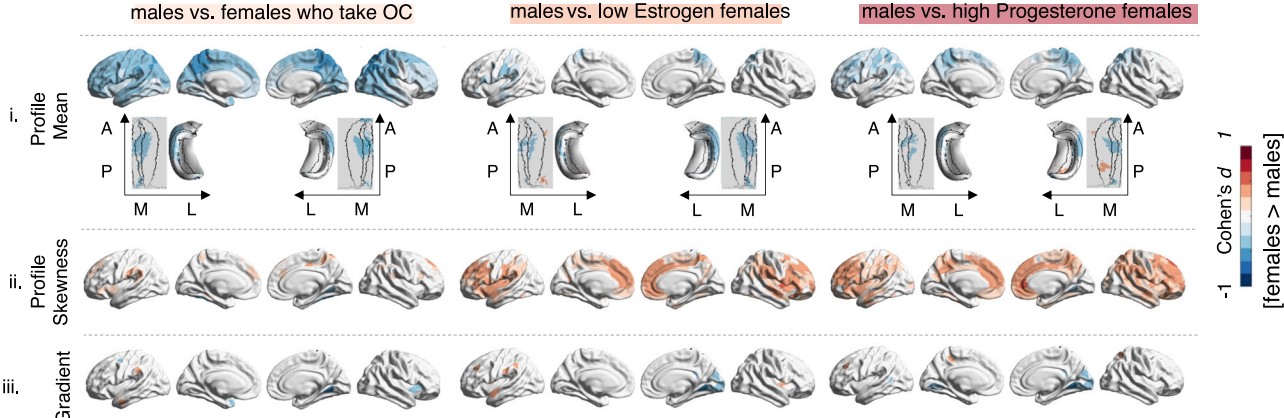

**Fig. 3 | Comparing males to different female sub-samples, grouped by menstrual cycle phase. A** Schematic of normative trajectories of estrogen and progesterone fluctuations during the menstrual cycle, based on[43,44,135]. Horizontal lines under the x-axis indicate grouping in this work: purple reflects progesterone (dotted = low; solid = high); turquoise reflects estrogen (dotted = low; solid = high). **B** Distribution of sex-difference effect per parcel, by group comparison for each microstructural measure. Brackets indicate significant differences in the cortex-wide sex-difference effect distribution between respective groups, based on post-hoc contrasts of ANOVA results using Tukey's honest significant difference procedure ($n = 400$ parcels, two-sided, *** black brackets $p = 0.0000$; ** dashed black brackets = $p < 0.01$, * dashed grey brackets $p < 0.05$). Test details for tests with $0.05 > p > 0.0000$ (all skewness): cortex-wide average $d_{high\ prog\ females-males} = 0.1995$ vs

cortex-wide average $d_{all\ females-males} = 0.1681$: CI [0.0057–0.0569] p = 0.0066; cortex-wide average $d_{high\ prog\ females-males} = 0.1995$ vs cortex-wide average $d_{low\ prog\ females-males} = 0.1680$: [0.0058–0.0571]$p$ = .0062; cortex-wide average $d_{high\ prog\ females-males} = 0.1995$ vs cortex-wide average $d_{low\ estr\ females-males} = 0.1731$: CI[0.0007–0.0520] $p$ = 0.0396. **C** FDR-thresholded Cohen's d maps projected on the cortical surface and the hippocampus of T1w/T2w profile mean (i), T1w/T2w profile skewness (ii), and microstructural gradient (iii) between males and sub-samples of females divided by OC use and menstrual cycle phase. For completeness, all other FDR-thresholded Cohen's d maps (all group-comparisons, for each of the three measures) are plotted in the supplement. OC oral contraceptives, A anterior, P posterior, M medial, L lateral. Source data are provided as a Source Data file.

phase and regular OC intake. To further interpret these sex-bias variations by hormonal group, we additionally investigated if: (i)., the mean sex-difference effect across parcels was conserved between group-comparisons (Fig. 3B and supplement 6) and (ii)., if the microstructural measure of any region also systematically varied in a within-females comparison (Fig. 4, Supplementary Fig. 7). We also added an internal consistency analysis to determine the specificity of the reported effect for the male sample (Supplementary Fig. 8).

For the microstructural profile mean, only the OC-group replicated the average initial sex-bias (post-hoc contrast across 400 parcels between group comparisons *n.s.*; see supplementary Fig. 9 for parcel-wise effect distribution by cortical type). Similar to the previously reported effect, males had significantly higher microstructural profile means in most parcels across the cortex when comparing them to females who regularly took OC (cortex-wide average $d_{OC\ females-men} = -0.2973$). In contrast, sex-bias estimations based on any other subgroup yielded significantly different cortex-wide effect sizes from the OC and initial group comparison (all $p < 0.001$, Fig. 3B). This was especially evident for females who were estimated to have low estrogen or high progesterone levels at the time point of imaging (cortex-wide average $d_{low\ estr\ females-males} = -0.1176$; cortex-wide average $d_{high\ estr\ females-males} = -0.1285$. The previously reported negative

sex differences disappeared or even changed sign such that females presented a higher mean when comparing males to females estimated to have low estrogen or high progesterone levels (Fig. 3B, C). We found that the sex-bias in the average T1w/T2w microstructural measure was least stable in the occipital lobe (Fig. 3C). Here, the sex-bias was particularly large when comparing males to females who took OC, but disappeared for females in the low progesterone group. Accordingly, for an intra-female contrast, we found that the occipital lobe of females who regularly took OC had a significantly lower T1w/T2w profile mean than the occipital lobe of naturally cycling females, and in particular those grouped for low progesterone (Fig. 4A). The T1w/T2w profile mean of males was generally higher than that of females, which explains the bigger sex differences when comparing males exclusively to OC females. The within-female contrasts in T1w/T2w profile mean, between females low and high in progesterone and those low and high in estrogen were not significant at an FDR-corrected threshold (for uncorrected maps, see supplementary Fig. 7).

Just as for the overall cortex-mean, the initial sex-difference effect across all vertices in the hippocampus was the same, on average, only when comparing males to females who regularly took OC, but not if comparing males to any of the NC female groups (average $d_{females-males} = -0.0724$; average $d_{OC\ females-males} = -0.0745$; post-hoc

**Differences between NC females and OC females in each intracortical microstructure measure (FDR-thresholded)**

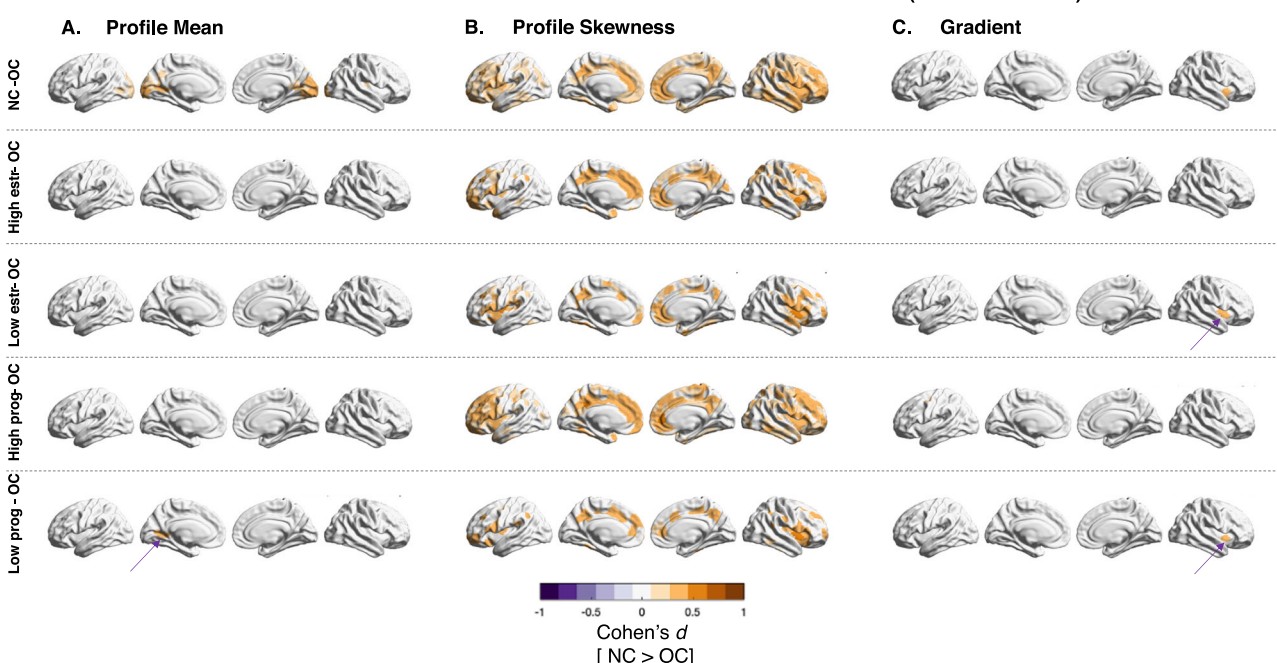

**Fig. 4 | Differences between NC females and OC females for each intracortical measure.** FDR-Thresholded Cohen's d for microstructural differences between female groups, comparing OC females with all NC females (n = 284), as well as OC females ($n$ = 170) with specific NC subgroups, divided by hormone estimations according to self-reported days after menstruation ($n_{low\ estrogen}$ = 100; $n_{low\ progesterone}$ = 171; $n_{high\ estrogen}$ = 184; $n_{high\ progesterone}$ = 113). Columns are the three microstructural measures T1w/T2w mean (**A**), T1w/T2w skewness (**B**), and the microstructural gradient (**C**). Purple areas are parcels which had significantly higher values in OC females, orange shows significantly higher values for NC females after FDR-thresholding (all Cohen's d). OC oral contraceptives, NC naturally cycling, prog progesterone, estr estrogen. Source data are provided as a Source Data file.

contrast *n.s.*, supplementary Fig. 6). Within the NC groups, there was no difference in hippocampal sex-bias if comparing males to females high or low in estrogen (average $d_{high\ estr-males}$ = −0.0318; average $d_{low\ estr-males}$ = −0.0337; post-hoc contrast *n.s.*). The low and the high progesterone groups were both significantly different from the initial, the OC, and the estrogen-group effects (average $d_{high\ prog-males}$ = −0.0266; average $d_{low\ prog-males}$ = −0.0381). However, in a parcel-by-parcel comparison within female sub-groups only, no area showed systematic differences when grouping females by estimated hormone levels (all *n.s.*). The variations of the hippocampal sex-bias between males and female subgroups should thus be interpreted with care.

Investigating microstructural intracortical profile skewness, the sex difference effects differed most between comparing males to females who took OC and the comparisons of males to any NC female subgroup (for parcel-specific comparisons, see supplementary Fig. 8). In fact, the previously reported sex-bias in microstructural profile skewness nearly disappeared when comparing males to females who regularly take OC (cortex-wide average $d_{OC\ females}$ = 0.07884, Fig. 3B), and was even more pronounced when comparing males only to females estimated to have high progesterone concentrations (cortex-wide average $d_{high\ prog\ females}$ = 0.1995). We show that this was because intracortical profile skewness values of females who took OC compared to NC females were significantly lower in precuneus, posterior and anterior cingulate, insula, and temporal pole (Fig. 4B). These areas are the same for which the T1w/T2w skewness sex-bias was smaller if one compares males to only females who take OC (Fig. 3C). Thus, the lower intracortical profile skewness in OC females converged to the male skewness levels which are generally lower than in females in these areas, demonstrating the steeper ratio of T1w/T2w signal intensity from superficial to deep cortical compartments in males and OC females. Females in their low progesterone group hereby were most similar to OC females, while the high estrogen and high progesterone

groups mainly drive the differences to OC females (Fig. 4B). The within-female contrast for T1w/T2w profile skewness between females low and high in progesterone and between females low and high in estrogen was not significant at an FDR-corrected threshold (for uncorrected maps, see supplementary Fig. 7).

Comparing the microstructural gradient of males only to subgroups of females of different estimated hormonal profiles changed the distribution, but not the center of the distribution of cortex-wide gradient sex differences (all cortex-wide effect size contrasts between any group comparison n.s, Fig. 3B). However, parcel and cortical wide specific analyses gave a more detailed overview of variations by hormonal subgroups (Fig. 3C; supplementary Fig. 9). The sex difference effect for the microstructural gradient varied strongest when comparing males to only OC takers versus comparing males to only females estimated to have high progesterone levels: Sex-bias between OC takers and males were least extreme (min $d_{OC\ females}$ = −0.4636, max $d_{OC\ females}$ = 0.3134), while sex differences between males and females in their high progesterone phase showed particularly big positive and negative effect sizes (min $d_{high\ prog\ females}$ = −0.5980, max $d_{high\ prog\ females}$ = 0.3398). In particular, the sex-difference effect for the gradient in the insula was negative between males and OC-taking females, but positive or *n.s.* between males and the different NC female groups. Investigating the female differences more closely, we found that the insula's microstructural profile covariance was closer with the paralimbic (fugal) anchor of the gradient in NC than in OC females, which seems to be associated with by the low estrogen and low progesterone groups (Fig. 4C). Within NC-females, contrasts for the microstructural gradient were not significant.

To summarize, sex-differences in intracortical microstructural measures differed in effect size if males were systematically compared to females roughly clustered in groups of different estimated hormonal profiles. These variations were mainly correlated with

microstructural differences between naturally cycling and regular OC intaking females and were most consistent for profile skewness. Between these two groups, the limbic, the prefrontal and the insular cortex showed particularly strong differences in profile skewness. Together, these results underline the importance of considering hormonal status when investigating sex differences or sex-specific cortical anatomy.

## Sex-biased intracortical structures spatially overlap with cortical expression patterns of sex hormone-related genes (Fig. 5)

Since microstructural sex differences varied strongly depending on which hormonal profile we approximated for females, we next asked whether the relevance of sex hormones in microstructural sex-bias could be supported on a molecular basis. A high density of sex hormone-relevant gene expression in areas in which we identified strong sex-bias would support the notion of sex hormones playing an important factor in sex-bias in cortical microstructure. Thus, we asked next whether transcriptomic maps of 25 sex steroid-relevant genes were generally linked to sex-difference effect maps for each microstructural measure (Cohen's d of sex differences in microstructural profile mean, skewness and covariance gradient), and then individually tested whether each of these 25 genes individually spatially overlapped with our microstructural sex-bias maps (Fig. 5). Please note that none of these individual links was significant at an FDR-corrected threshold, and should therefore not be considered more than trends.

We found that sex-hormone related genes were enriched in areas in which we found sex-differences in microstructural mean (F(336, 310) = 6.6, $p_{spin} < 0.05$), but not in microstructural profile skewness (F(336, 310) = 3, *n.s.*) or the microstructural gradient (F(336, 310) = 1.9, *n.s.*). Testing each transcriptomic map individually, we identified medium sized correlations, but not significant after spin-testing, between sex-differences in microstructural profile mean and the transcriptomic map of the androgen-receptor activation related genes *SRD5A3* (r = 0.31, $p_{spin} = 0.07$) and *AKR1C3* (r = −0.30, $p_{spin} = 0.11$), the androgen receptor gene AR (r = −0.31, $p_{spin} = 0.20$) and the progesterone receptor *PGRMC1* (r = 0.26, $p_{spin} = 0.17$). Further, after controlling for spatial auto-correlation, we found a small but significant spatial overlap with the sex steroid precursor gene *HSD17B3* (r = 0.13, $p_{spin} < 0.05$).

Sex-bias in T1w/T2w microstructural profile skewness demonstrated small spatial associations with Progesterone Immunomodulatory Binding Factor 1 (*PIBF1*, r = −0.25, $p_{spin} < .05$), the estrogen receptor 1 (*ESR1*, −0.18, $p_{spin} < 0.05$), the estrogen receptor beta (*ESRB*, −0.22, $p_{spin} < 0.05$), and the Growth Regulating Estrogen Receptor Binding 1 (*GREB1*, r = −0.24, $p_{spin} < 0.05$). There was a moderate but non-significant (after permutation tests) correlation between

skewness sex-differences and the estrogen receptor alpha (*ESRA*, r = −0.24, $p_{spin} = 0.27$) and the estrogen related receptor gamma (*ESRG*, r = −0.22, $p_{spin} = 0.23$). Lastly, sex differences in skewness also moderately overlapped with the sex-hormone synthesis relevant gene *AKR1C3*, which was not significant after controlling for spatial autocorrelation (r = 0.31, $p_{spin} = 0.05$). The gene specificity for profile mean and the profile skewness sex difference was supported by a non-significant and negligible correlation with the baseline gene map we extracted. This was, however, not the case for the microstructural gradient, which correlated stronger with the baseline gene factor than with any other transcriptomic map (r = −0.28, $p_{spin} < 0.05$, significant at FDR-corrected threshold).

Note that the AHBA dataset from which we derived the transcriptomic maps is composed of only one female and five male donors. We thus tested if the results identified here generally trend in the same directions if rerunning the analysis with the female or male donors only (supplementary Fig. 9). We found that this was the case for the results for profile mean ($r_{female-all} = 0.4638$; $r_{female-male} = 0.5119$) and profile skewness ($r_{female-all} = 0.7754$; $r_{female-male} = 0.6028$), but not for the microstructural gradient ($r_{female-all} = 0.2$; $r_{female-male} = 0.0603$). This analysis demonstrates that small correlations were particularly sensitive to donor sex (supplementary Fig. 10). Therefore, in this work, we focused on those that presented most reliably, independent of the sample composition.

## Sex differences differ in strength as a function of cytoarchitectural type Fig. 6)

Lastly, we investigated the cytoarchitectural communalities of areas in which we had identified microstructural sex differences. Cytoarchitectural properties, such as laminar differentiation, are suggested to relate to plasticity by the Structural Model[16]. We thus tested if microstructural sex-bias was associated with the level of cytoarchitectural laminar differentiation by computing spatial correlations between effect maps of the identified sex differences and the hierarchy of cortical types. These cortical types, originally defined by von Economo and Koskinas[14] and recently revised and histologically validated[17], describe a hierarchy of laminar differentiation from highly structured koniocortex to more diffusely structured agranular areas (Fig. 6A)[16]. As before, we report statistical correlation values and the respective permutation test *p*-value after spherical spin-tests (p-spin), and indicate if they remained significant at an FDR-corrected threshold.

We found that the effect maps of sex differences in microstructural skewness and the microstructural gradient significantly correlated with the hierarchy of cortical types at an FDR-corrected threshold, but not for the microstructural mean (Fig. 6B). A moderate

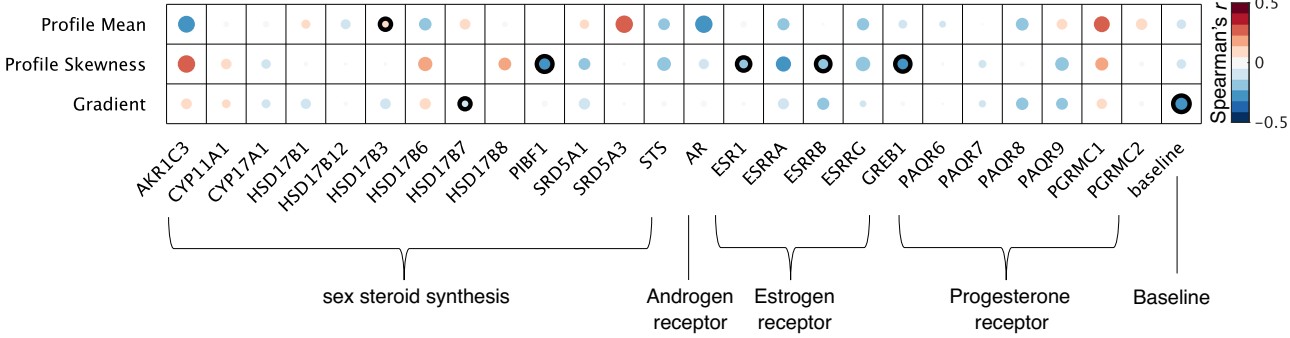

**Fig. 5 | Spatial overlap between effect maps of sex differences for the microstructural gradient, profile mean, and profile skewness.** Transcriptomic maps of genes are sorted by categories: sex hormone synthesis-related genes, and androgen, estrogen, and progesterone receptor-related genes. We test for spatial specificity by comparing against the principal component of all genes (baseline). Shades of red represent positive r-values, shades of blue represent negative correlations; circle size and shading indicate size of correlation. Values with significant *p*-values ($p < 0.05$) after permutation spin-testing are marked with a black outline (one-sided). Note that no correlation is significant when accounting for multiple testing at an FDR-threshold. Source data are provided as a Source Data file.

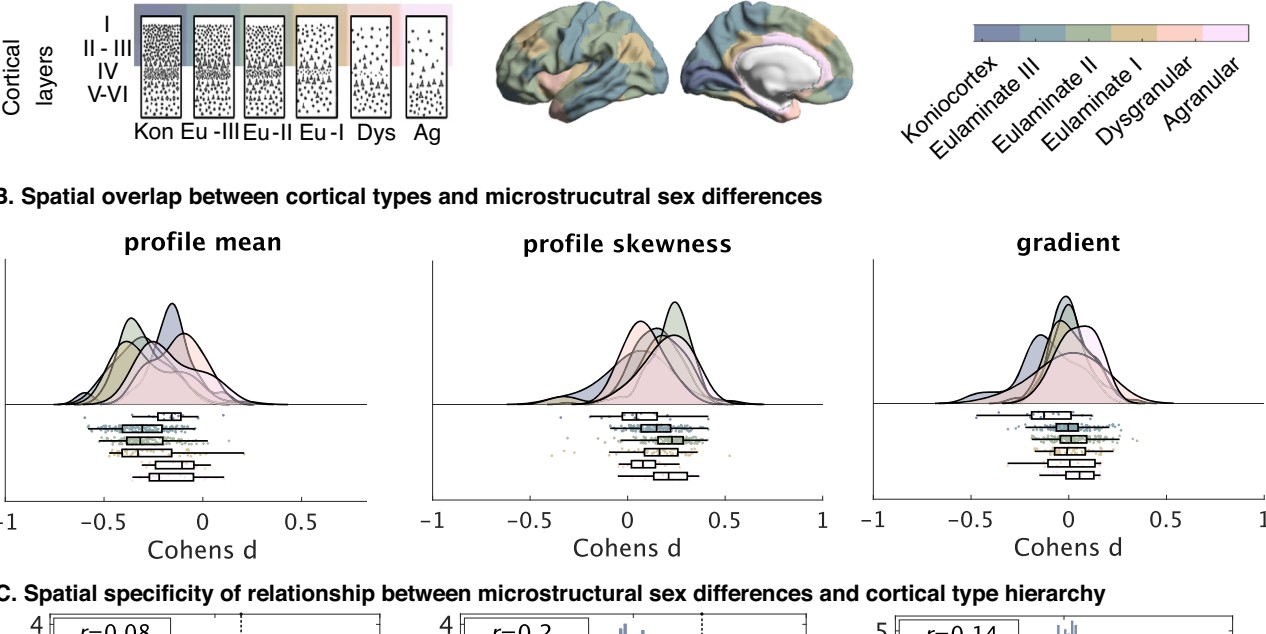

**Fig. 6 | Contextualization of effects by histological decoding. A** Schematic of cortical types according to von Economo and Koskinas and Garcia-Cabezas[14,17,131]. **B** Results were put into context by spatial correlations with a hierarchy of laminar differentiation (cortical types). Figures show links between cortical type hierarchy and effect values (Cohen's d for each of the 400 parcels) for each of the T1w/T2w profile-based intracortical measures. Raincloud plots[135] show distribution of sex-difference effects per parcel, binned by cortical type. Also binned by cortical type, boxes show the median and interquartile (25–75%) range of the respective distribution of sex-difference effects, whiskers depict the 1.5*IQR from the quartile. **C** Zero-distributions between random hierarchies and effect maps in comparison to the statistical r-value (one-sided). Profile skewness and gradient correlate significantly with histological hierarchy according to spatial autocorrelation significance level; profile mean does not. Kon koniocortex, EU-III Eulaminate III, EU-II Eulaminate II, Eu-I Eulaminate I, Dys Dysgranular, Ag Agranular. Source data are provided as a Source Data file.

positive correlation between T1w/T2w profile skewness (r = 0.20, $p_{spin}$ < 0.05) and cortical types indicated that differences in profile skewness were stronger for areas that can be characterized by less laminar differentiation (agranular mean Cohen's d = 0.21, eulaminate cortex II mean Cohen's d = 0.23). These results overlap with predictions about higher plasticity of these cortical types, and thus higher sensitivity to modulatory factors of plasticity such as sex hormones. Lastly, sex-difference effects in the microstructural gradient showed a small overlap with the hierarchy of cortical types (r = 0.14, $p_{spin}$ < 0.05. However, those regions with the strongest microstructural gradient sex-bias were in the most structured konio-cortical areas (mean Cohen's d = −0.13) and the most diffuse agranular areas (mean Cohen's d = 0.06)

### Cerebrovascular control analyses

Changes in cerebrovascular blood flow could pose a potential confound that explains microstructural variation with the menstrual cycle. In addition to including intracranial volume as a covariate in every linear model, we thus tested if the relation to sex hormone concentration would covary with the local density of cerebral vasculature (supplementary Fig. 11). No correlation remained significant at an FDR-corrected threshold, and here we report spin-permutation corrected p-values (p-spin).

We found that sex-differences in T1w/T2w profile mean overlapped moderately with cerebral vein density (r = 0.28,

$p_{spin}$ = 0.075). This overlap was stronger for sex differences in profile mean, comparing males to ovulating female sub-groups: profile mean differences in males compared to females in their low progesterone phase (r = .28, $p_{spin}$ < 0.05), in males compared to females in their low estrogen phase (r = 0.29, $p_{spin}$ < 0.05), in males compared to females in their high estrogen phase (r = .32, $p_{spin}$ < 0.05), and in males compared to females in their high progesterone phase (r = 0.35, $p_{spin}$ < 0.05), were significantly associated with cerebral vein density. No sex difference effect in T1w/T2w profile skewness overlapped significantly with any cerebral vasculature atlas (all $p_{spin}$ n.s.).

Lastly, while differences in the microstructural gradient between males and the collapsed female group did not correlate with the artery (r = 0.07, $p_{spin}$ = 0.156) or vein (r = −0.11, $p_{spin}$ = 0.162) atlas, they did for some female subgroups. The effect maps showed greater overlap with cerebral artery density (r = 0.17, $p_{spin}$ < 0.05) and in cerebral vein density (r = −0.21, $p_{spin}$ < 0.05), if only females in their high progesterone phase were considered when looking at sex differences in the microstructural gradient.

### Discussion

To link brain structure to behavioral and clinical outcomes, it is crucial to have a nuanced understanding of individual differences and the contextualization of such differences. Nevertheless, most brain structure studies fail to take major sources of structural variation into

account, such as sex-bias and systematic variations in gonadal hormones at the time of the neural measurement. Gonadal hormones systematically shape brain structure through molecular signaling cascades, being one of the major sources of sex differences in brain structure. While a large amount of neuroimaging studies have reported the effect of sex hormones on macro-level brain structure, the microstructural changes that underpin these variations are not well understood. To close this gap, we set out to characterize systematic variations in microstructural measures between males and females, and tested if this bias was linked to gonadal hormones. We approximated hormone concentration from cross-sectional menstrual cycle data, and investigated the spatial specificity of sex hormone-related gene-expression levels for the observed sex differences. Making use of T1w/T2w signal profiling within the cortical sheath, we derived an average measure of intracortical tissue properties (microstructural profile mean), a measure of intracortical local signal intensity distribution (microstructural profile skewness), and a measure of the covariation of microstructural organization across the cortex (principal microstructural gradient). This approach was inspired by traditional cyto- and myeloarchitectonic metrics. While it requires interpolation of data points in the cortical sheath cross-sectionally, its biological validity had previously been demonstrated with an ultra-high resolution cytoarchitectural ex-vivo dataset[64]. We first examined sex differences in a large cross-sectional cohort and then used a multimodal approach to identify neuroendocrine correlates of identified differences, with particular focus on sex hormones. We find that the cortical microstructure of males and females differs regionally for each of these microstructural measures. The effect size of the observed sex-differences depends on the estimated estrogen and progesterone levels of females at the time of the brain scan. In particular, we observe systematic differences between NC and OC females in all three microstructural measures. We find that the measure of microstructural skewness, being a proxy measure of laminar differentiation, proves particularly prone to hormonal variations across all analyses and controls. The hormone-related effect size variations are strongest for this microstructural measure. We further find that sex-bias in this proxy for cortical lamination moderately overlaps with expression levels of several sex-hormone-relevant genes, in particular of estrogen receptors. Lastly, sex-bias mainly appears in dys- and agranular areas, which are types of less laminated cortex that have been suggested to display comparably strong plasticity effects. We statistically controlled for ICV in all models and provided evidence that the observed effect for cortical lamination is not confounded with hormone-induced fluctuations in cerebrovascular blood flow.

In a quest to personalize clinical approaches, an understanding of systematic differences between males and females beyond the macro-scale is crucial, and knowing how to contextualize these differences is even more so. Here, we thus first described sex-bias in a big dataset based on T1w-T2w profiling, an approach inspired by classical histology studies[15,69], and then associated these effects with gonadal hormones in two complementary analyses.

We found systematic differences in all three microstructural measures when dividing the group into self-reported males and females. First, we found the average T1w/T2w signal intensity to be higher in the largest part of the male cortex, except for bilateral insular and medial temporal areas. This measure reflects intracortical myelin[22,26], as well as iron concentration[70], cell density, and water content. Mean T1w/T2w profiles thus describe differences and similarities between a combination of intracortical tissue properties. Myelin[4,26], iron concentration[71], cell density[57], and water content[72–74] are all subject to sexual differentiation triggered by sex hormones especially during critical periods of development. For example, sex hormones released during puberty lead to myelination sex differences in the rat prefrontal cortex[4]. The widely spread sex differences we found in this microstructural measure confirms this notion. On the other hand, the combination molecules that determine mean T1w/T2w signal intensity also make it the most prone to confounds, such as transmit bias field effects[75] and sex hormone effects on cerebral fluids. The moderate (but non-significant) overlap between mean T1w/T2w effects with cerebral vein density furthermore might reflect an interaction with the effect of venous blood on $T_2w$ signals[76]. Since the other two measures we analyzed, profile skewness and the microstructural gradient, are based on relative variations of T1wT2w, they do not suffer from the same limitations. We thus speculate that our mean microstructure results may be explained either by stronger cortical myelination in males or by female hormonal effects on cerebral fluids. Histological studies are required to disentangle the two.

Previous studies report hormone-related morphometric cortical differences in regions including the precuneus, insula, ACC, the middle temporal lobe, inferior frontal gyrus, middle frontal gyrus, superior frontal cortex, and hippocampus[12,77]. Here, we add a more nuanced characterization of these differences by showing that females in comparison to males show a more skewed microstructural pattern in temporo-parietal, precuneus, insular, and frontal areas. Phrased differently, these areas exhibit a comparatively less pronounced differentiation between supra and infragranular cortical layers in females compared to males. Only in the left medial occipital cortex does this trend invert, revealing a heightened dominance of signal intensity in deeper layers in females' cortex. Overall, the sex difference effects in mean and skewness tended to be opposite, i.e. the T1w/T2w signal in females generally had a lower mean intensity than in males, and the signal intensity distribution within the cortex was less evenly. This does not mean, however, that the measure of mean and skewness are perfect opposites and therefore redundant. Rather, our results identify important regional differences in these measures that vary by sex, demonstrating the value of each measure. In fact, in our subsequent analyses, we found the measure of skewness to be most reliably related to sex hormones.

We observed less pronounced but similar findings using a gradient approach, demonstrating that bilateral temporal-parietal regions were characterized by higher gradient loadings, i.e. were more similar to paralimbic, fugal, anchors, in females relative to males, yet visual areas were more similar to the sensory anchor in females relative to males. The occipital lobe is a koniocortical area characterized by six clearly distinguishable cortical layers[78]. Our findings indicate that the clear cytoarchitectural differentiation in this area was stronger in females in comparison to males. The male's occipital lobe, comparably, had more cytoarchitectural similarities with less clearly structured cortical types. Conversely, the temporo-parietal junction and medial sensory-motor areas in females were more similar to areas that are typically characterized by less structured cytoarchitecture. Thus, the covariance of microstructural profiles shifted more towards the extremes in females, while males exhibited a more gradual change of cytoarchitectural profiles. The overlap of significant areas with our skewness results shows that this can be mainly explained by differences in profile skewness, indicating a stronger differentiation between upper and deeper cortical features in females relative to males in terms of microstructure. Notably, while the overlap of these findings validates the measures respectively, it is important to note that our reliability measure was least consistent for the gradient approach.

Overall, the regional distribution of sex differences in microstructural measures roughly overlaps with previous reports on sex differences in gray matter volume in the same cohort of participants, in particular in cingulate and frontal areas[3]. Moreover, temporal-parietal, frontal, and insular regions were also found to display a diverging coupling of structure and function between sexes[79]. Indeed, in related work in the same sample[80], our group observed increased coupling of function and microstructure in females in regions that show heightened skewness in females. At the same time, sex differences in

microstructural measures were consistent above and beyond morphometric measures such as cortical thickness. How these different markers relate to each other, and what the functional implications of the demonstrated effects are, will be a topic of future work. Follow-up studies that focus on the functional implications of the reported microstructural measures are required to shine light on functional implications of the reported microstructural sex differences.

To put the identified sex-bias into context, we investigated a potential link between these effects and sex hormones with two orthogonal analyses. We showed that sex differences in all microstructural measures changed in effect size or even disappeared if males were compared to females of certain estimated hormonal profiles, while randomly subsampling the male group yields coherent results. This suggests that female sex hormones may play a role in microstructural sex differences in the human cortex. We furthermore demonstrate that there is a particularly big difference in cortical microstructure between females who took OC and NC females who reported menstruation within 28 days of the scan, as supported by significant within-females effects between these groups. Areas in which we observed these variations largely overlapped with regions that had previously been named as key regions for volumetric menstrual cycle differences (hippocampus, cingulate cortex, insula, inferior parietal lobule, prefrontal cortex[47]), or gray matter volume differences due to oral contraceptive use (prefrontal cortex[81] and the cingulate cortex[46]). Importantly, our findings do not extend to significant differences within cycle phases for any microstructural measure. Together, adding to previous observations of the effect of sex hormones on macro-level brain structure, our results demonstrate microstructural variability as a function of exogenous and endogenous sex hormones in females in the long and medium term.

We showed that we could only replicate the overall sex-bias in mean microstructure if comparing males selectively to females who took oral contraceptives. In contrast, comparisons between males with all other (NC) female subgroups decreased the sex-bias effect size for the average microstructural measure. This effectively implies that the average T1w/T2w signal intensity in NC females is more similar to the one in males, while the signal intensity is weaker in females who regularly took OC. We found this to be true specifically in the occipital lobe, where the average microstructural intensity for females who took OC was significantly lower than in NC females. This further underscores the potential influence of exogenous hormone manipulation on microstructure intensity, potentially explained by an interplay between glia and sex steroids, impacting myelin formation and organization[82]. The effect was particularly driven by the low progesterone subgroup, extending evidence from a recent preprint reporting progesterone-related white-matter microstructural and cortical-thickness variations in the occipital lobe[83]. Even though we observed more local variations in the sex-difference effect-size by hormonal subgroup comparison in the collapsed microstructural measure, these were not strong enough to show in a within-female comparison after correction for multiple comparisons. We thus conclude that sex differences in average cortical microstructure are at least partly dependent on long-term OC use; but that here, we did not find robust evidence for short-term, cycle-dependent variations within the female subgroups.

In contrast to the mean microstructural intensity, the sex-difference effect in microstructural skewness was driven by NC females, while OC females exhibited profiles more similar to males. The low progesterone and low estrogen groups replicated the initial sex-bias in the dominance of higher versus lower cortical compartments intensity. However, the effects were different from the main effect when examining females who regularly took oral contraceptives or had high progesterone concentrations. Specifically, there was nearly no difference in intra-cortical microstructure skewness between males and females who took OC (weak average effect), but there was

an even stronger average difference in intra-cortical microstructure skewness between males and females with high progesterone concentrations. OCs suppress circulating estradiol and progesterone levels[84–86]. Though no study to date has investigated such effects, we draw analogies between a recent morphological study focusing on the medial temporal lobe and its link to progesterone as well as chronic progesterone suppression (such as OCs): here progesterone was shown to shape MTL volume throughout the menstrual cycle, and ceases to do so when suppressed[87]. Speculatively, this effect might appear through progesterone's effect on myelination[88–90]. The variations we observed were mainly driven by stronger effects in the precuneus, prefrontal, anterior and posterior cingulate, and tempoparietal areas, which are explained by robust differences in skewness in these areas between females who take OC and NC females, and more specifically the high progesterone and high estrogen groups. This suggests that effects of oral contraceptives specifically contribute to a reduction or exacerbation of depth varying microstructural intensity, making this microstructural feature in OC females more similar to males. The strong hormone-related effect on microstructure skewness is particularly interesting when considering the fact that estrogen receptor expression is highly depth-specific, and particularly pronounced in the deeper cortical layers (V and IV)[91]. Behaviorally relevant sex hormone-related spiking pattern changes are also layer-specific particularly pronounced in deeper cortical layers[92], potentially driving structural plasticity.

The global relative measure of the microstructural covariance pattern (gradient) mirrored the trends of the microstructural skewness findings. Strongest deviations from the initial sex differences were found for females who took OC, and for females in their high progesterone phase. The most robust hormone-grouping related effect was for the insula, where the microstructural covariance profile of OC females was shifted to the lower anchor, and for NC females towards the fugal anchor of the microstructural gradient.

To support the evidence of our first endocrine analysis, we added a second, independent one. We show that the differences that we systematically observed between males and females present moderately overlap with areas of elevated expression levels of sex hormone-related genes. This offers a translation of a recent rodent study to humans, where sex differences in brain structure occurred particularly in regions enriched with sex hormone genes[93], and furthermore yields the second piece of evidence that sex hormones contribute to sex-bias in human intra-cortical microstructure using an independent hormonal analysis.

Regions in which the microstructural intensity mean was higher in males than females are rich in androgen receptors (AR) and regions where this microstructural intensity was higher in females than males strongly express the membrane-associated progesterone receptor (PGRMC1). Both the androgen receptor[94] and progesterone receptors[95] have a key roles in myelination. Progesterone and its metabolites interact with oligodendrial differentiation and thus affect development of oligodendrocytes and myelination[96]. Since microstructural profile mean (at least in parts) reflects intracortical myelin levels and the spatial resolution of MRI data is not sufficient, we hypothesize that myelination rather than synaptic plasticity effects or dendritic remodeling could drive the results observed in this study. Experimental studies observing causal links on a molecular level are needed to confirm this hypothesis.

In contrast, while sex-bias in the average cortical microstructure measure tended to overlap with myelin-plasticity-related genes, systematic differences in microstructural skewness rather overlapped with expression levels of genes that were linked to neural plasticity mechanisms. For example, the estrogen receptor genes are implicated in glutaminergic synapse formation[97], neurogenesis, synaptic spine density[98], synaptic plasticity[99] and neural differentiation[100]. We furthermore found selective links between sex differences in

cytoarchitectural lamination and genes important in the metabolism and thus supply of sex hormones such as *AKR1C3*[77] and *PIBF1*[3]. Together, this multitude of possible plasticity effects linked to the observed microstructural differences suggests a complex molecular interaction rather than a linear causal chain in the role of sex hormones in cortical microstructure.

Importantly, while our analyses demonstrate a general link between sex-hormone-specific genes and the microstructural mean, gene specificity for sex steroid synthesis, and sex hormone receptor genes, and account for auto-correlations, the links to individual hormones were not significant at an FDR threshold, controlling for number of genes and measures. Furthermore, even though our analyses suggest that these results are broadly similar across sex of the six donors that make up this transcriptomic sample, it will be important to revisit this analysis once a sex-balanced dataset becomes available.

It is also important to note that rather than longitudinally following up on microstructural changes going along with hormonal variations intra-individually, or post-mortem tissue analysis, we computed inter-individual contrasts on an indirectly approximated correlative hormonal measure. We acknowledge the extreme simplification for both NC and OC females, where we ignored the specific hormonal formulation of the pill and the initiation and duration of use due to a lack of data. We also limited the analysis to individuals who reported having a regular menstrual cycle, while ignoring perimenopausal hormonal changes as well as other endocrine conditions. To provide more robust evidence for a link between gonadal hormones and microstructure, it will be important to follow pioneering macro-scale studies in the future that investigate densely sampled intra-individual hormonal fluctuations as measured by blood-tests, which will measure female hormonal fluctuations more precisely and allow male diurnal hormonal fluctuations to be taken into account. Such studies will further help understand the association between the anatomy of the brain and hormonal variation and potential functional consequences. Similarly, despite moderate correlation of effect sizes, none of the individual transcriptomic map results remained significant at an FDR-threshold. We thus merely interpret our results as tendencies, which underline the importance of considering the complexity of hormones in the study of brain structure. However, since we benefit from a big sample size and thoroughly analyzed the microstructural sex differences with two independent hormonal analyses, we stress the importance of moving beyond a simple binarized understanding of sex differences and towards considering hormonal plasticity effects as crucial factors when investigating brain structure.

Numerous hormone-related neuroimaging studies find the hippocampus to be affected by sex-hormone-induced plasticity[41,42,48,50]. We thus made efforts to extend our cortical-surface-based analysis to the hippocampus by projecting an average measure of cortical microstructure on this unfolded surface. As expected, we found marked sex-differences that differed as a function of hippocampal subfield, with the subiculum showing higher T1wT2w values in females, but CA2/3 showing higher T1wT2w mean in males relative to females. Though research on sex differences in hippocampal structure in humans mainly focuses on the whole hippocampal volume, recent work has indicated marked changes in subicular microstructure[101] and volume[102]. Previous work in rats has shown marked changes in CA1 and CA3 but not CA2[103]. Such differences between sexes were mainly attributed to hormone-related effects. Indeed, when evaluating effects of sex hormones on hippocampal structure, we observed that the relatively stronger T1wT2w signal in females in the subiculum wasn't present when comparing males to females taking oral contraceptives or females in their low estrogen phase. A recent longitudinal study furthermore found that CA1 volume decreases for high progesterone concentration in the menstrual cycle[42]. Here, we complement this finding and show that sex differences in this area nearly disappear when females are in this part of their menstrual cycle. Importantly,

however, we couldn't identify a robust effect when computing inter-female contrasts for any region in the hippocampus. Thus, while we here show that taking the hormonal profile into account matters when investigating hippocampal-wide microstructural sex-differences, this study does not yield evidence for systematic hormone-related differences within females.

Overall, these findings extend previous work showing region-specific hippocampal sex differences and variations in these effects in relation to sex hormones. Similar to previous studies we again found that anterior-posterior differences within the hippocampus were substantial and need to be considered[27,104]. Through unfolding the hippocampus we increased regional specificity, considering the morphology of the hippocampus[63]. Further work studying the impact of sex hormones on hippocampal structure may use similar techniques to capture regional variation.

Lastly, given our technique was heavily inspired by traditional neuro-anatomy procedures[14,105,106], we aimed at embedding our results within current cytoarchitectural models. Demands to dynamically adapt brain structure vary across the whole brain: for example, it is adaptive that adult sensory areas respond in the same way to the same sensory inputs, while higher-order areas need to flexibly adjust their ways of processing depending on previous life experiences[80,107–109]. Consequently, stability of neural structures is thought to be adaptive in sensory cortices, while plasticity is adaptive in higher-order structures. This divergence between the need of plasticity and stability covaries with cortical microstructure. Sensory input to the brain is perceived and processed in idiotypic and unimodal areas, which are the cytoarchitectonically most elaborate areas (highly structured konio-cortex and eulaminate cortex III). The paralimbic cortex, on the other hand, receives projections both from other cortical areas, mainly higher-order sensory and association cortices, such as the auditory association cortex, temporal sensory association areas, and other prefrontal cortices[110,111], as well as from subcortical structures, such as the amygdala and the thalamus[112]. Supporting the hypothesis of a higher need of plasticity in areas that are not directly linked to sensory or motor organs, plasticity has been shown to systematically vary with laminar elaboration[18], so that less elaborate structures are the most plastic, and most elaborate structures present the most stability markers.

Here we show that this gradient from plastic to static cytoarchitecture extends to hormone-related sex differences, specifically for our proxy of cortical laminar differentiation. The more plastic the cortical type of a brain area, the stronger the effect of sex hormones we observed for this measure. Apart from sex hormone-triggered second-messenger cascades, other molecular factors additionally support the plasticity of these cortical types. For example, cortical types of less elaborate structures are richer in calmodulin-dependent protein kinase II (CaMKII), which is an enzyme known to be crucial for synaptic plasticity. Areas of higher granulation are richer in parvalbumin neurons, which limit plasticity via peri-somatic inhibition of neighboring pyramidal neurons[18]. Further, dendritic spine pruning is reduced in the adult limbic cortex in comparison to eulaminate areas, supporting LTP and synapse formation[19]. Together, these cytoarchitectural properties, paired with the appropriate hormone receptor infrastructure, may allow the cortex to adapt readily to varying demands commanded by fluctuating levels of sex hormones over short, medium, and long-time scales.

Work in the field of endocrine plasticity has been critiqued in the past for its potential confounds with hormone-related blood flow changes and water shifts in the brain. Since the MR signal and in particular T1wT2w contrast is both sensitive to myelin[22], but also to water and iron, and is non-uniformly impacted by the radiofrequency transmit field (B1 + ), it is not straight-forward to disentangle which was the source of the effects in our analyses. To limit these uncertainties, we first included intracranial volume as a covariate in each of our linear

models, statistically controlling for hormone-induced volume fluctuations; and secondly, we quantified the overlap between cerebral vasculature and the areas in which we identified sex-bias. The correlation with hormonal mean T1w/T2w profile, but not skewness supports the notion that T1w/T2w signal may indeed be globally modulated by the effect that sex hormones exert on water-balance and lipid metabolism. Thirdly, conceptually intra-cortical metrics may be least biased by these features as they reflect a relative metric perpendicular to the cortical sheet, implicitly correcting for spatial variations across the cortex. Combining evidence of effect size, correlations with hormonal measures, inter-female measures, control-analyses, and reliability analyses, we conclude that sex differences in our proxy for cortical lamination specifically are linked to sex-hormones.

In this study, we investigated whether sex-biases in three microstructural cortical measures - an average measure of cortical microstructure, a proxy for laminar differentiation within the cerebral cortex and the microstructural gradient - could be linked to sex-hormones, with two complementary correlative analyses in a large cross-sectional sample. We found that sex difference effects on one of our microstructural measures, laminar differentiation, were consistent across the sample, varied systematically between hormonal subgroups, particularly between NC and OC females, and overlapped with sex hormone gene expression levels. Adding to this, we found that this measure was not affected by vasculature, and remained robust in several permutation-control analyses. Together, our study is a nuanced investigation of microstructural sex differences and offers an explanatory link to sex hormones, adding to a well understood macrostructural account. The results emphasize the need to consider sex hormone concentration when investigating such differences, as systematic differences between groups can yield diverse and seemingly contradictory outcomes. Moving forward, longitudinal and interventional research in this area will contribute to a more comprehensive understanding of the interplay between sex differences, brain structure, and hormones, ultimately enhancing our insight into the intricate underpinnings of human brain function and behavior.

## Methods

### Participants and study design
In this study, we leveraged the HCP S1200 young adult data release[58]. In the following, we reiterate the most important details of this work, but more details of the HCP study design are described elsewhere[58]. The HCP dataset includes functional and structural MRI data acquired with 3T scanners from a total of 1206 healthy adult twins and their non-twin siblings born in Missouri, as well as behavioral and cognitive measures and extensive demographic and health-related data. Participants were recruited based on data from the Missouri Department of Health and Senior Services Bureau of Vital Records. The HCP consortium aimed at collecting a representative sample in respect to behavioral, ethnic, and socioeconomic diversity. To allow for sufficient variability in the healthy sample, only severe neurodevelopmental, neuropsychiatric and cardiovascular illnesses were excluded. For the present study, we only used structural MRI data and removed all subjects with missing MRI values, so that we included $n = 1093$ individuals ($n = 298$ monozygotic, $n = 188$ dizygotic twins, $n = 449$ not related individuals), out of which $n = 594$ were female. We classified those individuals as females who reported a female gender and are or have been menstruating in their lives, and all others as male. Note that all datasets collected in this study fall into one of these two categories, but that we distance ourselves from a sex- and gender-binary. We speculate that a more precise classification into gender and sex might lead to re-classification of some individuals, and take this into account as a source of random noise. The age mean +− SD was 28.8 +− 3.7 years (age range = 22-37 years). The current research complies with all relevant ethical regulations as set by The Independent Research Ethics

Committee at the Medical Faculty of the Heinrich-Heine-University of Duesseldorf (study number 2018-317).

### Neuroimaging data acquisition and preprocessing
We obtained readily preprocessed T1-weighted (T1w) and T2-weighted (T2w) structural scans in 0.7 mm isotropic resolution from the HCP openly available dataset. MRI data used in this study were acquired with Siemens Skyra 3 T scanners (32 channels) customized for the HCP. Two T1w and two T2w images were collected in a total of 32 minutes, using identical parameters respectively. T1w was acquired with the 3D MPRAGE sequence[113] in 256 sagittal slices with an echo time of 2.14 ms, an inversion time of 1000 ms, and a repetition time of 2400 ms (flip angle = 8°; matrix = 320). The T2w images with identical geometry as the T1w ones were acquired with the turbo spin-echo sequence[114] allowing for variable flip angles, with an echo time of 565 ms, a repetition time of 3200 ms, and a bandwidth of 744 Hz per pixel. Data was preprocessed with the Freesurfer version 5.3. Amongst other steps, T1w and T2w images were aligned, corrected for field bias, segmented and their ratio (T1w/T2w) was projected to the cortical surface in FSaverage5. Detailed pipelines and preprocessing steps are described in ref. 115.

### Intracortical microstructure profiling & parcellation
In traditional neuroanatomy, intracortical depth-dependent measures are commonly used to describe micro-architectural characteristics of brain regions[15,107,116]. Analogous to this approach, cross-sectional profiles of T1w/T2w MRI signal intensity across the cortical mantle build the basis for several local and global estimates of different cytoarchitectural properties[64,69]. First, the mean of T1w/T2w profiles perpendicular across the cortical mantle reflects the local mean T1w/T2w signal intensity across the cortical gray matter. Second, the skewness of T1w/T2w signal intensity across cortical depths contrasts local dominance of superficial to deep cortical layers[15]. Thus, microstructural skewness yields a proxy of laminar differentiation relating to cytoarchitectural complexity. In addition to these two regional measures of cortical microstructure, microstructural profile covariance (MPC) quantifies global variation of inter-regional microstructural similarity across the cortex[64]. The utility of this approach has been demonstrated in previous studies that showed a gradient of microstructural differentiation is mirroring brain function and orchestrates brain development in adolescence[117], and has been validated by comparing the MRI measure with traditional microscopy-based profiles[64]. Using the two local metrics of microstructural profile mean and skewness, as well as the global metric of the microstructural profile covariance gradient thus allows the analysis of different biologically relevant aspects of intracortical microstructure in vivo.

To build these measures from the preprocessed T1w/T2w surface data, we first computed 12 equivolumetric surfaces between the pial and white matter surface, generated by FreeSurfer. To compensate for cortical folding, the model varies the Euclidean distance ρ between two intracortical surfaces and thus preserves the fractional volume between them. The following formula was used to calculate ρ ($A_{out}$ = outer cortical surface, $A_{in}$ = inner cortical surface, α = fraction of the total volume of the segment accounted for by the surface).

$$p = \frac{1}{A_{out} - A_{in}} * \left(-A_{in} + \sqrt{\alpha A^2_{out} + (1 - \alpha)A^2_{in}}\right) \tag{1}$$

We sampled signal intensities along all linked vertices from the pial to the white matter surface across the whole cortex. To reduce computational efforts and multiple testing problems while accounting for a biologically relevant heterogeneity and spatial specificity, we parcellated the data into 400 Schaefer parcels by computing the average value of T1w/T2w signal intensity per sample point for each of

the 400 parcels[59]. The Schaefer parcellation approach is based on resting-state functional MRI (rs-fMRI), using a gradient-weighted Markov Random Field model which takes local transitions in rs-fMRI patterns as well as the similarity of global rs-fMRI patterns into account to define the parcels.

## Analysis of microstructural profiles across the whole sample: mean, skewness, and microstructural gradient

From the resulting $400 \times 12$ data matrix (12 sample points across intracortical depth), we computed the microstructural profile mean and skewness separately for each parcel for each subject. For better interpretability, the skewness values were then rescaled to values between 0 and 1.

To extract the principal gradient of microstructural covariation in the cortex, we first built a MPC matrix by pairwise Pearson product-moment correlations, taking the average whole-cortex intensity profile into account. MPC was thus defined for each microstructural profile pair i, j, and each participant s as:

$$MPC(i,j) = \frac{1}{n}\sum_{s=1}^{n}\left(\frac{r_{ij} - r_{ic}r_{jc}}{\sqrt{(1-r_{ic}^2)(1-r_{jc}^2)}}\right)s \qquad (2)$$

where $r_{ic}$ is the correlation coefficient of the intensity profile at parcel i with the average intensity profile across the entire cortex, $r_{jc}$ is the correlation between the intensity profile at each parcel j with the average intensity profile across the cortex, and n is the number of participants. Finally, we log-transformed and thresholded the MPC above zero and then only kept the top 10% of the strongest microstructural similarity pairs.

We decomposed the MPC matrix ($400 \times 400$ parcels) onto its low-dimensional representations by implementing the diffusion map embedding algorithm[118] using the BrainSpace toolbox[119]. First, we calculated an affinity matrix of the MPC with a normalized angle kernel function and then decomposed it nonlinearly onto a set of 10 principal eigenvectors, namely the gradients[64,120]. In this gradient space, parcels that have similar microstructural profiles are situated closely to each other, whereas parcels that have distinct profiles fall apart. For each participant, gradients of MPC were obtained separately. However, to increase comparability across gradients, we then aligned the individual gradients with the gradient derived from the group-average MPC. All individual gradients were then rescaled to values between 0 and 1.

## Hippocampal unfolding and projection

The hippocampus is both a highly plastic brain structure and a structure with a high density of sex hormone receptors and thus is a region of interest for our analysis. It is, however, not included in the Free-Surfer cortical projections. To nevertheless include this crucial brain structure in our analysis, we used an automated hippocampal segmentation pipeline - HippUnfold[121], which projects hippocampal MRI values to a 2D surface while preserving its topological structure, similar to cortical surface projections. Since this procedure is more sensitive to poor data quality, we performed a more stringent quality assessment. Out of the initial 1206 subjects, we excluded n' = 160 subjects with anatomical anomalies or tissue segmentation errors, n' = 93 subjects for which no preprocessed T1w images were available, and n' = 86 with morphological outliers (thickness, surface area, gyrification, curvature, or T1w/T2w values exceeding 2.5 sd of group average), such that overall we included n = 867 subjects (n = 500 female, n = 367 male) in the hippocampal analysis. The pipeline for the surface projection is described in detail elsewhere[61,63]. In short, the hippocampal regions of interest were first cropped from the preprocessed T1w/T2w data, and the hippocampal cortical surface was further segmented with a U-Net neural network architecture[122]. By solving Laplace's equation, HippUnfold then transforms the segmented MR data from Cartesian

native space to unfolded space. The transformed data was then stored in GIFTI files, so that any following analyses could take place as if it were surface data. However, due to the thin subregions and the complicated folding of the hippocampus, the previously described procedure of computing equivolumetric surfaces between outer and inner layer was not possible for the hippocampus. Instead, only the mean T1w/T2w ratio was used as a hippocampal MR measure, yielding one instead of three microstructural measures for the hippocampus.

## Sex-difference and proxies for links to sex hormones

We used different estimates of sex hormones to determine links between sex-biased microstructure and sex hormones in the short, medium, and long term. As a proxy for long-term effects of sex hormones, we first computed sex differences for each of the three T1w/T2w measures. We estimated sex differences with linear mixed effect models (LME) using the Matlab module of SurfStat[123]. Since the microstructural measures exhibit small to moderate correlations with intracranial volume (ICV, see supplementary Fig. 12), in each model we accounted for ICV, as well as age and the Euler number as a movement-related data quality measure:

$$\frac{T1w}{Tw2}\text{measure(parcel)} \sim b0*1 + b1*\text{sex} + b2*\text{age} + b3*\text{ICV} + b4*\text{euler\_no}$$

$$(3)$$

We then computed the t-statistics for the contrast females - males for each of our three microstructural measures. LMEs were estimated separately for each parcel and t-statistics were projected back to the cortical surface. We then two-sidedly corrected the t-values for a false discovery rate (FDR) of .05[67,124]. For easier comparison between tests, we report the effect size quantified by Cohen's d for all results that reached significance after this defined FDR-correction threshold. We repeated the analysis of all three measures regressing out cortical thickness and including the family structure (interaction between zygosity and family status) as a random effect to demonstrate that our results were not affected by these variables (supplementary Fig. 3). This suggests sex differences in cortical microstructure go above and beyond local variations in cortical thickness. We furthermore tested for spatial correlations between sex-bias in cortical thickness and microstructural markers using spin-tests as described above.

Secondly, we approximated hormone concentration levels in females to estimate links with sex hormones plasticity in the medium or short term. We used self-reported days since menstruation from the day of the scan and regular OC intake as a grouping variable. First, we subdivided females into two groups, those who regularly took OC (n = 170) and naturally cycling females (NC). Lastly, we built groups in which the estimated progesterone and estrogen concentration of NC females differed the strongest according to a normative trajectory of hormonal fluctuations within the menstrual cycle[125]. Since estrogen and progesterone concentration peak at different points within the menstrual cycle, we subdivided NC females into low and high progesterone and low and high estrogen groups. Importantly, since these peaks occur at different points in time, the grouping of estrogen and progesterone partly overlap and are thus not independent of each other. In total, we thus compared five subsamples of females against the cortical microstructure of males: an OC group, a high and low estrogen group, and a high and low progesterone group. We included all females who reported regular menstrual cycles within 28 days of the scan with their last menses between 0 and 28 days (n = 284), which is considered the length of a normal menstrual cycle[43]. Those reporting recent pregnancy, IUDs, hysterectomy, endometriosis, and similar conditions were excluded. Unfortunately, the current sample did not have information about perimenopausal staging or possible endocrine conditions, posing a potential source of noise. Estrogen is low in the beginning of the cycle and starts to rise before ovulation, with a second peak premenstrual in the luteal phase, before it drops again just before

and during menstruation (Fig. 3 A). Accordingly, we built a high estrogen group for females who reported they were in the middle of their menstrual cycle between day 7 and day 23, $n = 184$), and a low estrogen group for females who were just before and during menstruation ($n = 100$). Progesterone surges after ovulation during the luteal phase, and was thus defined as low before day 15 ($n = 171$), and high after day 14 ($n = 113$). This classification is in accordance with common comparisons between the time window of menstruation and the one around ovulation (high and low estrogen) and the luteal vs. follicular phase (high and low progesterone)[48,81,126,127]. While this best accounts for differences in concentration for each of these hormones, progesterone and estrogen groups do overlap due to this classification. Based on these groups, we then modeled the three microstructural measures with five LMEs, in which we included all males together with one respective female subsample. As before, we included age, sex, and ICV as covariates, and computed a contrast for females - males for each of the five models. Because we were interested in comparing the effect sizes of each group-comparison with each other but sample sizes differed, we computed Cohen's d for each parcel that survived FDR correction for multiple comparisons. Lastly, to determine whether effect sizes differed per group comparison, we computed a one-way ANOVA on the Cohen's d values across the 400 parcels between groups and post-hoc contrasts based on Tukey's honest significant difference procedure.

## Transcriptomic analyses

To complement our macro-level analysis of the effect of sex hormones on cortical microstructure on a molecular level, we then tested if sex hormone-linked gene expression patterns would overlap with the observed effects. We selected genes of interest (GOIs) via open ontologies such as KEGG and genecards (https://www.genecards.org/). We included genes that were either relevant in the synthesis process of the standard sex hormones (testosterone, estrogen, progesterone, adrenal androgens: dehydroepiandrosterone and androstenedione, progesterone-derived neurosteroids: allopregnanolone and pregnenolone), or linked to androgen, estrogen or progesterone receptors. In the end, we included n = 25 GOIs for which we had access to transcriptomic expression maps on the cerebral surface (see supplementary table 1). Our list of GOIs largely overlapped with previous selections for similar analyses (e.g.[3]) and was deemed a reasonable selection by an expert.

We used brain-wide gene-expression data provided by the Allen Human Brain Atlas (AHBA). The AHBA dataset consists of 3702 tissue samples and respective microarray expression data from six human donors[128]. For this dataset, the transcriptomic expression levels of more than 20,000 genes were measured in more than 50,000 probes across different cerebral and cerebellar regions. We accessed the AHBA database via the brainstat and abagen toolboxes[129], which allows retrieval of the transcriptomic data in Schaefer 400 parcellations. Brainstat fetches the tissue samples of all donors in MNI space and then applies intensity-based filtering, so that probes where more than 50% of samples exceeded a background noise threshold were excluded. It furthermore identifies differential stability across donors for probes indexing the same gene, and selects the most stable one. The remaining $n = 15.631$ genes were then matched to the respective Schaefer400 parcels and expression values were normalized across samples and genes with a scaled robust sigmoid normalization function. In the last step, expression values were averaged within each parcel and then averaged again across all six donors. For our analysis, we only considered the left hemisphere of transcriptomic maps, where expression profiles for nearly all Schaefer parcels were available, whereas the right hemisphere lacks sufficient sample density as it had only been sampled for two out of the six donors.

We followed a two-step procedure. First, we tested if hormone-related genes overall were related to the sex-difference maps by running a multivariate regression including all transcriptomic maps. To test for significance, we randomly permuted the sex-difference maps 1000 times, and ran a multivariate regression each time, computing a distribution of F-values. In the end, we computed the spin-corrected p-value by computing the proportion of permuted F-statistics that are greater than the original F-statistic. Second, we tested the relationship between the individual genes and the sex-difference maps of each microstructural measure. We computed Spearman correlations between gene expression enrichment for each of the selected GOIs with the observed differences in cortical microstructure between males and females. To control for spatial autocorrelations of gene enrichment analysis due to spatial non-independence of brain maps, we tested for significant spatial overlap between the respective transcriptomic map relative to randomly spun phenotype maps (i.e. our effect maps of sex differences). For that, we adjusted the spin-test function from the ENIGMA toolbox, so that spherical representations of the unthresholded three phenotypic maps were randomly spun in 1000 permutations and correlated with the 25 transcriptomic maps of our GOIs[130]. This procedure accounts for spatial autocorrelations by leveraging the spherical representations of the cerebral cortex. We report the frequency in which the true correlation between phenotypic maps and genes exceeded a test statistic generated of correlation values from randomly permuted phenotypic maps as spin-$p$-value. To account for multiple-tests, we furthermore computed FDR-thresholds for each of these spin-$p$ values. Additionally, to provide a measure of genetic specificity, we generated a measure of brain-gene-baseline and tested our effects against the baseline. We built the baseline transcriptomic map by extracting the principal component of all available transcriptomic maps in the left hemisphere. We provide spatial correlations (Spearman) between phenotypic maps of sex differences in profile mean, skewness and gradients with the brain gene baseline as a reference.

Lastly, to account for the sex-imbalance in the AHBA dataset (one female and five male donors), we reran the analysis as described above separately for the male and female donors only (supplementary Fig. 9). We then computed Spearman's rank correlation to test if results statistically trend in the same direction.

## Histological decoding

To contextualize our findings histologically, we chose a theory-driven, manual histological characterization of cortical types, defined by von Economo[14], digitalized to FSaverage space by Scholtens[131], and recently re-analyzed by Garcia-Cabezas[17]. Here, cortical areas are characterized according to different laminar features that are extracted from Nissl-stained sections. Among others, the authors characterized the sharpness of boundaries between layers, prominence of deep (layers V and VI) or superficial (layers II and III) layers, degree of granularity of cells and presence of layer IV. Based on these features, the cortex was then divided into cortical types. The hierarchy from agranular cortex, dysgranular cortex, eulaminate cortex 1-3 and koniocortical areas is analogous to a hierarchy of more diffuse to more elaborate laminar differentiation of cortical histology. We assigned each Schaefer400 parcel to one of the six cortical types and ran the previously described permutations-based spin-test based on Spearman correlations between cortical types and the identified sex differences for our three cortical microstructure measures, as such controlling for spatial autocorrelations.

## Control analyses: Vascular-hormonal coupling

Sex hormones influence structural plasticity and also modulate blood flow, vasodilation[132], and hemoglobin concentration[133]. We had previously included ICV as a covariate in our models to account for potential hormone-related volume shifts in the brain, which may influence MR signal intensity[72]. However, since the T1w/T2w measure reflects water and iron content in the brain[72,74] in addition to

myeloarchitectural features[26], we performed an additional control analysis post-hoc to our main analysis.

To identify which tissue changes may underlie the T1w/T2w hormonal effect, we investigated the spatial coupling of our effect maps with cortical vasculature maps. Veins and arteries in the brain are not distributed homogeneously, such that areas are differentially impacted by vasculature. We hypothesized that if the observed effects in T1w/T2w measures were mainly due to changes in hormone-related blood-flow changes, then areas in which the MRI signal is more strongly influenced by the vasculature should express the strongest effects. We used surface projections of an atlas that maps the distribution of arteries and veins in the brain[134]. To identify the influence of arteries and veins per parcel on the three microstructural measures, we first computed spatial Spearman correlations between these two atlases and our main sex-difference effect maps. In a second step, we aimed at identifying the relationship between vasculature and the effect of hormones. We thus computed spatial correlations between hormonal-subgroup effect maps, and the two atlases, respectively. Again, to address the problem of statistical auto-correlation and multiple comparisons, we used our adjusted spin-test function for all spatial correlations for max permutation testing, and built a zero-distribution out of spatial correlations between 1000 randomly spun cortical spheres of the unthresholded t-statistics for sex differences for profile mean, profile skewness and in turn the two cerebrovascular atlases.

### Reporting summary

Further information on research design is available in the Nature Portfolio Reporting Summary linked to this article.

## Data availability

This study followed institutional review board guidelines of corresponding institutions. All data analyzed in this study is publicly available. MRI data were obtained from the open-access HCP S1200 young adult sample (HCP: http://www.humanconnectome.org/). We accessed transcriptomic maps provided by the Allen Human Brain Atlas (AHBA) via the BrainStat and abagen toolboxes. Atlases used for the histological analyses were made available by the Dutch Connectome Lab (http://www.dutchconnectomelab.nl/economo/), and Bernier et al. [134], https://github.com/braincharter/vasculature). We further provide source data within this paper, and make all code available in the project's Github repository. Source data are provided with this paper.

## Code availability

All code is available here https://github.com/svennikue/sex-hormones-x-cortical-structure.git.

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

## Acknowledgements

We would like to thank the various contributors to the open-access databases that our data was downloaded from. HCP data were provided by the Human Connectome Project, Washington University, the University of Minnesota, and Oxford University Consortium (Principal Investigators: David Van Essen and Kamil Ugurbil;1U54MH091657) funded by the 16 NIH Institutes and Centers that support the NIH Blueprint for Neuroscience Research; and by the McDonnell Center for Systems Neuroscience at Washington University. We also thank Dr. Josh Grant for proofreading the manuscript. This study was supported by the Deutsche Forschungsgemeinschaft (DFG, EI 816/21-1), the Helmholtz Portfolio Theme Supercomputing and Modeling for the Human Brain. S.K. and A.S. were funded by the Helmholtz Association's Initiative and Networking Fund under the Helmholtz International Lab grant agreement InterLabs-0015, and the Canada First Research Excellence Fund (CFREF Competition 2, 2015–2016) awarded to the Healthy Brains, Healthy Lives initiative at McGill University, through the Helmholtz International Big-Brain Analytics and Learning Laboratory (HIBALL). A.S. was further funded by the Max Planck Society (Otto Hahn award). R.Z.'s time was, in part, supported by U54 MH118919 (Goldstein, Tobet, mPIs). S.W. was supported by the European Union's Horizon 2020 Research and Innovation Programme under grant agreement no. 945539 (HBP SGA3), and the DFG (491111487). J.S. was funded through SA 2285/3-1 as P.I., as well as the Brain-Hatch Funding from Max Planck Society. S.E. was funded by the European Union's Horizon 2020 Research and Innovation Program (945539 [HBP SGA3], 826421 [VBC], and 101058516), the DFG (SFB 1451 and IRTG 2150), and the NIH (R01 MH074457). S.L.V. was funded by the Max Planck Society through the Otto Hahn Award.

## Author contributions

S.K. conceived the project, which S.V. supervised. S.K. and S.V. analyzed the data and created figures. S.B. pre-processed hippocampal data. S.K. wrote the manuscript. S.K., S.B., R.G.Z., A.S., B.B., S.W., H.L.S, J.S., S.E., and S.V. revised and finalized the manuscript for publication.

## Funding

## Competing interests

The authors declare no competing interests.
