## [Peer Review File · Nature Communications]

Relating sex-bias in human cortical and hippocampal microstructure to sex hormonesReviewer #1 (Remarks to the Author):

Thank you for the opportunity to review "Relating sex differences in cortical and hippocampal microstructure to sex hormones" by Küchenhoff and colleagues. The manuscript describes sex differences in three metrics (profile mean, skewness and gradients) derived from T1w/T2w and their associations with hormonal profiles, transcriptomic maps of sex hormone related genes, and cytoarchitectural structure in the brain using the HCP dataset. The manuscript has several strengths including great clarity in its writing style, use of cutting edge methods, integration of information for different modalities of brain measurement, careful attention to that sex differences do and don't mean, and excellent use to control and sensitivity analyses. Overall, this is an important and well-executed study that highlights the importance of taking sex hormone into consideration when investigating brain structure and plasticity.

Nevertheless, there are some key points that would be good to address in revision.

Major comments:

(1) Lines 79-81, the authors argue that using T1w/T2w could yield a more nuanced characterization of sex differences free from biases due to insufficient control of systematic sex differences in brain size. This reasoning assumes that regional microstructure is independent from brain size – which is not the case (e.g. Warling, McDermott, et al, *Jnl Neuroscience*, 2021). The authors do in fact end up controlling for brain size in their analyses despite this initial statement regarding the attraction of microstructural measures for being independent of brain size. It would be good to do the following and show results in Supplementary Materials : (i) their models already include brain size, so please show the cortical and subcortical maps of eTIV relationship with each microstructural measure; (ii) show maps of the sex effect without controlling for brain size . Depending on the outcome of these two, the authors may need to modify their statement in lines 79-81.

(2) Why did the authors choose to use a purely fMRI derived parcellation (Schaffer) when looking at structural dependent variables? It would be good to justify this and/or switch to e.g. Glasser parcellation (which incorporates microstructural information) – either instead of Schaffer, or as supplement to show parcellation independence for key effects.

(3) The direction of mean and skew sex differences in T1w/T2w profile is opposite. Is it the case within each sex that having a higher mean value is associated with a lower skew? That is - does the coordination of sex differences in mean and skew values cohere with the coordination of these two properties across individuals? The answer to this question (in either direction) would be highly informative and helpful for interpretation of findings.

(4) In lines 192-194, the authors state that "We repeated all analyses additionally controlling cortical thickness as well as for family structure to account for potential confounds of twins in the dataset. Neither changed the original results (supplement 1)". This is good to know, but a complementary and equally important question whether the spatial patterning of sex effects on microstructure related to the spatial patterning of sex effects on thickness? This is biologically important as it speaks to the spatial congruence or not of sex effects on two intimately related metrics, but it is also methodologically important because sex effects on CT could potentially bias estimation of sex effects on T1/Tw metrics calculated between gray/white and pial surfaces.

(5) The section "Sex differences in intracortical microstructure vary as a function of approximated sex hormone concentration (Fig 3.)". Interpretation of the findings using different female subgroups would benefit from some additional analyses/visualizations. First, it would be good to see a scatterplot matrix where each cell has a scatterplot of ROIs effect sizes as points and the axes being contrasts of different female subgroups with males. Second, it would also be important to run these subgroup comparisons using independent subgroups of males so that you don't have coloring of the effects by a shared property of the male comparison group. Third, it would also be important to mention Sup Fig 4 results more in the main text here and use this to specify which regions are significant. In this Sup Fig, I think it would be clearer if visualizations simply used 3 block colors given these are post thresholding: white no sig effect and red/blue for sig

effects in each direction. Fourth, are the different effects seen for different female subgroups reflecting changes in the magnitude of the effects, or differences in interindividual variability between the different female subgroups? This would be important to clarify empirically. Fifth, it is striking that the OC group show preservation of the full group effect for mean T1/T2, but loss of the full group effect for skew. Moreover – the situation was not the same in the hippocampus. These are very challenging dissociations to explain biologically. What thoughts to the authors have? Finally, given the complexity of this results section, I would suggest providing readers a “mini-summary” with some key take aways at the end, around line 285.

(6) The section “Endocrine plasticity effects on intracortical structure spatially overlap with cortical expression patterns of sex hormone related genes (Figure 4)”. First, was correction made for multiple comparisons across genes and maps? Second, for most of genes examined, their spatial correlations with the three metrics did not reach statistical significance as it, i.e., $p_{spin} < 0.05$. However, the authors describe these nonsignificant findings as in line 311, Strong overlap were additionally presented by the androgen receptor gene AR ($r = -.31$, $P_{spin} = .15$) and the progesterone receptor PGRMC1 ($r = .26$, $P_{spin} = .20$), and using them as support (line 536 to 538) to draw a conclusion, strongly overlapped with sex hormone gene expression levels (line 624). This should be reworded. Third, there is insufficient attention given - in analytic design, presentation of results and discussion of results - to the fact that the AHBA dataset contains only one female. Therefore, all the expression maps examined are predominantly from the 5 male donors. Analytically, it would be important to provide some evidence that the reported connections between imaging and transcriptomics are at least trending in the same direction when expression maps are based on the single female donor. It is also important to say much more in Discussion and Limitations regarding the problem of sex imbalance in AHBA and what it means the authors can and can't say regarding their results. Finally, the corresponding Discussion section title should be changed to “Transcriptomics” decoding rather than “Genetic decoding”. The authors are not looking at genetic variation.

(7) Sections comparing sex difference to cytoarchitecture and cerebral blood flow. The question of multiple comparisons comes up here too. Also, the authors discuss similar correlations with some inconsistency. For example in the section starting on line 336, the authors state Sex differences differ in strength as a function of cytoarchitectural type (Figure 5), show that A positive correlation between T1w/T2w profile skewness ($r = .20$, $P_{spin} < .05$) and cortical types (line 348) and sex difference effects in the microstructural gradient showed moderate overlap with the hierarchy of cortical types ($r = .14$, $P_{spin} < 0.05$). Then on the section starting in line 367 the authors find some similarly sized correlations with maps of cerebral vasculature. However, these similar correlations are interpreted in different ways in the Discussion section, where the relationship with cytoarchitecture is treated a positive finding, whereas the vasculature correlations are downplayed. For example: we provide evidence that the observed effect was not confounded with hormone-induced fluctuations in cerebrovascular blood flow (line 412), The moderate overlap mean T1w/T2w effects with cerebral vein density furthermore (line 430), and Adding to this, we found that this measure was not affected by vasculature (line 625). It would be important to address such imbalances in interpretation.

(8) Discussion. Line 420 “The male cortex was characterized by ...” This suggests a typology (which the authors carefully push back against themselves in authors note) so should be reworded.

Minor Comments:

(1) Line 246, should cortex-wide average d_{high} estr female-male = -0.12846 be d_{high} progesterone female-male?

(2) Line 250, We found that sex differences in the cingulate cortex, the insula, the orbitofrontal cortex and the hippocampus were most affected by the menstrual cycle phase and exogenous sex hormone intake (Figure 3C). It is hard to see these regional differences from three comparisons side-by-side in Figure 3C. Just a suggestion, running a separate anova model in females alone, a F test map of group effect in either all five subgroups or the three groups in Figure 3C (taking OC,

low estrogen, and high progesterone) across 400 parcels may help to illustrate region variations in the menstrual cycle phase.

(3) The display of labels in Supplemental Figure 4 seems off, like FDR corr. Cohens d for contrast Men vs high estr, fo.

(4) In Supplemental Table 5, it is surprising to see $P_{spin} < P$ for spatial correlations between gene expression and sex differences in three metrics. I would expect spin tests are more stringent, yielding larger P_{spin} values.

(5) Line 372, We found that sex-differencers should be sex-differences.

(6) Line 428. Typo "the combination molecules".

(7) Line 430. Typo "moderate overlap mean T1w/T2w effects"

(8) Line 524 - should be "large" rather than "big"

Reviewer #2 (Remarks to the Author):

This article analyzes the cross-sectional MRI images of 992 young subjects from the Human Connectome Project. It calculates regional variation in cortical microstructure based on the T1/T2 ratio and analyzes how these metrics differ based on sex and menstrual phase (using self-reported days since menstruation). The authors also assess the spatial correspondence of MRI-derived maps with ex-vivo maps of sex hormone receptor gene expression. Although the results are very interesting, we have the following concerns:

The most important concern is that, although the authors state in the discussion that "It is important to note that instead of longitudinally following microstructural changes associated with hormonal variations within individuals, we computed inter-individual contrasts based on an indirectly approximated correlative hormonal measure. Therefore, we interpret our results as tendencies that highlight the importance of considering the complexity of hormones in the study of brain structure. However, due to our large sample size and a second, independent hormonal analysis, our results emphasize the importance of moving beyond a generalized understanding of sex differences and considering hormonal profiles as a crucial factor in interpreting and explaining these differences", the abstract and the paper are full of terms such as "influence of sex hormones (conclusion)" or "endocrine neuroplasticity". This can lead to an over-interpretation of the results. We suggest that the abstract and conclusion clearly reflect the cross-sectional nature of the MRI data and the absence of hormonal measures, and avoid terms such as "influence of sex hormones", "endocrine plasticity", etc., when discussing their own data.

Introduction:

We believe that the writing of the paper would benefit from narrowing and focusing the introduction, especially if this article is intended to be directed to the readers of a broad-scope journal such as Nature Communications. Along the same line, we believe that the introduction would benefit if the authors explain the biological interpretation of the extracted brain metrics to make it more accessible to a non-expert scientific audience.

Methods:

We recommend authors to include the Freesurfer-derived Euler Number as an additional covariate in the models, along with intracranial volume, age, and sex, to control for motion-related data quality.

We believe that authors should provide a clearer description of how they categorize the groups of interest, specifically females and males. The authors explain the criteria for classifying the female category (self-reported as females and being or having been menstruating), but they do not specify how they classify males. We assume that the male categorization follows the same logic as the female category (self-reported as males and not menstruating), leaving outside other categories (self-reported as females and not menstruating or self-reported as males and being or having been menstruating), but this should be explicitly stated. Also, authors sometimes mix the terms sex (female/male/intersex) and gender (women/men/other genders). For instance, when they define the female category, they state, "We classified individuals of female sex if they self-

reported their gender as female and indicated that they are or have been menstruating in their lives." We believe that a more appropriate definition should be: "We classified individuals of female sex if they self-reported their sex as female and indicated that they are or have been menstruating in their lives." Authors should homogenize the use of the terms males/females vs men/women throughout the manuscript. We suggest sticking to the male/female categories since this article focuses on sex-specific factors rather than gender.

If we understand correctly, the authors are parcellating the cortex into 12 sections. However, this parcellation is based on the information provided by approximately 4 voxels (as estimated by the voxel size of HCP images and the mean cortical thickness). We assume that the authors might have interpolated some of the values. In the same line, is the number of voxels different depending on the orientation of the perpendicular line used to calculate the layers? How does this might affect the calculated metrics, especially the skewness?

Discussion:

One strong point of the article is that it detects sex differences in brain structure when grouping individuals into the female-male categories. However, when dividing females into the five subgroup categories, these sex differences only replicate in the OC users. We believe this should be further discussed and treated as one of the main results of the article, especially since the authors disclose at some points that studies that merely test sex differences are over-simplistic and that considering sex-specific factors such as hormonal levels is essential.

Reviewer #3 (Remarks to the Author):

The authors interrogated microstructural differences in the context of sex and menstrual cycle phase on three distinct levels, providing a novel account of sex-specific cytoarchitectural profiles in the brain. I am excited by this work, beautifully executed, and offer several insights that may improve its impact.

1. Given the age distribution of the sample (22-37), I wonder if the authors considered potential influences of perimenopause (I have seen females of their mid to late 30s in this stage before, though rare) and/or possible endocrine conditions (e.g., PCOS, history of hysterectomy, etc) that may have impacted hormonal levels. It would be important to at least report the lack of this information for the sake of transparency on potential heterogeneity of the sample, in terms of female hormone concentrations.

2. On a similar note, it would be useful to clarify the criterion of those that "are or have been menstruating in their lives" - Was this explicit to those currently menstruating at the time of the study on a regular basis, or could some females who have not menstruated for months or years on end, but at some point in their lives (as suggested by this criterion), have been included? If so, that could certainly skew the hormonal distribution of the sample.

3. Regarding the inclusion of a subset of females using OC - More details regarding the type of birth control (estrogen only, progesterone only, or combination), the length of exposure (being mindful of any who have recently started OC and may, therefore, still be adjusting), and the like is needed, considering that these variables play a significant role in the efficacy of OC. I would also encourage the authors to be as explicit as possible when discussing past literature about the effects of OC - For instance, lines 69-71 on page 3 could use more detail (i.e., type of OC, length of OC exposure, age and menopausal status of the sample). In sum, what is meant by "regular" OC?

4. It would also be beneficial to expand on the cross-sectional limitations of this study as baseline hormone levels were not acquired from females. Though there is a "usual range" which we might expect reproductive females to fall within in terms of hormone levels at each menstrual stage, what is "normal" for these instances can vary across individuals. Though cross-sectional work is still very informative, a thorough acknowledgement of this limitation, especially in the context of

this study, is lacking.

5. Was the time of day held consistent across subjects when collecting hormone information? Were hormones also measured in the males? I wonder if diurnal testosterone fluctuations in males might have an influence on the current results.

6. Relatedly, were the hormonal assessments, MRI, and menstrual questions completed within the same day? Or could a few females have transitioned to a different menstrual phase over the course of data collection?

7. I also wonder if comparisons within females, between the various stage-associated subgroups, might be useful to further interpret the results presented here. If no variations between female groups are found, this may be attributed to the over-generalization of hormone levels by stage rather than on an individual or change-from-baseline degree. If variations are found, however, this could corroborate the authors' grouping approach.

8. Regarding the results showing differences between males and high progesterone females, I would be interested to see a more in-depth interpretation from the authors to offer potential explanations for this specific finding.

9. In general, I would also encourage the authors to take a more careful approach with their discussion of results. The female subgroups may be a bit over-simplified, especially considering the moderate presence of estrogen in what the authors refer to as only the "high progesterone" stage. I am very pleased to see a paper that covers this topic, but am eager to see more unique conclusions that pose important questions while also being mindful of limitations. There is more room for discussion in this manner.

Thank you to the authors for taking on this work. I look forward to seeing it published.

Reviewer #4 (Remarks to the Author):

I co-reviewed this manuscript with one of the reviewers who provided the listed reports as part of the Nature Communications initiative to facilitate training in peer review and appropriate recognition for co-reviewers.

**Response to Reviewers (NCOMMS-23-52974) - Reviewer 1**

We would like to thank the Editors and Reviewers for their positive evaluations, constructive
comments, and for the opportunity to submit a revised manuscript. We feel that the comments and
suggestions have greatly improved our manuscript. In this covering letter, we outline the steps we
took to address the suggestions of the Reviewers in a point-by-point fashion below and highlighted
the corresponding changes in the manuscript in yellow , and marked additions to the manuscript in
*italic*.

**REVIEWER COMMENTS**

**Reviewer #1 (Remarks to the Author):**

**Thank you for the opportunity to review “Relating sex differences in cortical and hippocampal**
**microstructure to sex hormones” by Küchenhoff and colleagues. The manuscript describes sex**
**differences in three metrics (profile mean, skewness and gradients) derived from T1w/T2w and their**
**associations with hormonal profiles, transcriptomic maps of sex hormone related genes, and**
**cytoarchitectural structure in the brain using the HCP dataset. The manuscript has several strengths**
**including great clarity in its writing style, use of cutting edge methods, integration of information**
**for different modalities of brain measurement, careful attention to that sex differences do and don’t**
**mean, and excellent use to control and sensitivity analyses. Overall, this is an important and well-**
**executed study that highlights the importance of taking sex hormone into consideration when**
**investigating brain structure and plasticity.**

We thank the Reviewer for the appreciation of our work and the insightful comments, which we have
addressed below.

**Nevertheless, there are some key points that would be good to address in revision.**

**Major comments:**

**(1) Lines 79-81, the authors argue that using T1w/T2w could yield a more nuanced characterization**
**of sex differences free from biases due to insufficient control of systematic sex differences in brain**

size. This reasoning assumes that regional microstructure is independent from brain size – which is
not the case (e.g. Warling, McDermott, et al, Jnl Neuroscience, 2021). The authors do in fact end up
controlling for brain size in their analyses despite this initial statement regarding the attraction of
microstructural measures for being independent of brain size. It would be good to do the following
and show results in Supplementary Materials : (i) their models already include brain size, so please
show the cortical and subcortical maps of eTIV relationship with each microstructural measure;

We agree, and added the following text to the methods, and figure to the supplement.

**methods, p. 27b ll. 881:** *Since the microstructural measures exhibit small to moderate correlations*
*with intracranial volume (ICV, see supplementary figure 10), in each model we accounted for ICV, as*
*well as age and the euler number as a movement-related data quality measure: [...]*

T1w/T2w Mean

T1w/T2w Skewness

Microstructural Gradient

**Supplementary Figure 10. Intracortical T1w/T2w signal intensity profiling, correlation with ICV.**
*Parcel-wise correlation between ICV per subject and microstructural measures across the cortex. Pink*
*reflect positive, green reflect negative correlations. No value is higher than $r = .34$ for T1w/T2w mean,*
*the peak value for T1w/T2w skewness is $r = .21$, and the highest correlation between ICV and a gradient*
*parcel is $-.21$.*

**(ii) show maps of the sex effect without controlling for brain size. Depending on the outcome of**
**these two, the authors may need to modify their statement in lines 79-81.**

The authors thank the Reviewer for this comment. In fact, we feel the need to adjust the text to the
following in order to avoid confusion about what T1w/T2w profiling can and cannot do (p.3, line 101
note that we adjusted this section also according to comment 4):

*“[...] Together, these studies help to identify brain areas that are implicated in sex differences and*
*influenced by sex hormones, however, they cannot show which microstructural features underpin these*
*macro-level differences. In fact, morphometrical sex differences don’t necessarily overlap. For example,*
*while males are characterized by overall higher gray matter volume, females have a generally higher*
*gray matter density, and sex differences in cortical thickness are apparent in development, but become*
*less pronounced in adulthood (Gennatas et al., 2017). Similarly, microstructural effects don’t seem to*
*have a direct one-to-one match with macro-level anatomy. For example, quantitative brain-wide*
*mapping of cell type distributions revealed lower cell density in volumetric larger brain regions in male*
*mice in comparison to the female counterpart (Kim et al., 2017). There has not been a characterisation*
*of human cortical microstructure sex differences in vivo, and it remains elusive if sex hormones might*
*play a role in these variations. This study will thus aid in developing a more nuanced understanding of*
*these anatomical variations. [...]*”

**(2) Why did the authors choose to use a purely fMRI derived parcellation (Schaefer) when looking at**
**structural dependent variables? It would be good to justify this and/or switch to e.g. Glasser**
**parcellation (which incorporates microstructural information) – either instead of Schaefer, or as sup**
**mat to show parcellation independence for key effects.**

We thank the Reviewer for this remark. In previous work we have found little difference between
Schaefer 400 and Glasser 360 parcellations for neuroanatomical studies (Valk, 2022). A potential
benefit of the Schaefer parcellation scheme is that the parcels link to global and local functional
profiles, and thus account for functional (re)organization, and have roughly equal size, whereas for
the Glasser atlas primary areas are large and association areas smaller, creating a potential bias when

averaging anatomical values in this schema when not taking individualized parcellations. Nevertheless,
we have now also ran the analyses using the Glasser parcellation and observed consistent results. We
have added these findings to the Supplementary Materials for completeness.

**Supplement 1. Intracortical T1w/T2w signal intensity profiling sex difference, Glasser 360**
**parcellation.** Shown are Cohen's *d* values for the female > males contrast, controlling for family
structure (including the interaction between twin status and family status), only coloring in parcels
with a *p*-value lower than the FDR threshold.

*Results [addition, p.4 ll.135]*

*"We additionally demonstrate our results are not sensitive to other parcellations (Glasser, 2016;*
*supplementary figure 1)."*

**(3) The direction of mean and skew sex differences in T1w/T2w profile is opposite. Is it the case**
**within each sex that having a higher mean value is associated with a lower skew? That is - does the**
**coordination of sex differences in mean and skew values cohere with the coordination of these two**
**properties across individuals? The answer to this question (in either direction) would be highly**
**informative and helpful for interpretation of findings.**

We agree with the Reviewer that this is an informative additional piece of information. We assessed
the relation between mean and skewness in more detail and reached the conclusion that the relation
is informative, but not big enough to make mean and skewness redundant. We thus add the following
to the text and supplement:

Results [addition, p. 6 ll.205]

“Cortex-wide patterns in mean and skewness sex differences showed a negative spatial correlation ($r=-0.412$, $p_{spin} < 0.01$). Further regional assessment of the association between mean and skewness showed that these measures showed particular negative relationships in higher association regions, whereas they have a positive relationship in anterior insula and mid/anterior cingulate and temporal pole (Supplementary Figure 2). This association, however, was mainly driven by females, where the average correlation between each parcel of baseline mean and skewness was $r = -0.1156$, while the average correlation between each parcel of baseline mean and skewness for males was -0.0451 . This was mainly due to positive associations between mean and skewness in temporal and cingulate areas for males, but not females (Supplementary Figure 2).”

Discussion [addition, p.17, ll.532]

“Overall, the sex-difference effects in mean and skewness tend to be opposite, i.e. the T1w/T2w signal in females generally has a lower mean intensity than in males, and the signal intensity distribution within the cortex is less evenly. This does not mean, however, that the measure of mean and skewness are perfect opposites and therefore redundant. Rather, our results identify important regional differences in these measures that vary by sex, demonstrating the value for either measure. In fact, in our subsequent analyses, we find the measure of skewness to be most reliably related to sex hormones.”

Supplementary Figure 2. Parcel-wise correlation between baseline T1w/T2w mean and skewness profiles, for the whole group, for females only and for males only. Red areas represent positive correlation between skewness and mean T1w/T2w, blue represent negative correlations.

(4) In lines 192-194, the authors state that “We repeated all analyses additionally controlling cortical thickness as well as for family structure to account for potential confounds of twins in the dataset. Neither changed the original results (supplement 1).”. This is good to know, but a complementary and equally important question whether the spatial patterning of sex effects on microstructure related to the spatial patterning of sex effects on thickness? This is biologically important as it speaks to the spatial congruence or not of sex effects on two intimately related metrics, but it is also methodologically important because sex effects on CT could potentially bias estimation fo sex effects on T1/Tw metrics calculated between gray/white and pial surfaces.

We thank the Reviewer for their comment. We agree that it is important to prevent cortical thickness bias of T1w/T2w derived metrics sex-effects, which is why we chose a control analysis that takes the spatial variance of this metric into account. We furthermore agree that the discussion of previous findings on morphometrical neuroimaging is informative to the reader and helps to interpret the results biologically and methodologically. We thus addressed this issue in several ways throughout the manuscript: First, we add the following section to the introduction to inform the reader on the congruence between cortical microstructure and morphometrical sex differences (p.3, ll 101; note that we adjust this section also according to suggestion 1):

Introduction [Addition]

“[...] Together, these studies help to identify brain areas that are implicated in sex differences and influenced by sex hormones, however, they cannot show which microstructural features underpin these macro-level differences. In fact, morphometrical sex differences don’t necessarily overlap. For example, while males are characterized by overall higher gray matter volume, females have a generally higher gray matter density, and sex differences in cortical thickness are apparent in development, but become less pronounced in adulthood (Gennatas et al., 2017). Similarly, microstructural effects don’t seem to have a direct one-to-one match with macro-level anatomy. For example, quantitative brain-wide mapping of cell type distributions revealed lower cell density in volumetric larger brain regions in male mice in comparison to the female counterpart (Kim et al., 2017). There has not been a characterisation of human cortical microstructure sex differences in vivo, and it remains elusive if sex hormones might play a role in these variations. This study will thus aid in developing a more nuanced understanding of these anatomical variations. [...]”

Second, we added an additional control analysis, where we compute the overlap between sex-
differences in cortical thickness and the three microstructural measures. We detail both control
analysis in the relevant methods section (“Sex-difference and proxies for links to sex hormones”).

*Methods [adjustments, p. 28, ll 916]*

*“We repeated the analysis of all three measures regressing out cortical thickness and including*
*the family structure (interaction between zychosity and family status) as a random effect to*
*demonstrate that our results were not affected by these variables (supplement 3). This suggests sex*
*differences in cortical microstructure go above and beyond local variations in cortical thickness. We*
*furthermore tested for spatial correlations between sex difference in cortical thickness and*
*microstructural markers using spin-tests as described above.”*

We accordingly add the results of this supplementary analysis to the results section, the supplement
and modified the discussion:

*Results [Addition, p. 7 ll 227]*

*“We repeated all analyses additionally controlling cortical thickness as well as for family*
*structure to account for potential confounds of twins in the dataset. Neither changed the original*
*results (supplement 3). To receive a more nuanced understanding of the relationship between the*
*morphological measure of cortical thickness and our microstructural measures, we additionally*
*computed correlations between effect maps and found that only sex differences in the microstructural*
*mean were negatively related to sex differences in cortical thickness, but the relationship was not*
*significant if correcting for FWE with spin tests ($r = -0.36$, $p_{spin} = 0.092$, supplementary figure 4).”*

Supplementary [Addition]

**Supplementary Figure 4.** Associations between cortical thickness and microstructural sex differences.
 (A) FDR-thresholded Cohen's d maps showing significant sex differences (females-males) in cortical
 thickness, Red colors represent microstructural values were higher for females, blue represent values
 higher for males. B) Associations between sex differences in cortical thickness and effect values
 (Cohen's d per parcel) for each of the T1w/T2w profile-based intracortical measures. The upper row
 visualizes zero-distributions between random hierarchies and effect maps in comparison to the
 statistical r -value, the bottom row plots cortical thickness sex differences on the X-axis, and sex
 differences of microstructural measures on the Y-axis.

Discussion [addition/adjustment, p. 18, ll. 567]

“[...] Indeed, in related work in the same sample (Valk et al., 2022), our group observed
 increased coupling of function and microstructure in females in regions that show heightened skewness
 in females. At the same time, sex differences in microstructural measures were consistent above and
 beyond morphometric measures such as cortical thickness. How these different markers relate to each
 other, and what the functional implications of the demonstrated effects are, will be a notion of future
 work. Follow-up studies that focus on the functional implications of the reported microstructural
 measures are required to shine light on functional implications of the reported microstructural sex
 differences.”

(5.1) The section “Sex differences in intracortical microstructure vary as a function of approximated
 sex hormone concentration (Fig 3.)”. Interpretation of the findings using different female subgroups

would benefit from some additional analyses/visualizations. First, it would be good to see a
scatterplot matrix where each cell has a scatterplot of ROIs effect sizes as points and the axes being
contrasts of different female subgroups with males.

Thank you for this suggestion. To further illustrate the difference between males and females as a
function of female hormonal variation, we added scatter plots illustrating the relative difference
between males and females as a function of hormonal status in females (supplementary figure 7). We
also added a direct contrast between female subgroups which we deemed to be highly informative of
true systematic hormone-related group-differences as well (figure 3).

*Results [Addition]*

*“To further interpret these sex-bias variations by hormonal group, we additionally investigate*
*if i), the mean sex-difference effect across parcels is conserved between group-comparisons (Figure*
*3B and supplement 6) and ii), if the microstructural measure of any region also systematically varies*
*in an within-females comparison (Figure 3D). We furthermore added an internal consistency analysis*
*to determine the specificity of the reported effect on the male sample (supplementary Figure 7).”, p.9,*
*ll 274*

*[..]*

*For the microstructural profile mean, only the OC-group replicated the average initial sex*
*difference effect (post-hoc contrast across 400 parcels between group comparisons n.s.; see*
*supplementary Figure 8 for parcel-wise effect distribution by cortical type). p.9, ll.284*

*[...]*

*Investigating microstructural skewness, the sex difference effects were most different*
*comparing males with OC vs. any NC female subgroup (for parcel-specific comparisons, see*
*supplementary figure 7)., p.11 ll.330*

*[...]*

*Comparing the microstructural gradient of males only to subgroups of females of different*
*estimated hormonal profiles changed the distribution, but not the mean of cortex-wide sex differences*
*(all cortex-wide effect size contrasts between any group comparison n.s, Figure 3B). However, parcel*
*and cortical wide specific analysis give a more detailed overview of variations by hormonal subgroups*
*(Figure 3C; supplementary figure 7). p.11 ll. 349*

Cortical Types

Correlation between effect sizes by parcel, comparing males against different female subgroups: Profile Mean

Cortical Types

Correlation between effect sizes by parcel, comparing males against different female subgroups: Profile Skewness

 **Supplementary Figure 7.** Effect sizes for sex differences in T1w/T2w profile mean, skewness and
 microstructural gradient per parcel, and how these effects change depending on which female
 subgroups the male subjects are compared to. Females were divided into females who took OC,
 females estimated to be in the high progesterone phase of their menstrual cycle, in the low
 progesterone phase, in the high estrogen phase and in the low estrogen phase, respectively. The
 diagonal shows the sex-difference effect size distributions, and how they shift depending on the
 contrast. Scatter plots show correlation between two respective effects, and the deviance of each
 parcel from the other contrast's effect size. The first column (black box) is the original all females vs.
 all males sex difference effect, compared to all contrasts between males and female subgroups. All
 values represent Cohen's d values (females - males). Parcels are coloured by cortical types (left).

**(5.2) Second, it would also be important to run these subgroup comparisons using independent**
 **subgroups of males so that you don't have coloring of the effects by a shared property of the male**
 **comparison group.**

Thank you for this suggestion. We added a Monte-Carlo analysis to analyze the dependence on the
 male sample of the effects at hand. We did so by firstly re-computing the contrasts with 2 randomly
 chosen sub-samples of males that were equally sized to the female subgroup. In $n = 1000$ splits, we
 then correlated the effect sizes (Cohen's d) of these randomly chosen male subsamples with each
 other. We include the result of this internal consistency analysis as a supplement.

Internal consistency - male sample, hormonal contrasts.

A) Males vs. OC females

B) Males vs. High estrogen

C) Males vs. Low estrogen

D) Males vs. High progesterone

E) Males vs. Low progesterone

Supplementary Figure 7. Split-correlation of 1000 random permutations for all hormonal contrasts and each microstructural measure. For every split, we computed the contrast between males and females, randomly choosing only a subsample of males, such that $n(\text{males}) = n(\text{females})$. We then computed the internal consistency for this randomly chosen male subsample by correlating the effect sizes of this contrast with the Cohen's d effect sizes of an equally sized and randomly chosen subsample of males. Datapoints represent correlation values for each split.

Here, we find that the sex difference effect is generally least dependent on N and the male sample for profile mean, stable for profile skewness and only moderately consistent for the gradient. Furthermore, the contrast with OC females, low estrogen and high progesterone females prove to be most stable across folds, yielding a mean consistency of higher than 0.9 for mean, higher than 0.8 for skewness, and higher than 0.7 for the gradient. Notably, these consistency values are higher than for our complete sample (Figure 2C).

Results, p. 8, ll.270

„To further interpret these sex-bias variations by hormonal group, we additionally investigate if first,
the mean sex-difference effect across parcels is conserved between group-comparisons (**figure 3B** and
**supplement 6**) and second, if the microstructural measure of any region also systematically varies in
an within-females comparison (**Figure 3D**). We furthermore added an internal consistency analyses to
determine the specificity of the reported effect on the male sample (**supplementary figure 7**).“

*Discussion, p. 18 || 577*

„We show that sex differences in all microstructural measures change in effect size or even disappear
if males are compared to females of certain estimated hormonal profiles, while randomly subsampling
the male group yields coherent results. This suggests that female sex hormones may play a role in
microstructural sex differences in the human cortex.“

**(5.3.1) Third, it would also be important to mention Sup Fig 4 results more in the main text here**
**here and use this to specify which regions are significant.**

Thank you for this suggestion. We decided to include part of Sup Fig 4 into the main figure about this
part of the study, Figure 3C; and add an additional analysis for this section which further helps to
understand regional specificity of results (Figure 3D). We revised the entire section (results, discussion
and supplement) to adjust it to your and the other Reviewers' comments. Please find key highlights of
these adjustments below:

*[...] We show that this is because intracortical profile skewness values of females who take OC*
*compared to NC females are significantly lower in precuneus, posterior and anterior cingulate, insula*
*and temporal pole (**Figure 3D**). These are the same areas in which the T1w/T2w skewness sex*
*differences are smaller if one compares males only to females who take OC (**Figure 3C**). This was*
*expected as the intracortical profile skewness in these areas is generally lower than in females,*
*demonstrating the more steep ratio of T1w/T2w signal intensity from pial to GM/WM surface in males.*
*Females in their low progesterone group hereby were most similar to OC females, while the high*
*estrogen and progesterone group seem to mainly drive these differences (**Figure 3D**).*

*“[...] However, parcel and cortical wide specific analysis give a more detailed overview of variations by*
*hormonal subgroups (**Figure 3C**; supplementary figure 7). The sex difference effect varied strongest*
*when comparing males to only OC takers versus comparing males to only females estimated to have*
*high progesterone levels: Sex differences between OC takers and males were least extreme (min*
*$d_{OC\ females} = -.4636$, max $d_{OC\ females} = .3134$), while sex differences between males and females in*

*their high progesterone phase showed particularly big positive and negative effect sizes (min*
*$d_{high\ prog\ females} = -.5980$, $max\ d_{high\ prog\ females} = .3398$)."*

*[...] Investigating the female differences more closely, we find that the insula's microstructural profile*
*covariance is closer with the fugal anchor of the gradient in NC than in in OC females; which seems to*
*be associated with by the low estrogen and low progesterone groups (Figure 3D).*

**(5.3.2) In this Sup Fig, I think it would be clearer if visualizations simply used 3 block colors given**
**these are port thresholding: white no sig effect and red/blue for sig effects in each direction.**

Thank you for this suggestion. It was indeed challenging to visualize the results in a fashion that is
clean and easily readable. We now include 3 analyses in this section, for which we provide 4
visualizations (figure 3B, 3C, 3D and supplementary figure 7): first, we show how the distribution and
mean of the overall effect size varies between group-comparisons (3B), second, we show how the
effect size varies across brain areas for three exemplary group-comparisons (3C) third, we show how
robust the microstructural differences in certain brain areas are by running a within-females group
comparison (Figure 3D), and fourth, we show how the effect-sizes between subgroup-comparisons
correlate with each other per parcel (supplementary figure 7).

However, we would like to avoid block colors for the following reason: In this part of the analysis, we
show differences in effect-size as well as significance. With the shading of blue and red, one can see if
the effects are stronger or weaker in the different group comparisons. Using block-colors would
prevent the reader from seeing the point of this analysis: the effect size changes depending on the
female subgroup. We hope this clarifies.

**(5.4) Fourth, are the different effects seen for different female subgroups reflecting changes in the**
**magnitude of the effects, or differences in interindividual variability between the different female**
**subgroups? This would be important to clarify empirically.**

Thank you for this valuable comment. This is one of the key changes we made to this
resubmission and we believe that with this piece of feedback, we could substantially improve the
results and robustness of this work. The main finding of this analysis is that there is a systematic
difference between females who take OC and females who naturally cycle. We discuss these results
both in relation to the sex-differences and hormonal grouping, add more explanation and dive deeper

into the details and interpretation of this result, such that its overall importance is further underscored
in the text.

Specifically, we followed up our initial analyses with seven additional GLMs in which we only
included females and then computed contrasts between the respective groups: naturally cycling vs.
taking OC, high estrogen vs. low estrogen, and high progesterone vs. low progesterone; OC vs high
estrogen; OC vs low estrogen; OC vs high progesterone, OC vs low progesterone. Note that since the
estrogen and progesterone groups are not mutually exclusive, we did not compute this contrast.

*results, p.9 ll.279*

[...] “To further interpret these sex-bias variations by hormonal group, we additionally
investigate if first, the mean sex-difference effect across parcels is conserved between group-
comparisons (**figure 3B** and **supplement 6**) and second, if the microstructural measure of any region
also systematically varies in an within-females comparison (**Figure 3D**). We furthermore added an
internal consistency analysis to determine the specificity of the reported effect on the male sample
(**supplementary figure 7**).

For the microstructural profile mean, only the OC-group replicated the average initial sex
difference effect (post-hoc contrast across 400 parcels between group comparisons n.s.; see
supplementary figure X for parcel-wise effect distribution by cortical type). [...] We found that the sex
bias in the average T1w/T2w microstructural measure was least stable in the occipital lobe (**Figure 3C**).
Here, the sex bias was particularly large when comparing males to females who took OC, but
disappeared for females in their low progesterone group. Accordingly, for an intra-females contrast,
we find that the occipital lobe of females who regularly take OC have a significantly lower T1w/T2w
profile mean than the occipital lobe of naturally cycling females, and in particular those grouped for
low progesterone (**Figure 3D**). The T1w/T2w profile mean of males is generally higher than those of
females, which explains the bigger sex differences when comparing males exclusively to OC females.”

[...]

“Investigating microstructural cortical layer skewness, the sex difference effects were most
different comparing males with OC vs. any NC female subgroup (for parcel-specific comparisons, see
supplementary figure 7). In fact, the previously reported sex difference in microstructural profile
skewness nearly disappeared when comparing males to females who regularly take OC (cortex-wide
average $d_{OC\ females} = 0.0788$, **Figure 3B**), and was even more pronounced when comparing males only
to females estimated to have high progesterone concentrations (cortex-wide average
$d_{high\ prog\ females} = 0.1995$). We show that this is because intracortical profile skewness values of
females who take OC compared to NC females are significantly lower in precuneus, posterior and

*anterior cingulate, insula and temporal pole (Figure 3D). These are the same areas in which the*
*T1w/T2w skewness sex differences are smaller if one compares males only to females who take OC*
*(Figure 3C). This was expected as the intracortical profile skewness in these areas is generally lower*
*than in females, demonstrating the more steep ratio of T1w/T2w signal intensity from superficial to*
*deep cortical layers in males. Females in their low progesterone group hereby were most similar to OC*
*females, while the high estrogen and progesterone group seem to mainly drive these differences*
*(Figure 3D)."*

**Figure 3. Comparing males to different female sub-samples, grouped by menstrual cycle phase.** (A)
 Estrogen and progesterone fluctuate with the menstrual cycle. Horizontal lines under the x-axis
 indicate grouping: purple reflects progesterone (dotted = low; solid = high); turquoise reflects estrogen
 (dotted = low; solid = high) (B) Hormones determine cortex-wide sex-difference effect sizes based on
 post-hoc contrast on cortex-wide effect sizes. Cohen's d per parcel is plotted separately for the three
 intracortical measures profile mean, profile skewness and the gradient, respectively for each sub-
 group-comparison. All shown contrasts were significant ($p < .001$). (C) FDR-thresholded Cohen's d maps
 of T1w/T2w profile mean (i) between males and subsamples of females divided by OC use and
 menstrual cycle phase projected on the cortical surface and the hippocampus. (ii) FDR-thresholded
 Cohen's d maps of T1w/T2w profile skewness between males and female subsamples mapped on the

cortex. (iii) FDR-thresholded Cohen's *d* map of differences in the microstructural gradient between
males and different female sub-samples. For completeness, all other FDR-thresholded Cohen's *d* maps
(all group-comparisons, for each of the three measures) are plotted in supplementary figure 4. D)
Microstructural differences between female groups, comparing OC females with all NC females, as well
as OC females with specific NC subgroups, divided by their hormonal period. Columns are the three
microstructural measures T1w/T2w mean, T1w/T2w skewness, and the microstructural gradient.
Purple areas are parcels which had significantly higher values for OC females, orange had significantly
higher values for NC females after FDR-thresholding (all Cohen's *d*).

**(5.5) Fifth, it is striking that the OC group show preservation of the full group effect for mean T1/T2,**
**but loss of the full group effect for skew. Moreover – the situation was not the same in the**
**hippocampus. These are very challenging dissociations to explain biologically. What thoughts do the**
**authors have ?**

Thank you for this important note. In fact, it was our wording that was misleading, while the
results are not contradictory in itself. To allow for a straight-forward interpretation, we added the
same ANOVA and post-hoc contrasts for the hippocampus as we did for the cortex-wide analysis (see
**supplementary figure 6**); and adjusted the wording in the text. These analyses clarify that in fact there
is a differentiation between mean and skew in the isocortex, but consistent changes in the isocortex
and hippocampus with respect to mean T1wT2w. Different effects between mean and skewness of
intracortical profiles point towards a differentiation between microstructural changes with respect to
sex differences and OC in superficial and deeper cortical compartments, possibly linked to sex
hormone receptor expression that has been reported to vary across cortical layers

Just as for the overall cortex-mean, the *mean* of the initial sex-difference effect across all
vertices in the hippocampus was the same only when comparing males to females who regularly took
OC, but not if comparing males to any of the NC female groups (post-hoc contrast n.s.). Visualizing the
parcels, however, it shows that this is most likely due to the fact that more negative and more positive
effects in the collapsed comparison cancel each other out, and thus lead to the same average effect
for both comparisons (see supplementary **figure 6**). Within the NC groups, there was no difference in
sex-bias if comparing males to females in their low or high estrogen group, but the low and the high
progesterone group were both significantly different from the initial, the OC, and the estrogen-group
effects. However, there was no significant group-difference in T1w/T2w mean in the hippocampus
between any female group.

Furthermore, the within-female groups analysis above also provides some clarity for this
question: More areas change their skewness with hormones than the mean T1w/T2w profile; so on a
parcel-wise level (but not average across the whole brain), skewness varies more with hormones than

the meant T1w/T2w measure. For the hippocampus, we were only able to analyze the profile mean,
 but not the profile skewness, since we couldn't build meaningful profiles between the outer and inner
 hippocampal layers due to technical limitations. Similar to the mean T1w T2w, we only see small
 changes in sex-difference effects. Contrasting NC and OC and high and low estrogen and progesterone
 females does not survive multiple comparisons in any parcel in the hippocampus, furthermore
 supporting the notion of smaller variations in T1w/T2w mean with sex hormones.

 **Supplementary Figure 6.** FDR-thresholded Cohen's d maps of T1w/T2w profile mean between males
 and subsamples of females divided by OC use and menstrual cycle phase projected on the unfolded
 hippocampus. On average, all effects are different from each other. Brackets (n.s.) on the right show
 where this is not the case, i.e. where the average effect across vertices replicates.

 *Discussion - hippocampus, from p.22, ll.716 [Adjustment]*

 “[...] Importantly, however, we couldn't identify a robust effect when computing inter-female contrasts
 for any region in the hippocampus. Thus, while we here show that taking the hormonal profile into
 account matters when investigating hippocampal-wide microstructural sex-differences, this study does
 not yield evidence for systematic hormone-related differences within females.

Overall, these findings extend previous work showing region-specific hippocampal sex
differences and variations in these effects in relation to sex hormones. Similar to previous studies we
again find that anterior-posterior differences within the hippocampus are substantial and need to be
considered (Masouleh et al. 2020, Genon et al., 2021). Through unfolding the hippocampus we
increased regional specificity, considering the morphology of the hippocampus⁶⁴. Further work
studying the impact of sex hormones on hippocampal structure may use similar techniques to capture
regional variation. “

**(5.6) Finally, given the complexity of this results section, I would suggest providing readers a “mini-
summary” with some key take aways at the end, around line 285.**

Thank you for this suggestion. We agree that this will be very helpful to clarify the main message of
this section. We add the following mini-summary after this section of the **results (p.11, ll. 358):**

*“To summarize, sex-differences in intracortical microstructural measures differ in effect size if
males are systematically compared to females roughly clustered in groups of different estimated
hormonal profiles. These variations are driven mainly by microstructural differences between naturally
cycling and regular OC intaking females and are most consistent for profile skewness. Between these
two groups, in particular the limbic, the prefrontal and the insular cortex showed strong differences in
profile skewness. Together, these results underline the importance of considering hormonal profiles
when investigating sex differences or sex-specific brain anatomy.”*

**(6.1) The section “Endocrine plasticity effects on intracortical structure spatially overlap with
cortical expression patterns of sex hormone related genes (Figure 4)”. First, was correction made
for multiple comparisons across genes and maps ?**

Thank you for spotting this crucial omission of ours - they were not. Our results don't remain
significant at a FDR-corrected threshold, which we now add explicitly in the text and discuss as a
limitation. We demonstrate, however, that instead of computing multiple tests, one can demonstrate
the link between the mean sex-difference map and sex-hormone-relevant transcriptomic maps with
a single multiple regression which we now include in the analysis.

Methods, p.30; ll.998

*“We followed a two-step procedure. First, we tested if hormone-related genes overall were*
*related to the sex-difference maps by running a multivariate regression including all transcriptomic*
*maps. To test for significance, we randomly permuted the sex-difference maps 1000 times, and ran a*
*multivariate regression each, computing a distribution of F-values. In the end, we computed the spin-*
*corrected p-value by computing the proportion of permuted F-statistics that are greater than the*
*original F-statistic. Second, we tested the relationship between the individual genes and the sex-*
*difference maps of each microstructural measure. We computed spearman correlations between gene*
*expression enrichment for each of the selected GOIs with the observed differences in cortical*
*microstructure between males and females. To control for spatial autocorrelations of gene enrichment*
*analysis due to spatial non-independence of brain maps, we tested for significant spatial overlap*
*between the respective transcriptomic map relative to randomly spun phenotype maps (i.e. our effect*
*maps of sex differences). For that, we adjusted the spin-test function from the ENIGMA toolbox, so that*
*spherical representations of the unthresholded three phenotypic maps were randomly spun in 1000*
*permutations and correlated with the 25 transcriptomic maps of our GOIs (Alexander-Bloch et al.,*
*2018). This procedure accounts for spatial autocorrelations by leveraging the spherical representations*
*of the cerebral cortex. We report the frequency in which the true correlation between phenotypic maps*
*and genes exceeded a test statistic generated of correlation values from randomly permuted*
*phenotypic maps as spin-p-value. To account for multiple-tests, we furthermore compute FDR-*
*thresholds for each of these spin-p values. Additionally, to provide a measure of genetic specificity, we*
*generated a measure of “brain-gene-baseline” and tested our effects against the baseline. We built*
*the baseline transcriptomic map by extracting the principal component of all available transcriptomic*
*maps in the left hemisphere. We provide spatial correlations (spearman) between phenotypic maps of*
*sex differences in profile mean, skewness and gradients with the brain gene baseline as a reference.”*

Results, p.12 ll. 377:

*“[...] We thus next asked whether transcriptomic maps of 25 sex steroid relevant genes were*
*generally linked to sex-difference effect maps for each microstructural measure (Cohen’s d of sex*
*differences in microstructural profile mean, skewness and covariance gradient), and then tested for*
*each of these 25 gene individually if they spatially overlapped with our microstructural sex difference*
*maps. Please note that none of these individual links was significant at a FDR-corrected threshold, and*
*should thus not be considered more than trends.*

*We found that sex-hormone related genes were enriched in areas in which we found sex-*
*differences in microstructural mean ($F(336, 310) = 6.6, p_{spin} < .05$), but not in microstructural profile*
*skewness ($F(336, 310) = 3, n.s.$) or the microstructural gradient ($F(336, 310) = 1.9, n.s.$).*

[...]

**Figure 4. Spatial overlap between effect maps of sex differences for the microstructural gradient,**
**profile mean and profile skewness.** Transcriptomic maps of genes are sorted by categories: sex
hormone synthesis related genes, androgen receptor related, estrogen receptor related genes, and
progesterone receptor related genes. We test for spatial specificity by comparing against the principal
component of all genes (baseline). Shades of red represent positive r -values, shades of blue represent
negative correlations; circle size and shading indicate size of correlation. Values with significant p -
values after permutation spin-testing are marked with a black outline. **Note that no correlation is**
**significant when accounting for multiple testing at an FDR-threshold.**

**(6.2) Second, for most of genes examined, their spatial correlations with the three metrics did not**
**reach statistical significance as it, i.e., $p_{spin} < 0.05$. However, the authors describe these**
**nonsignificant findings as in line 311, Strong overlap were additionally presented by the androgen**
**receptor gene AR ($r = -.31$, $P_{spin} = .15$) and the progesterone receptor PGRMC1 ($r = .26$, $P_{spin} = .20$),**
**and using them as support (line 536 to 538) to draw a conclusion, strongly overlapped with sex**
**hormone gene expression levels (line 624). This should be reworded.**

We reworded it as follows:

Results, p.12, ll. 385:

"Testing each transcriptomic map individually, we identified **medium sized correlations, but**
**not significant after spin-testing,** between sex-differences in microstructural profile mean and the
transcriptomic map of the androgen-receptor activation related genes SRD5A3 ($r = .31$, $p_{spin} = .07$)
and AKR1C3 ($r = -.30$, $p_{spin} = .11$), the androgen receptor gene AR ($r = -.31$, $p_{spin} = .20$) and the
progesterone receptor PGRMC1 ($r = .26$, $p_{spin} = .17$). We further found a **significant after controlling**
**for spatial auto-correlation, but small spatial overlap** with the sex steroid precursor gene HSD17B3 (r
$= .13$, $p_{spin} < .05$).

Sex-bias in T1w/T2w microstructural profile skewness demonstrated **small spatial associations**
with Progesterone Immunomodulatory Binding Factor 1 (PIBF1, $r = -.25$, $p_{spin} < .05$), the estrogen
receptor 1 (ESR1, $r = -.18$, $p_{spin} < .05$), the estrogen receptor beta (ESRB, $r = -.22$, $p_{spin} < .05$), and the Growth

*Regulating Estrogen Receptor Binding 1 (GREB1, $r = -.24$, $p_{spin} < .05$). There was a moderate but non-*
*significant (after permutation tests) correlation between skewness sex-differences and the estrogen*
*receptor alpha (ESRA, $r = -.24$, $p_{spin} = .27$) and the estrogen related receptor gamma (ESRG, $r = -.22$,*
*$p_{spin} = .23$). Lastly, sex differences in skewness also moderately overlapped with the sex-hormone*
*synthesis relevant gene AKR1C3, which was not significant after controlling for spatial auto-correlation*
*($r = .31$, $p_{spin} = .05$). The gene specificity for profile mean and the profile skewness sex difference was*
*supported by a non-significant and negligible correlation with the baseline gene map we extracted.*
*This was, however, not the case for the microstructural gradient, which correlated stronger with the*
*baseline gene factor than with any other transcriptomic map ($r = -.28$, $p_{spin} < .05$, significant at FDR-*
*corrected threshold)."*

Discussion

*"To support the evidence of our first endocrine analysis, we added a second, independent one. We*
*show that the differences that we systematically observe between males and females present*
*moderate overlap with areas of elevated expression levels of sex hormone related genes.*

[...]

*Importantly, while our analyses demonstrate a general link between sex-hormone specific*
*genes and the microstructural mean, gene specificity for sex steroid synthesis and sex hormone*
*receptor genes, and account for auto-correlations, the links to individual hormones were not significant*
*at an FDR threshold, controlling for number of genes and measures. [...]"*

**(6.3) Third, there is insufficient attention given - in analytic design, presentation of results and**
**discussion of results - to the fact that the AHBA dataset contains only one female. Therefore, all the**
**expression maps examined are predominantly from the 5 male donors. Analytically, it would be**
**important to provide some evidence that the reported connections between imaging and**
**transcriptomics are at least trending in the same direction when expression maps are based on the**
**single female donor. It is also important to say much more in Discussion and Limitations regarding**
**the problem of sex imbalance in AHBA and what it means the authors can and can't say regarding**
**their results.**

This is a very valuable comment, we thank the Reviewer for pointing out this limitation. We followed
up the initial analysis by separating the AHBA dataset by sex of its donors and computed if the overlap

between sex difference maps with only female, only male, and all AHBA donors correlate with each
other:

- ● Overlap between results based on female only and male only
 - ○ gradient Spearman's $r = 0.0603$
 - ○ mean Spearman's $r = 0.5119$
 - ○ skewness Spearman's $r = 0.6028$
- ● Overlap between results based on female only and all AHBA donors
 - ○ gradient Spearman's $r = 0.2$
 - ○ mean Spearman's $r = 0.4638$
 - ○ skewness Spearman's $r = 0.7754$
- ● Overlap between results based on males only and all AHBA donors
 - ○ gradient Spearman's $r = 0.7688$
 - ○ mean Spearman's $r = 0.7964$
 - ○ skewness Spearman's $r = 0.8172$

This result indicates that albeit, as expected, the results are mainly driven by the male donors, there
is moderate to high overlap when only considering the female donor with the initial results for
skewness. Being the most robust result throughout all analyses, this further supports the notion that
the skewness-sex differences are indeed related to sex hormones. The limitation for considering only
the female donor is that with a dataset of $n = 1$, we are fully susceptible to the individual specifics of
that one donor. We add the following sections in methods, results and discussions on this matter:

**Methods, p. 31, from II 1005:**

*"Lastly, to account for the sex-imbalance in the AHBA dataset (one female and five male donors), we*
*reran the analysis as described above separately for the male and female donors only (supplementary*
*figure 8). We then computed Spearman's rank correlation to test if results statistically trend in the*
*same direction."*

**Results, p. 13, from II. 396:**

*"Note that the AHBA dataset from which we derived the transcriptomic maps is composed from only*
*one female and five male donors. We thus tested if the results identified here trend in the same*
*directions if rerunning the analysis with the female or male donors only (supplementary figure 8). We*
*find that this is the case (profile skewness: $r_{female-all} = 0.7754$; $r_{female-male} = 0.6028$)."*

**Discussion, p. 21, from ll. 660:**

*“Furthermore, even though our analyses suggest that these results are broadly similar across*
*sex of the six donors that make up this transcriptomic sample, it will be important to revisit this analysis*
*once a sex-balanced dataset becomes available.”*

**Only female donor.**

**Only male donors.**

**Supplementary Figure 8. Spatial overlap between effect maps of sex differences for the**
**microstructural gradient, profile mean and profile skewness, split by AHBA donor-sex.** Top and
bottom are the same analysis, but considering only the female (top) and male (bottom) AHBA donors
to derive the transcriptomic maps. Transcriptomic maps of genes are sorted by categories: sex
hormone synthesis related genes, androgen receptor related, estrogen receptor related genes, and
progesterone receptor related genes. We test for spatial specificity by comparing against the principal
component of all genes (baseline). Shades of red represent positive r-values, shades of blue represent
negative correlations; circle size and shading indicate size of correlation. p-values < 0.05 after
correcting for auto-correlations using spin-testing are marked with a black outline.

**(6.4) Finally, the corresponding Discussion section title should be changed to “Transcriptomics”**
**decoding rather than “Genetic decoding”. The authors are not looking at genetic variation.**

This is of course correct, thank you for spotting this error. We changed the heading in the Methods
section from “Genetic Decoding” to “Transcriptomics”.

**(7.1) Sections comparing sex difference to cytoarchitecture and cerebral blood flow. The question**
**of multiple comparisons comes up here too.**

As for the genetic analysis, we add FDR adjusted Benjamini-Hochberg significance thresholds to the
cytoarchitectural results, as well as to the cerebral blood flow results and report this accordingly. This
did not change the cytoarchitectural result. None of the cerebrovasculature correlations were
significant at an FDR-threshold.

*Cytoarchitecture:*

*[...] As before, we report statistical correlation values and the respective max-permutation test p-value*
*after spherical spin-tests (p-spin), and indicate if they remain significant at a FDR-corrected threshold.*

*We found that the effect maps of sex differences in microstructural skewness and the*
*microstructural gradient significantly correlated with the hierarchy of cortical types at a FDR-corrected*
*threshold, but not for microstructural mean (Figure 5B).*

*Cerebrovascular control analyses:*

*[...] "In addition to including intracranial volume as a covariate in every linear model, we thus tested if*
*the relation to sex hormone concentration would covary with the local density of cerebral vasculature*
*(supplementary figure 9). Since no correlation remains significant at a FDR-corrected threshold, we still*
*report spin-permutation corrected p-values (p-spin)."*

**(7.2) Also, the authors discuss similar correlations with some inconsistency. For example in the**
**section starting on line 336, the authors state Sex differences differ in strength as a function of**
**cytoarchitectural type (Figure 5), show that A positive correlation between T1w/T2w profile**
**skewness ($r = .20$, $P_{spin} < .05$) and cortical types (line 348) and sex difference effects in the**
**microstructural gradient showed moderate overlap with the hierarchy of cortical types ($r = .14$,**
**$P_{spin} < 0.05$). Then on the section starting in line 367 the authors find some similarly sized**
**correlations with maps of cerebral vasculature. However, these similar correlations are interpreted**
**in different ways in the Discussion section, where the relationship with cytoarchitecture is treated**
**a positive finding, whereas the vasculature correlations are downplayed. For example: we provide**
**evidence that the observed effect was not confounded with hormone-induced fluctuations in**
**cerebrovascular blood flow (line 412), The moderate overlap mean T1w/T2w effects with cerebral**
**vein density furthermore (line 430), and Adding to this, we found that this measure was not affected**
**by vasculature (line 625). It would be important to address such imbalances in interpretation.**

Thank you for pointing out this imprecision. It is important to us to not give the impression of artificially
downplaying or inflating our results to our subjective liking, so we appreciate the feedback and hope
we were able to address it accordingly. In particular after adding the Benjamini-Hochberg FWE
correction to all correlative results, we adjusted the wording such that in particular the genetic results
should be considered with care. We furthermore coherently use wording such that correlation values
adhere to effect-size conventions as suggested by Cohen (1988) which we hope now overall improved
the consistency in which we discuss results and their respective effect.

*Discussion p.20, ll 664*

*“To support the evidence of our first endocrine analysis, we added a second, independent one.*
*We show that the differences that we systematically observe between males and females present*
*moderate overlap with areas of elevated expression levels of sex hormone related genes. This offers a*
*translation of a recent rodent study to humans, where sex differences in brain structure occurred*
*particularly in regions enriched in sex hormone genes⁹³, and furthermore yields the second piece of*
*evidence that sex hormones contribute to sex-bias in human intra-cortical microstructure with a*
*completely independent hormonal analysis. [...]*

*Importantly, while our analyses demonstrate a general link between sex-hormone specific*
*genes and microstructural skewness, gene specificity for sex steroid synthesis and sex hormone*
*receptor genes, and account for auto-correlations, the links to individual hormones were not significant*
*at an FDR threshold. [...]*”

We furthermore now address the overlap between cerebrovasculature with sex differences in profile
mean and with hormonal subgroup sex differences for profile mean and the gradient in the discussion
as a limitation of the first hormonal analysis, for example:

*p. 17, ll.518*

*[...] „On the other hand, the combination molecules that determine mean T1w/T2w signal intensity*
*also make it the most prone to confounds, such as transmit bias field effects (Glasser et al., 2022), and*
*sex hormone effects on cerebral fluids. The moderate (but non-significant) overlap between mean*
*T1w/T2w effects with cerebral vein density furthermore might reflect an interaction with the effect of*
*venous blood on T₂w signals (Sedlacik et al., 2008). Since profile skewness and the microstructural*
*gradient are based on relative variations of T1wT2w, they do not suffer from the same limitations.“*

*p.23, ll.749*

[...] To limit these uncertainties, we firstly included intracranial volume as a covariate in each of our
linear models, statistically controlling for hormone-induced volume fluctuations; and secondly, we
quantified the overlap between cerebral vasculature and the areas in which identified sex difference
effects. The correlation with hormonal mean T1w/T2w profile, but not skewness supports the notion
that T1w/T2w signal may indeed be globally modulated by the effect that sex hormones exert on
water-balance and lipid metabolism.

**(8) Discussion. Line 420 “The male cortex was characterized by ...” This suggests a typology (which**
**the authors carefully push back against themselves in authors note) so should be reworded.**

Thank you for pointing out our blind spots, it is extremely important to us to avoid these wordings.

We changed it to: p. 17, ll.514

“[...] We found systematic differences in all three microstructural measures when dividing the group
into self-reported males and females. First, we found the average T1w/T2w signal intensity to be higher
in the largest part of the male cortex, except for bilateral insular and medial temporal areas [...]”

**Minor Comments:**

**(1) Line 246, should cortex-wide averaged high estr female-male = -0.12846 be high progesterone**
**female-male?**

Yes, the first value is for low estrogen, the second for high progesterone. We corrected the typo in the
manuscript.

“This was especially evident for females who were estimated to have low estrogen or high
progesterone levels at the time point of imaging (cortex-wide average $d_{low\ estr\ females - males} = -$
0.1176 ; cortex-wide average $d_{high\ prog\ females - males} = -0.12846$).”

**(2) Line 250, We found that sex differences in the cingulate cortex, the insula, the orbitofrontal**
**cortex and the hippocampus were most affected by the menstrual cycle phase and exogenous sex**
**hormone intake (Figure 3C). It is hard to see these regional differences from three comparisons side-**
**by-side in Figure 3C. Just a suggestion, running a separate anova model in females alone, a F test**
**map of group effect in either all five subgroups or the three groups in Figure 3C (taking OC, low**

**estrogen, and high progesterone) across 400 parcels may help to illustrate region variations in the**
**menstrual cycle phase.**

We indeed now include an additional within-females analysis, which we add to Figure 3. We hope that
the updated Figure 3 (see page 18) addresses this comment.

**(3) The display of labels in Supplemental Figure 4 seems off, like FDR corr. Cohen's d for contrast**
**Men vs high estr, fo.**

All supplementary figures have been corrected.

**(4) In Supplemental Table 5, it is surprising to see Pspin < P for spatial correlations between gene**
**expression and sex differences in three metrics. I would expect spin tests are more stringent,**
**yielding larger Pspin values.**

Thank you for this comment. Our previous notation was misleading: if no spin-test was computed, the
table indicated '0' for the p-spin of the respective measure. To avoid confusion, we now emptied these
fields in the supplementary table and added in our methods-section that we compute spin-
permutation distributions for *“for every correlation value that has a lower uncorrected p-value than*
*0.05.”*

**(5) Line 372, We found that sex-differencers should be sex-differences.**

Thank you. Adjusted.

**(6) Line 428. Typo “the combination molecules”.**

Thank you. Adjusted to:

*“On the other hand, the combination of molecules that influence mean T1w/T2w also make it the most*
*prone to confounds, such as transmit bias field effects 78, and sex hormone effects on cerebral fluids”*

**(7) Line 430. Typo “moderate overlap mean T1w/T2w effects”**

Thank you. Adjusted to:

*“The moderate overlap of mean T1w/T2w effects with cerebral vein density furthermore might be an*
*interaction with the effect of venous blood on T₂w signals⁷⁹.”*

**(8) Line 524 - should be “large” rather than “big”**

Thank you. Adjusted to:

*“However, since we benefit from a large sample size and a second, independent hormonal analysis,*
*our results underscore the importance of moving beyond a generalized understanding of sex*
*differences and considering hormonal profiles as a crucial factor in interpreting and explaining these*
*differences.”*

**Reviewer #2 (Remarks to the Author):**

**This article analyzes the cross-sectional MRI images of 992 young subjects from the Human**
**Connectome Project. It calculates regional variation in cortical microstructure based on the T1/T2**
**ratio and analyzes how these metrics differ based on sex and menstrual phase (using self-reported**
**days since menstruation). The authors also assess the spatial correspondence of MRI-derived maps**
**with ex-vivo maps of sex hormone receptor gene expression.**

We thank the Reviewer for the appreciation of our work and the insightful comments, which we have
addressed below.

**Although the results are very interesting, we have the following concerns:**

**The most important concern is that, although the authors state in the discussion that "It is important**
**to note that instead of longitudinally following microstructural changes associated with hormonal**
**variations within individuals, we computed inter-individual contrasts based on an indirectly**
**approximated correlative hormonal measure. Therefore, we interpret our results as tendencies that**
**highlight the importance of considering the complexity of hormones in the study of brain structure.**
**However, due to our large sample size and a second, independent hormonal analysis, our results**
**emphasize the importance of moving beyond a generalized understanding of sex differences and**
**considering hormonal profiles as a crucial factor in interpreting and explaining these differences",**
**the abstract and the paper are full of terms such as "influence of sex hormones (conclusion)" or**
**"endocrine neuroplasticity". This can lead to an over-interpretation of the results. We suggest that**
**the abstract and conclusion clearly reflect the cross-sectional nature of the MRI data and the**
**absence of hormonal measures, and avoid terms such as "influence of sex hormones", "endocrine**
**plasticity", etc., when discussing their own data.**

Thank you for this comment. We revised the manuscript in a manner that avoids wording which
implies influence and causality, and underlines the correlative and indirect manner of the analyses
more. We now avoid words that imply causality, don't refer to our results as influence of hormones
on brain structure or as endocrine plasticity, and added limitations where necessary. We adjusted our
framing of the study such that we aim to contextualize sex differences, rather than investigate
'endocrine plasticity'. We marked these changes in the revised manuscript. Here are a few examples
of our adjustments of a more careful phrasing:

Abstract:

“[...] Investigating quantitative intracortical profiling in-vivo using the T1w/T2w ratio in 1093 healthy
females and males of the cross-sectional Human Connectome Project young adult sample, we found
that regional cortical and hippocampal microstructure differed between males and females, and that
the effect size of this sex-bias varied depending on self-report hormonal status in females.

[...]

Albeit correlative, our study underscores the importance of incorporating sex hormone variables into
the investigation of brain structure and plasticity“

Introduction:

“To understand the source of systematic structural variations and its implications, it is crucial to further
contextualize observed sex-differences, going beyond a sex binary. [...] Out of these, activational sex
hormone levels have a particularly strong and dynamic effect on influencing a sex-specific phenotype
(Blencowe et al., 2022; Gegenhuber et al., 2022; Rehbein et al., 2021; Romeo et al., 2004; de Castilhos
et al., 2008, Arnold & Breedlove, 1985; ...). In an effort to bridge traditional neuroanatomy and
neuroimaging, we here investigated sex differences in intracortical microstructure in-vivo based on the
ratio of T1- over T2 weighted (T1w/T2w) MRI intensities, and how these sex differences could be
systematically linked to gonadal hormones specifically.”

[...]

“There has not been a characterisation of human cortical microstructure sex differences in
vivo, and it remains elusive if sex hormones might play a role in these variations.”

[...]

“We then contrasted these microstructural measures between females and males, tested how these
sex-differences vary if systematically comparing males with females of particular hormonal profiles
(approximated by self-reported menstrual cycle phase and OC use) and quantified how these effects
overlap with transcriptomic maps of sex-hormone related genes.”

Discussion:

“To put the identified sex-differences into context, we investigated a potential link between these
effects and sex hormones with two orthogonal analyses. We show that sex differences in all
microstructural measures change in effect size or even disappear if males are compared to females of
certain estimated hormonal profiles, while randomly subsampling the male group yields coherent

results. This suggests that female sex hormones may play a role in microstructural sex differences in
the human cortex. We furthermore demonstrate that there is a particularly big difference in cortical
microstructure between females who take OC and naturally cycling females, as supported by
significant within-females effects.”

“Similarly, despite moderate correlation effect sizes, none of the transcriptomic map results remain
significant at a FDR-threshold. We thus merely interpret our results as tendencies which underline the
importance of considering the complexity of hormones in the study of brain structure. However, since
we benefit from a big sample size and thoroughly analyzed the microstructural sex differences with
two independent hormonal analyses, we stress the importance of moving beyond a simple binarized
understanding of sex differences and towards considering hormonal plasticity effects as crucial factors
when investigating brain structure. “

“In this study, we investigated if sex-biases in three microstructural cortical measures- an average
measure of cortical microstructure, a proxy for laminar differentiation within the cerebral cortex and
the microstructural gradient - could be linked to sex-hormones, with two complementary correlative
analyses in a large cross-sectional sample.”

**Introduction:**

**We believe that the writing of the paper would benefit from narrowing and focusing the**
**introduction, especially if this article is intended to be directed to the readers of a broad-scope**
**journal such as Nature Communications. Along the same line, we believe that the introduction**
**would benefit if the authors explain the biological interpretation of the extracted brain metrics to**
**make it more accessible to a non-expert scientific audience.**

Thank you for giving us the chance to convince the audience of the value of our manuscript with a
more narrowed and focused introduction. The major adjustment we made is the first paragraph,
where we are trying to slowly introduce the audience to the topic, incorporating the previously
mentioned more careful framing of the role of gonadal hormones. We furthermore re-ordered the
following paragraphs, starting with an introduction to the brain metrics to make it more accessible to

a non-expert scientific audience. Here, we paste the major changes. Please refer to the updated
manuscript for the complete introduction.

*“Determining sex and gender differences in brain structure is of great societal interest to ultimately*
*improve diagnostics and treatment of brain-related disorders. While macro-scale morphometrical sex*
*differences are well documented, intracortical microstructural differences between sexes have not yet*
*been characterized. To understand the source of systematic structural variations and its implications,*
*it is crucial to further contextualize observed sex-differences, going beyond a sex binary. Underlining*
*the overly simplified nature of a sole division into a self-reported sex-binary, sex differences are*
*determined by a complex combination of societal and epigenetic factors (McCarthy et al., 2009; Ratnu*
*et al., 2017), sex chromosomes (Liu et al, Ratnu et al) and gonadal hormones (Barha & Galea, 2010;*
*Been et al., 2022; Cooke & Woolley, 2005; Hara et al., 2015; Patel et al., 2013; Woolley & McEwen,*
*1993). Out of these, activational sex hormone levels have a particularly strong and dynamic effect on*
*influencing a sex-specific phenotype (Blencowe et al., 2022; Gegenhuber et al., 2022; Rehbein et al.,*
*2021; Romeo et al., 2004; de Castilhos et al., 2008, Arnold & Breedlove, 1985; ...). In an effort to bridge*
*traditional neuroanatomy and neuroimaging, we here investigated sex differences in intracortical*
*microstructure in-vivo based on the ratio of T1- over T2 weighted (T1w/T2w) MRI intensities, and how*
*these sex differences could be systematically linked to gonadal hormones specifically.*

*Human brain structure is most commonly characterized in-vivo by determining the macro-*
*scale morphometry of the cortex. Analyses of volume- or thickness- variations based on the inner and*
*outer cortical boundaries, however, are blind to microstructural variations within the cortical sheath.*
*Microstructural changes within the cortical sheath are traditionally examined post mortem using cell-*
*staining procedures⁴³⁻⁴⁵. On this micro-level, the human cortex is structured into several cell layers.*
*The amount and prominence of each layer as well as the sharpness of their boundaries varies across*
*the cortex, so that cortical areas can be classified into different types according to their laminar*
*elaboration^{44,46,47}. These variations in cortical types are systematically linked to the cortex' inherent*
*property of plasticity^{46,48}, such that simpler laminar structures (e.g. paralimbic structures) are*
*hypothesized to be more plastic than highly elaborate areas (e.g. primary visual cortex)^{48,49}. Amongst*
*others, one explanatory factor for this covariation of laminar differentiation with plasticity is the*
*amount of intracortical myelin, which inhibits plasticity in the brain⁵⁰⁻⁵⁵. Intracortical myelin content*
*correlates with laminar differentiation so that more elaborate laminar architecture is characterized by*
*higher intracortical myelin content and higher stability^{48,56}. Lastly, gradients of microstructural*
*variation running along major axes of organization in the cortex support variation in brain function⁵⁷⁻*
*⁵⁹. Multiple neuroanatomical accounts have illustrated the intrinsic link between microstructural*

*properties, inherent brain organization principles, and brain function* ³⁹⁻⁴². Thus, examining variations
*in i) microstructural tissue properties, ii) cortical lamination and iii) the microstructural inter-regional*
*organization in-vivo will yield a more specific understanding of sex differences in brain structure.*

[...]

**Methods:**

**We recommend authors to include the Freesurfer-derived Euler Number as an additional covariate**
**in the models, along with intracranial volume, age, and sex, to control for motion-related data**
**quality.**

Thank you for this suggestion. We added the euler number as a covariate in our analysis which did not
change our results. We marked this in the methods:

**Methods, p. 29, ll. 886**

*"Since the microstructural measures exhibit small to moderate correlations with intracranial volume*
*(ICV, see supplementary figure 10), in each model we accounted for ICV, as well as age and the euler*
*number as a movement-related data quality measure:*

*T1w/Tw2 measure (parcel) ~ b0 * 1 + b1 * sex + b2 * age + b3 * ICV + b4 * euler_no"*

**We believe that authors should provide a clearer description of how they categorize the groups of**
**interest, specifically females and males. The authors explain the criteria for classifying the female**
**category (self-reported as females and being or having been menstruating), but they do not specify**
**how they classify males. We assume that the male categorization follows the same logic as the**
**female category (self-reported as males and not menstruating), leaving outside other categories**
**(self-reported as females and not menstruating or self-reported as males and being or having been**
**menstruating), but this should be explicitly stated. Also, authors sometimes mix the terms sex**
**(female/male/intersex) and gender (women/men/other genders). For instance, when they define**
**the female category, they state, "We classified individuals of female sex if they self-reported their**
**gender as female and indicated that they are or have been menstruating in their lives." We believe**
**that a more appropriate definition should be: "We classified individuals of female sex if they self-**
**reported their sex as female and indicated that they are or have been menstruating in their lives."**
**Authors should homogenize the use of the terms males/females vs men/women throughout the**

**manuscript. We suggest sticking to the male/female categories since this article focuses on sex-**
**specific factors rather than gender.**

We thank the Reviewers for this remark. We agree that due to the focus on biological sex and the lack
of focus on societal variables, we want to avoid “men” and “women”, and thus homogenized the text
such that we exclusively use the words “males” and “females”.

Our grouping in male and females followed the two items collected in the HCP dataset: One of their
items is “gender” - a binary self-report of ‘female’ and ‘male’; another item is ‘menstrual age began’.
We used a combination of the item the HCP authors call ‘gender’ with ‘menstrual age began’ to divide
the sample into what we term ‘sex’. There were no individuals who self-identified as male and
menstruated. In our methods section, we add the following clarification:

Methods, p. 26, from ll. 778:

*“We classified those individuals as females who reported a female gender and are or have been*
*menstruating in their lives, and all others as male. Note that all datasets collected in this study fall into*
*one of these two categories, but that we distance ourselves from a sex- and gender-binary. We*
*speculate that a more precise classification into gender and sex might lead to re-classification of some*
*individuals, and take this into account as a source of random noise.”*

Please also note our footnote on p.3:

*“In this manuscript, the terms ‘female and male sex’ refers to a combination of self-reported binary*
*gender and the report of having menstruated in one’s life. The authors appreciate the complexity of*
*biological sex and the influence of gender on biology, and do not postulate a sex binary. “*

**If we understand correctly, the authors are parcellating the cortex into 12 sections. However, this**
**parcellation is based on the information provided by approximately 4 voxels (as estimated by the**
**voxel size of HCP images and the mean cortical thickness). We assume that the authors might have**
**interpolated some of the values. In the same line, is the number of voxels different depending on**
**the orientation of the perpendicular line used to calculate the layers? How does this might affect**
**the calculated metrics, especially the skewness?**

Yes, the Reviewer is correct: the microstructural measures used in this study do rely on interpolation
of data points (similar to classical histological analyses, albeit this field is in need of down-sampling),
which is a clear limitation of our study. However, the approach has been previously successfully
validated in an ultra-high resolution ex-vivo histological dataset, recovering highly consistent brain
maps based on quantitative microstructural profiling in MRI and histology (Paquola et al., 2019 PLOS).
Moreover, we take a parcellation approach and that is naive to the structure of the cortex, but rather
is based on local and global functional organization. Yet irrespective of this, it is possible that the
number of voxels may impact the skewness, yet the number of voxels included randomly varies across
the cortical mantle due to the curved shape of the human cortex. Moreover, maps of skewness look
fairly smooth and we furthermore don't have reason to expect that potential problems arising with
the interpolation may bias males and females in diverging ways and thus in the context of the current
study, we interpret this as a factor of random noise.

We add the following sentence to our **discussion, p. 17, from ll. 479:**

*"This approach has been inspired by traditional cyto- and myeloarchitectonic metrics. While it requires*
*interpolation of data points in the cortical sheath cross section, it has previously been validated with*
*an ultra-high resolution cytoarchitectural ex-vivo dataset (Paquola, Wael, et al., 2019). We first [...]"*

**Discussion:**

**One strong point of the article is that it detects sex differences in brain structure when grouping**
**individuals into the female-male categories. However, when dividing females into the five sub-**
**group categories, these sex differences only replicate in the OC users. We believe this should be**
**further discussed and treated as one of the main results of the article, especially since the authors**
**disclose at some points that studies that merely test sex differences are over-simplistic and that**
**considering sex-specific factors such as hormonal levels is essential.**

We thank the reviewer for this suggestion. In fact, all reviewers had comments about this part of the
paper, offering several ideas on how to make the results easier to interpret, and how to underline its
importance. We now take a step-wise approach to interpret these findings and discuss them
appropriately:

- - as before, create subgroups and repeat male vs. female contrast only considering subgroups
of females
- - check consistency by randomly subsampling males and repeating the analysis
 - - create an average effect over all parcels, and with ANOVA check if *on average* the
effect-size changes between subgroup GLMs
 - - on a regional level, report in which parcels we observe variations in effect size
- - create a between-female subgroups contrast to explain why sex-difference effect size changes
- We discuss these results both in relation to the sex-differences and hormonal grouping, add more
explanation and dive deeper into the details and interpretation of this result, such that its overall
importance is further underscored in the text.
- Updated results figure:

Figure 3. Comparing males to different female sub-samples, grouped by menstrual cycle phase. (A) Estrogen and progesterone fluctuate with the menstrual cycle. Horizontal lines under the x-axis indicate grouping: purple reflects progesterone (dotted = low; solid = high); turquoise reflects estrogen (dotted = low; solid = high) (B) Hormones determine cortex-wide sex-difference effect sizes based on post-hoc contrast on cortex-wide effect sizes. Cohen's d per parcel is plotted separately for the three intracortical measures profile mean, profile skewness and the gradient, respectively for each subgroup-comparison. All shown contrasts were significant ($p < .001$). (C) FDR-thresholded Cohen's d maps of T1w/T2w profile mean (i) between males and subsamples of females divided by OC use and menstrual cycle phase projected on the cortical surface and the hippocampus. (ii) FDR-thresholded Cohen's d maps of T1w/T2w profile skewness between males and female subsamples mapped on the

1082 cortex. (iii) FDR-thresholded Cohen's d map of differences in the microstructural gradient between
1083 males and different female sub-samples. For completeness, all other FDR-thresholded Cohen's d maps
(all group-comparisons, for each of the three measures) are plotted in supplementary figure 4. D)
Microstructural differences between female groups, comparing OC females with all NC females, as well
as OC females with specific NC subgroups, divided by their hormonal period. Columns are the three
microstructural measures T1w/T2w mean, T1w/T2w skewness, and the microstructural gradient.
Purple areas are parcels which had significantly higher values for OC females, orange had significantly
higher values for NC females after FDR-thresholding (all Cohen's d).

Discussion:

“We found that the cortical microstructure of males and females differ regionally in each of these
microstructural measures. The effect size of the observed sex-differences depended on the estimated
estrogen and progesterone levels of females at the time of the brain scan. In particular, we observe
systematic differences between NC and OC females in all three microstructural measures. We
furthermore find that the measure of microstructural skewness, being a proxy measure of laminar
differentiation, proves particularly robust for several control analyses, and furthermore spatially
overlapped with expression levels of sex-hormone-relevant genes.”

“In contrast to the mean microstructural intensity, the sex-difference effect in microstructural
skewness was driven by NC females, while OC females exhibited profiles more similar to males. The
low estrogen, low progesterone, and high estrogen groups all replicated the initial sex difference in the
dominance of higher versus lower cortical compartments intensity. However, the effects were different
from the main effect when examining females who regularly took oral contraceptives or had high
progesterone concentrations. Specifically, there was nearly no difference in lamination between males
and females who took OC (weak average effect), but there was an even stronger average difference in
lamination between males and females with high progesterone concentrations. OCs suppress
circulating estradiol and progesterone levels⁸⁴⁻⁸⁶. Though no study to date has investigated such
effects, we draw analogies between a recent morphological study focussing on the medial temporal
lobe and its link to progesterone as well as chronic progesterone suppression (such as OCs): here
progesterone was shown to shape MTL volume throughout the menstrual cycle, and ceases to do so
when suppressed⁸⁷. Speculatively, this effect might appear through progesterone's effect on
myelination⁸⁸⁻⁹⁰. The variations we observed were mainly driven by stronger effects in the prefrontal,
anterior cingulate and tempo-parietal areas, which are explained by robust differences in skewness in
these areas between females who take OC and any NC female subgroup, but most strongly the high
progesterone and high estrogen groups. This suggests that effects of oral contraceptives specifically

*contribute to a reduction or exacerbation of depth varying microstructural intensity, making this*
*microstructural feature in OC females more similar to males. The strong hormone-related lamination*
*effect is particularly interesting when considering the fact that estrogen receptor expression is highly*
*depth-specific, and particularly pronounced in the deeper cortical layers (V and IV⁹¹). Behaviourally*
*relevant sex hormone-related spiking pattern changes also are layer-specific particularly pronounced*
*in deeper cortical layers⁹², potentially driving structural plasticity.*

*[...]”*

**Reviewer #3 (Remarks to the Author):**

**The authors interrogated microstructural differences in the context of sex and menstrual cycle**
**phase on three distinct levels, providing a novel account of sex-specific cytoarchitectural profiles in**
**the brain. I am excited by this work, beautifully executed, and offer several insights that may**
**improve its impact.**

We thank the Reviewer for the appreciation of our work and the insightful comments, which we have
addressed below.

**1. Given the age distribution of the sample (22-37), I wonder if the authors considered potential**
**influences of perimenopause (I have seen females of their mid to late 30s in this stage before,**
**though rare) and/or possible endocrine conditions (e.g., PCOS, history of hysterectomy, etc) that**
**may have impacted hormonal levels. It would be important to at least report the lack of this**
**information for the sake of transparency on potential heterogeneity of the sample, in terms of**
**female hormone concentrations.**

This is an important point. The only information available to us was if individuals had a regular
menstrual cycle, and if they were within a 28 day window of menstruation. We excluded those that
report recent pregnancy, IUDs, hysterectomy, endometriosis and similar conditions. In the methods,
we state:

*Methods, adjustment; p. 29 ll.932:*

*“We included all females who reported regular menstrual cycles, and that their last menses was*
*between 0 and 28 days (n = 284), which is considered the length of a normal menstrual cycle 26.*
*Unfortunately, the current sample did not have information about perimenopausal staging or possible*
*endocrine conditions, posing a potential source of noise.”*

We further add that studies with direct hormonal samples, if possible, at several densely sampled
timepoints should follow this work to further advance research on female health and gonadal
hormones.

*Discussion, addition, p. 21, ll. 669*

*"We also limited the analysis to individuals that report having a regular menstrual cycle, while*
*ignoring perimenopausal hormonal changes as well as other endocrine conditions."*

**2. On a similar note, it would be useful to clarify the criterion of those that "are or have been**
**menstruating in their lives" - Was this explicit to those currently menstruating at the time of the**
**study on a regular basis, or could some females who have not menstruated for months or years on**
**end, but at some point in their lives (as suggested by this criterion), have been included? If so, that**
**could certainly skew the hormonal distribution of the sample.**

In this study, we used the HCP dataset which collected a broad range of items. One of their items is
"gender" but is operationalised as a binary self-report of 'female' and 'male'; another item is
'menstrual age began'; another item is 'regular menstrual cycle'. To account for the biological
implications of sex (vs. gender), we used a combination of the item the HCP authors call 'gender' with
'menstrual age began' to divide the sample into what we term 'sex'. In the second part of our paper,
we fine-tune our definition such that we include only individuals classified as 'females' who take OC
into one group; and only individuals who report a regular menstrual cycle and who have previously
been identified as 'females' to then generate the hormonal groupings. We thus hope to use a valid
estimate for our grouping, but recognise and underscore in the text that this is a rough, correlative
estimate which only gains validity through its big sample size and additional transcriptomic analysis.
We highlight these limitations as follows:

Methods, p. 28, ll. 915:

*"We used self-reported days since menstruation from the day of the scan and about regular*
*OC intake as a grouping variable."*

Discussion, p.21 ll 670

“To provide more robust evidence for a link between gonadal hormones and microstructure, it will be important to follow pioneering macro-scale studies in the future that investigate densely sampled intra-individual hormonal fluctuations as measured by blood-tests and to take both male and female hormonal diurnal fluctuations into account. Such studies will further help understand the association between the anatomy of the brain and hormonal variation and potentially functional consequences.”

3. Regarding the inclusion of a subset of females using OC - More details regarding the type of birth control (estrogen only, progesterone only, or combination), the length of exposure (being mindful of any who have recently started OC and may, therefore, still be adjusting), and the like is needed, considering that these variables play a significant role in the efficacy of OC. I would also encourage the authors to be as explicit as possible when discussing past literature about the effects of OC - For instance, lines 69-71 on page 3 could use more detail (i.e., type of OC, length of OC exposure, age and menopausal status of the sample). In sum, what is meant by "regular" OC?

We are currently grouping every female in this group who indicated ‘yes’ for the question ‘Is the participant using birth control pills, progesterone, or fertility drugs? Yes = 1, No = 0 (Asked of female participants only)’. There is a more fine-grained item which asks “Menstrual_BirthControlCode - What birth control, progesterone, or fertility drugs is the participant using? 1=OC's for contraception, 2=OC's primarily for menstrual regulation, 3=estradiol for menstrual regulation, 4=progesterone for menstrual regulation, 5=fertility therapy, 6=other, 7=unknown (Asked of female participants only)

“; however, all participants answer with ‘OCs for contraception’. We agree that again, this way of grouping is imprecise and does not account for the complexity of this topic. Unfortunately, we are limited by the sample we have and tried the best we could with the data available. To make this limitation more transparent for the reader, we add the following to our **discussion, p. 20, from ll. 649:**

“We acknowledge the extreme simplification for both NC and OC females, where we ignored the specific hormonal formulation of the pill and the initiation and duration of use due to a lack of data.”

**4. It would also be beneficial to expand on the cross-sectional limitations of this study as baseline**
**hormone levels were not acquired from females. Though there is a "usual range" which we might**
**expect reproductive females to fall within in terms of hormone levels at each menstrual stage, what**
**is "normal" for these instances can vary across individuals. Though cross-sectional work is still very**
**informative, a thorough acknowledgement of this limitation, especially in the context of this study,**
**is lacking.**

We agree and add this limitation to our discussion, and further underline the cross-sectional character
of our study in the abstract to inform all readers about the limitations of this study:

**Abstract**

*"[...] We assessed regional variation in cortical microstructure as a function of sex, hormonal status*
*and sex hormone receptor gene expression distribution based on quantitative intracortical profiling in*
*vivo using the magnetic resonance imaging based T1w/T2w ratio in 992 healthy females and males of*
*the **cross-sectional** Human Connectome Project young adult sample.*

*[...]*

*Together, our data thus are suggestive of sex differences in cortical and hippocampal microstructure,*
*as well as **a link of sex hormones with these differences. Albeit correlative,** this study underscores the*
*importance of incorporating sex hormone variables into the investigation of brain structure and*
*plasticity. "*

We also suggest that future work should work intr-individually and with direct blood-samples rather
than correlative measures.

**Discussion, p.21 || 670**

*"To provide more robust evidence for a link between gonadal hormones and microstructure, it will be*
*important to follow pioneering macro-scale studies in the future that investigate **densely sampled***
***intra-individual** hormonal fluctuations as measured by **blood-tests** and to take both male and female*
*hormonal diurnal fluctuations into account."*

**5. Was the time of day held consistent across subjects when collecting hormone information? Were**
**hormones also measured in the males? I wonder if diurnal testosterone fluctuations in males might**
**have an influence on the current results.**

There were no direct hormone measures in this dataset. We derive an indirect hormonal measure by
roughly dividing females into times judged by self-reported menstrual cycle that is generally
characterized by ‘higher’ and ‘lower’ progesterone and estrogen levels. We, however, agree that this
would be a valuable addition and thus add the following to our **discussion, p. 20, from II. 652:**

*“To provide more robust evidence for a link between gonadal hormones and microstructure, it will be*
*important to in the future follow pioneering macro-scale studies that **investigate densely sampled***
***intra-individual hormonal fluctuations** as measured by blood-tests and to take both male and female*
***hormonal diurnal fluctuations into account.**”*

We did, however, add a control analysis in which computed the contrast between the female
subsamples and 1000 permutations of randomly sampled groups of males. While this does not
account for systematic diurnal changes in testosterone, we show that the effects exhibit the same or
larger consistency values as the main sex-difference analysis, if different groups of males are
considered.

Internal consistency - male sample, hormonal contrasts.

*Supplementary Figure 7. Split-correlation of 1000 random permutations for all hormonal contrasts and*
*each microstructural measure. For every split, we computed the contrast between males and females,*
*randomly choosing only a subsample of males, such that $n(\text{males}) = n(\text{females})$. We then computed*
*the internal consistency for this randomly chosen male subsample by correlating the effect sizes of this*
*contrast with the Cohen's d effect sizes of an equally sized and randomly chosen subsample of males.*
*Datapoints represent correlation values for each split.*

**6. Relatedly, were the hormonal assessments, MRI, and menstrual questions completed within the**
**same day? Or could a few females have transitioned to a different menstrual phase over the course**
**of data collection?**

Yes, MRI and menstrual cycle assessments were acquired within the same day (Van Essen et al., 2013;
Supplemental Table S2, first visit).

We adjusted the methods, **p. 28 ll. 922** as follows:

*"We used self-reported days since menstruation from the day of the scan and about regular OC intake*
*as a grouping variable."*

**7. I also wonder if comparisons within females, between the various stage-associated subgroups,**
**might be useful to further interpret the results presented here. If no variations between female**
**groups are found, this may be attributed to the over-generalization of hormone levels by stage**
**rather than on an individual or change-from-baseline degree. If variations are found, however, this**
**could corroborate the authors' grouping approach.**

Thank you for this suggestion. We added an additional analysis in which we show that in particular OC
and NC females differ in their microstructural features. We found systematic differences between OC
and NC females, but not between any other group. We add this finding to our results and adjust the
discussion section accordingly. Since all reviewers had comments about this section, we adjusted the
text significantly. To not artificially blow-up this response letter, please refer to the updated result and
discussion section in the manuscript. We added an overview of this analysis into figure 3D:

Figure 3. Comparing males to different female sub-samples, grouped by menstrual cycle phase. (A) Estrogen and progesterone fluctuate with the menstrual cycle. Horizontal lines under the x-axis indicate grouping: purple reflects progesterone (dotted = low; solid = high); turquoise reflects estrogen (dotted = low; solid = high) (B) Hormones determine cortex-wide sex-difference effect sizes based on post-hoc contrast on cortex-wide effect sizes. Cohen's d per parcel is plotted separately for the three intracortical measures profile mean, profile skewness and the gradient, respectively for each subgroup-comparison. All shown contrasts were significant ($p < .001$). (C) FDR-thresholded Cohen's d maps of T1w/T2w profile mean (i) between males and subsamples of females divided by OC use and menstrual cycle phase projected on the cortical surface and the hippocampus. (ii) FDR-thresholded Cohen's d maps of T1w/T2w profile skewness between males and female subsamples mapped on the cortex. (iii) FDR-thresholded Cohen's d map of differences in the microstructural gradient between

1318 *males and different female sub-samples. For completeness, all other FDR-thresholded Cohen's d maps*
*(all group-comparisons, for each of the three measures) are plotted in supplementary figure 4. D)*
*Microstructural differences between female groups, comparing OC females with all NC females, as well*
*as OC females with specific NC subgroups, divided by their hormonal period. Columns are the three*
*microstructural measures T1w/T2w mean, T1w/T2w skewness, and the microstructural gradient.*
*Purple areas are parcels which had significantly higher values for OC females, orange had significantly*
*higher values for NC females after FDR-thresholding (all Cohen's d).*

**8. Regarding the results showing differences between males and high progesterone females, I would**
**be interested to see a more in-depth interpretation from the authors to offer potential explanations**
**for this specific finding.**

With the within-female analysis that we added, we hope to provide a better idea of robustness of the
variations of results that we observe. In fact, we now take a step-wise approach to interpret these
findings and discuss them appropriately:

- - as before, create subgroups and repeat male vs. female contrast only considering subgroups
of females
- - check consistency by randomly subsampling males and repeating the analysis
- - create an average effect over all parcels, and with ANOVA check if *on average* the
effect-size changes between subgroup GLMs
- - on a regional level, report in which parcels we observe variations in effect size
- - create a between-female subgroups contrast to explain why sex-difference effect size changes

With this analysis and adjustments raised by other reviewers, we have changed the discussion section
on the hormonal sub-group comparisons quite a lot. Here we copy the part which focuses on your
point of interest specifically:

[...]

*In contrast to the mean microstructural intensity, the sex-difference effect in microstructural*
*skewness was driven by NC females, while OC females exhibited profiles more similar to males. The*
*low estrogen, low progesterone, and high estrogen groups all replicated the initial sex difference in the*
*dominance of higher versus lower cortical compartments intensity. However, the effects were different*
*from the main effect when examining females who regularly took oral contraceptives or had high*
*progesterone concentrations. Specifically, there was nearly no difference in lamination between males*
*and females who took OC (weak average effect), but there was an even stronger average difference in*

lamination between males and females with high progesterone concentrations. OCs suppress
circulating estradiol and progesterone levels (Arnold, Tóth, & Faredin, 1980; Basu et al., 1992;
Thornycroft & Stone, 1972). Though no study to date has investigated such effects, we draw analogies
between a recent morphological study focussing on the medial temporal lobe and its link to
progesterone as well as chronic progesterone suppression (such as OCs): here progesterone was shown
to shape MTL volume throughout the menstrual cycle, and ceases to do so when suppressed (Taylor et
al., 2020). Speculatively, this effect might appear through progesterone's effect on myelination (Jung-
Testas et al., 1994, Koeniget al., 1995, Hussainet al., 2011, Koeniget al., 1995). The variations we
observed were mainly driven by stronger effects in the prefrontal, anterior cingulate and tempo-
parietal areas, which are explained by robust differences in skewness in these areas between females
who take OC and any NC female subgroup, but most strongly the high progesterone and high estrogen
groups. This suggests that effects of oral contraceptives specifically contribute to a reduction or
exacerbation of depth varying microstructural intensity, making this microstructural feature in OC
females more similar to males. The strong hormone-related lamination effect is particularly interesting
when considering the fact that estrogen receptor expression is highly depth-specific, and particularly
pronounced in the deeper cortical layers (V and IV; österlund et al., 2000). Behaviourally relevant sex
hormone-related spiking pattern changes also are layer-specific particularly pronounced in deeper
cortical layers (Clemens et al., 2019), potentially driving structural plasticity.

[...]

**9. In general, I would also encourage the authors to take a more careful approach with their**
**discussion of results. The female subgroups may be a bit over-simplified, especially considering the**
**moderate presence of estrogen in what the authors refer to as only the "high progesterone" stage.**
**I am very pleased to see a paper that covers this topic, but am eager to see more unique conclusions**
**that pose important questions while also being mindful of limitations. There is more room for**
**discussion in this manner.**

Thank you for this suggestion. We agree that we need to be careful and thus adjust the wording in
particular in reference to this section such that it will be clear to the reader that these results are
based on rough grouping, correlative, and to be mindful of limitations.

First we explicitly state this fact now in the methods:

*"Note however, that progesterone and estrogen groups do overlap due to this classification. "*

Second, we don't find significant within-female contrasts between naturally cycling groups. We thus
now focus more on the NC vs OC contrasts, reducing the impact of the over-simplified hormonal
classification. (see figure pasted in comment 7).

Third, we updated the limitations in the discussion, and generally adjusted our wording to more
careful conclusions, for example:

*"[...] This effect was particularly driven by the low progesterone subgroup, extending evidence*
*from a recent preprint that reports progesterone-related white-matter microstructural and cortical-*
*thickness variations in the occipital lobe (Rizor et al., 2023). Even though we observed more local*
*variations in the sex-difference effect-size by hormonal subgroup comparison in the collapse*
*microstructural measure, these were not strong enough to show in a within-female comparison. We*
*thus conclude that sex differences in average cortical microstructure are at least partly driven by long-*
*term OC use; but that here, we did not find robust evidence for short-term cycle dependent variations*
*in the sex difference effect.*

*[...]*

*It is furthermore important to note that rather than longitudinally following up on*
*microstructural changes going along with hormonal variations intra-individually or post-mortem tissue*
*analysis, we computed inter-individual contrasts on an indirectly approximated correlative hormonal*
*measure. We acknowledge the extreme simplification for both NC and OC females, where we ignored*
*the specific hormonal formulation of the pill and the initiation and duration of use due to a lack of data.*
*We also limited the analysis to individuals that report having a regular menstrual cycle, while ignoring*
*perimenopausal hormonal changes as well as other endocrine conditions. [...]"*

**Reviewer #4 (Remarks to the Author):**

**I co-reviewed this manuscript with one of the reviewers who provided the listed reports as part of**
**the Nature Communications initiative to facilitate training in peer review and appropriate**
**recognition for co-reviewers.**

Thank you a lot for your efforts!

Reviewer #1 (Remarks to the Author):

The authors have done a great job in revising their manuscript to respond to comments from our initial review. We would support publication of the manuscript in its current form with minor remaining suggested edits as listed below.

Lines 328-337 in the merged pdf, as there are comparisons for all three metrics (mean, skewness, gradient), please clarify which metric is referred to when the sex difference effect is mentioned.

Line 372 in the merged pdf, r value should be .13 not 13, right?

Lines 386-391, thanks authors for adding this supplemental analysis to quantify the effect of unbalanced male vs. female AHBA donors. The correlations did show a global agreement at least in results of skewness. However, it is worthy of note that male- and female-only results have different signs even in skewness for individual genes like HSD17B8.

Line 388 in the merged pdf, supplement figure 8 should be 9

Reviewer #2 (Remarks to the Author):

Thank you to the authors for carefully addressing all of our concerns and for the significant effort dedicated to enhancing the paper. We are pleased to acknowledge that all our suggestions and doubts have been effectively addressed. At this stage, we have no further comments and are ready to accept the paper for publication.

Reviewer #3 (Remarks to the Author):

I appreciate the authors' revision to this manuscript, which has improved significantly in result. I have responded to each of my initial claims and the authors' rebuttal in order below.

1. I am glad to learn that the authors included exclusionary criteria of "recent pregnancy, IUDs, hysterectomy, endometriosis and similar conditions" to control as best as possible for heterogeneity among the sample. Given that hormones are a central theme of the study and these criteria strengthen the validity of its sample, I would suggest stating directly in the manuscript.
2. Throughout the manuscript, I would reword "females who have been menstruating in their lives" to "females that report menstruation within 28 days of the scan" as this seems to be a more accurate representation according to the authors' response.
3. Thank you for your transparency on the generalization of your OC group and adding a note about this to the limitations.
4. The conclusions drawn from this cross-sectional work are more digestible now that the authors have applied their revisions.
5. Regarding the following statement within the revised manuscript: "We accordingly built a high estrogen group for females who were broadly around ovulation (between day 7 until day 23, n = 284), and a low estrogen group for females that were just before and during menstruation (n = 100). Progesterone surges after ovulation during the luteal phase, and was thus defined as low before day 15 (n = 171), and high after day 14 (n = 113)" - Does this mean that some females were included in multiple groups? Given the substantial overlap between them by days? I would recommend making independent groupings to avoid this.

I am also surprised by the large window for the ovulation/high estrogen group - Two weeks seems too broad. Ovulation typically occurs around day 14, and lasts only a day or two, so the current cutoff is likely grasping other hormonal extremes as well. I appreciate the added analysis by the

authors, but think the classification of female subgroups needs reworking, or at least stronger justification.

Regarding: "to take both male and female hormonal diurnal fluctuations into account" - I would reword this as only males experience noticeable diurnal fluctuations in sex hormones while females fluctuate over the course of 28 days. This sentence makes it sound as though diurnal is in reference to both sexes.

6. Thank you for clarifying this detail.

7. My initial suggestion was to compare female subgroups within those naturally cycling to support the authors' approach to classifying high/low estrogen and progesterone groups. If differences are found between high estrogen NC and low estrogen NC groups, for instance, this would validate their grouping method and provide more credibility for the hormonal differences between them. I appreciate the comparisons between each NC subgroup and the OC group, though would like to see the former as well.

8. The authors have added a thought-provoking discussion on the high progesterone vs. males comparison. This improves the impact and interpretation of results.

9. The claims made in the discussion have been appropriately softened to avoid lofty inferences.

Reviewer #4 (Remarks to the Author):

I co-reviewed this article with [REDACTED]. I agree with all the responses and how the authors addressed our comments. Congratulations.

Revision 2 - Letter to the Reviewers - NCOMMS-23-52974A

Relating sex-bias in human cortical and hippocampal microstructure to sex hormones

We would like to thank the Editors and Reviewers for their positive evaluations, constructive comments, and for the opportunity to submit a revised manuscript. We feel that the comments and suggestions have greatly improved our work. In this response letter, we outline the steps taken to address the suggestions of the Reviewers in a point-by-point fashion below and highlight the corresponding changes in the manuscript.

Reviewer #1 (Remarks to the Author):

The authors have done a great job in revising their manuscript to respond to comments from our initial review. We would support publication of the manuscript in its current form with minor remaining suggested edits as listed below.

Many thanks for the positive evaluations and helpful suggestions. We have incorporated them all.

Lines 328-337 in the merged pdf, as there are comparisons for all three metrics (mean, skewness, gradient), please clarify which metric is referred to when the sex difference effect is mentioned.

Thank you, we have further clarified this in the text. It now reads as follows:

*“Comparing the microstructural gradient of males only to subgroups of females of different estimated hormonal profiles changed the distribution, but not the centre of the distribution of cortex-wide gradient sex differences (all cortex-wide effect size contrasts between any group comparison n.s, **Figure 3B**). However, parcel and cortical wide specific analysis give a more detailed overview of variations by hormonal subgroups (**Figure 3C**; supplementary **Figure 8**). The sex difference effect for the microstructural gradient varied strongest when comparing males to only OC takers versus comparing males to only females estimated to have high progesterone levels: Sex-bias between OC takers and males were least extreme ($\min d_{OC\ females} = -.4636$, $\max d_{OC\ females} = .3134$), while sex differences between males and females in their high progesterone phase showed particularly big positive and negative effect sizes ($\min d_{high\ prog\ females} = -.5980$, $\max d_{high\ prog\ females} = .3398$). In particular, the sex-difference effect for the gradient in the insula is negative between males and OC taking females, but positive or n.s. between males and the different NC female groups. Investigating the female differences more closely, we find that the insula’s microstructural profile covariance is closer with the fugal anchor of the gradient in NC than in in OC females; which seems to be associated with by the low estrogen and low progesterone groups (**Figure 3D**). “*

Line 372 in the merged pdf, r value should be .13 not 13, right?

Thanks for spotting, we have corrected this. It now reads accordingly:

“We further found a significant after controlling for spatial auto-correlation, but small spatial overlap with the sex steroid precursor gene HSD17B3 ($r = .13$, $p_{spin} < .05$).”

Lines 386-391, thanks authors for adding this supplemental analysis to quantify the effect of unbalanced male vs. female AHBA donors. The correlations did show a global agreement at least in results of skewness. However, it is worthy of note that male- and female-only results have different signs even in skewness for individual genes like HSD17B8.

Thanks, we have updated this omission. We now highlight that in particular genes with small correlations such as the one that the Reviewer names (HSD17B8) are sensitive in their correlation effect to sample composition. The section accordingly now reads as follows:

“Note that the AHBA dataset from which we derived the transcriptomic maps is composed of only one female and five male donors. We thus tested if the results identified here generally trend in the same directions if rerunning the analysis with the female or male donors only (supplementary figure 9). We find that this is the case for the results for profile mean ($r_{female-all} = 0.4638$; $r_{female-male} = 0.5119$) and profile skewness ($r_{female-all} = 0.7754$; $r_{female-male} = 0.6028$), but not for the microstructural gradient ($r_{female-all} = 0.2$; $r_{female-male} = 0.0603$). This analysis demonstrated that small correlations are particularly sensitive to donor sex (supplementary figure 9). Therefore, in this work, we focus on those that presented most reliably independent of the sample composition.”

Line 388 in the merged pdf, supplement figure 8 should be 9

Many thanks, we have corrected this - see above.

Reviewer #2 (Remarks to the Author):

Thank you to the authors for carefully addressing all of our concerns and for the significant effort dedicated to enhancing the paper. We are pleased to acknowledge that all our suggestions and doubts have been effectively addressed. At this stage, we have no further comments and are ready to accept the paper for publication.

Many thanks for the feedback and appreciation of our work and the constructive revision round!

Reviewer #3 (Remarks to the Author):

I appreciate the authors' revision to this manuscript, which has improved significantly in result. I have responded to each of my initial claims and the authors' rebuttal in order below.

Many thanks for the positive feedback and the additional comments. We believe that they have been able to further clarify open points and improve our work. We have edited the manuscript according to the comments below.

1. I am glad to learn that the authors included exclusionary criteria of "recent pregnancy, IUDs, hysterectomy, endometriosis and similar conditions" to control as best as possible for heterogeneity among the sample. Given that hormones are a central theme of the study and these criteria strengthen the validity of its sample, I would suggest stating directly in the manuscript.

Thank you, this is a very valuable comment. We have now noted this in the respective methods section of the manuscript:

"We included all females who reported regular menstrual cycles, and that their last menses was between 0 and 28 days ($n = 284$), which is considered the length of a normal menstrual cycle⁴³, and excluded those that report recent pregnancy, IUDs, hysterectomy, endometriosis and similar conditions."

2. Throughout the manuscript, I would reword "females who have been menstruating in their lives" to "females that report menstruation within 28 days of the scan" as this seems to be a more accurate representation according to the authors' response.

We agree that this is an important piece of information we should remind the reader of. We have now updated this in our manuscript, e.g.:

Introduction:

"We then contrasted these microstructural measures between females and males, tested how these sex-differences vary if systematically comparing males with females of particular hormonal profiles (approximated by self-reported menstrual cycle phase at the day of the scan and OC use)"

Results:

"We repeated the previous male vs. female contrasts five times, every time considering only those subgroups of females that were characterized by a certain hormonal profile: females who regularly took OC ($n = 170$), females who reported to be around their menstruation at the day of the scan (low estrogen, $n = 100$); females who reported to be around their ovulation (high estrogen, $n = 184$); "

Discussion:

“We furthermore demonstrate that there is a particularly big difference in cortical microstructure between females who take OC and NC females who report menstruation within 28 days of the scan, as supported by significant within-females effects.”

Methods:

“We included all females who reported regular menstrual cycles within 28 days of the scan, and that their last menses was between 0 and 28 days (n = 284), which is considered the length of a normal menstrual cycle⁴³, and excluded those that report recent pregnancy, IUDs, hysterectomy, endometriosis and similar conditions.”

3. Thank you for your transparency on the generalization of your OC group and adding a note about this to the limitations.

Thanks a lot.

4. The conclusions drawn from this cross-sectional work are more digestible now that the authors have applied their revisions.

Thank you!

5. Regarding the following statement within the revised manuscript: "We accordingly built a high estrogen group for females who were broadly around ovulation (between day 7 until day 23, n = 284), and a low estrogen group for females that were just before and during menstruation (n = 100). Progesterone surges after ovulation during the luteal phase, and was thus defined as low before day 15 (n = 171), and high after day 14 (n = 113)" - Does this mean that some females were included in multiple groups? Given the substantial overlap between them by days? I would recommend making independent groupings to avoid this.

Yes, the Reviewer is correct, females can be included in multiple groups. We chose this grouping as the cyclic progesterone and estrogen peak and dips cannot be split coherently independently from each other in the current framework and dataset. Since our aim was to make this study accessible to an audience wider than experts of the menstrual cycle literature, we deemed a process close to the most well-known hormones to be best in the context of our study. We agree that this grouping comes with both upsides and downsides, which we accounted for when interpreting results. Nevertheless, we believe that it is the most sensible grouping for this dataset and this audience. We furthermore took great care in the text to not overstate the effects within NC females. We stress that the biggest effects can be seen when contrasting NC and OC females and that this part of the study requires future support of direct hormonal measurements, intra-individual comparisons or manipulations. We conclude that in this manuscript, we provide evidence that there can be systematic variations in the sex-bias effect if completely ignoring the cycle phase, but that these effects are not strong enough to show in intra-NC-female comparisons.

We took greater care in explaining the rationale of our grouping in the methods:

“Lastly, we built groups in which the estimated progesterone and estrogen concentration of NC females differed the strongest according to a normative trajectory of hormonal fluctuations within the menstrual cycle (e.g. Zlotnik et al., 2011). Since estrogen and progesterone concentration peak at different points within the menstrual cycle, we subdivided NC females in a low and high progesterone, and in a low and high estrogen group, respectively. Importantly, since these peaks occur at different points in time, the grouping of estrogen and progesterone partly overlap and are thus not independent of each other. In total, we thus compared five subsamples of females against the cortical microstructure of males: an OC group, a high and low estrogen group, and a high and low progesterone group. We included all females who reported regular menstrual cycles within 28 days of the scan, and that their last menses was between 0 and 28 days (n = 284), which is considered the length of a normal menstrual cycle⁴³, and excluded those that report recent pregnancy, IUDs, hysterectomy, endometriosis and similar conditions. Unfortunately, the current sample did not have information about perimenopausal staging or possible endocrine conditions, posing a potential source of noise.

Estrogen is low in the beginning of the cycle and starts to rise before ovulation, with a second peak premenstrual in the luteal phase, before it drops again just before and during menstruation (Figure 3 A). We accordingly built a high estrogen group for females who reported they were in the middle of their menstrual cycle (between day 7 until day 23, n = 284), and a low estrogen group for females that were just before and during menstruation (n = 100). Progesterone surges after ovulation during the luteal phase, and was thus defined as low before day 15 (n = 171), and high after day 14 (n = 113). This classification is in accordance with common comparisons between the time window of menstruation with the one around ovulation (high and low estrogen) and luteal vs. follicular phase (high and low progesterone)^{48,81,126,127}. While this best accounts for differences in concentration for each of these hormones, progesterone and estrogen groups do overlap due to this classification. ”

I am also surprised by the large window for the ovulation/high estrogen group - Two weeks seems too broad. Ovulation typically occurs around day 14, and lasts only a day or two, so the current cutoff is likely grasping other hormonal extremes as well. I appreciate the added analysis by the authors, but think the classification of female subgroups needs reworking, or at least stronger justification.

Thank you for this comment. We realize that our wording was not very clear. Our aim was not to refer to ovulation, but rather to the two estrogen peaks before and after ovulation. We thus corrected the text to the following:

“Estrogen is low in the beginning of the cycle and starts to rise before ovulation, with a second peak premenstrual in the luteal phase, before it drops again just before and during menstruation (Figure 3 A). We accordingly built a high estrogen group for females who reported they were in the middle of their menstrual cycle (between day 7 until day 23, n = 284), and a low estrogen group for females that were just before and during menstruation (n = 100). ”

Regarding: "to take both male and female hormonal diurnal fluctuations into account" - I would reword this as only males experience noticeable diurnal fluctuations in sex hormones while females fluctuate over the course of 28 days. This sentence makes it sound as though diurnal is in reference to both sexes.

Thank you for this recommendation. We now reworded the sentence as follows:

"To provide more robust evidence for a link between gonadal hormones and microstructure, it will be important to follow pioneering macro-scale studies in the future that investigate densely sampled intra-individual hormonal fluctuations as measured by blood-tests, which will measure female hormonal fluctuations more precisely and allow to also take male diurnal hormonal fluctuations into account."

6. Thank you for clarifying this detail.

Happy to clarify.

7. My initial suggestion was to compare female subgroups within those naturally cycling to support the authors' approach to classifying high/low estrogen and progesterone groups. If differences are found between high estrogen NC and low estrogen NC groups, for instance, this would validate their grouping method and provide more credibility for the hormonal differences between them. I appreciate the comparisons between each NC subgroup and the OC group, though would like to see the former as well.

Thanks for noting this, we are happy to also provide these contrasts and have now included them in the results and supplementary results. However, there were no significant differences between high and low estrogen; nor between high and low progesterone groups at a FDR threshold. We did, nevertheless, include the non-corrected maps in the supplement for future reference.

Supplement 7. Non-significant microstructural differences between NC females. NC females were divided by hormone estimations according to self-reported days after menstruation. Columns are the three microstructural measures T1w/T2w mean, T1w/T2w skewness, and the microstructural gradient. Purple areas are parcels which had higher values for females in the high estrogen or progesterone group, oranges indicate higher values for NC females in the respective lower hormonal group (all Cohen's *d*). Note that no parcel was significant at an FDR-threshold.

We furthermore made these results more explicit in the text:

Results

*“The within-female contrast between for the T1w/T2w profile mean between females in their low vs. high progesterone group and between females in their low vs. high estrogen group was not significant at an FDR-corrected threshold (for not-corrected maps, see **supplementary Figure 7**)*

[...]

*The within-female contrast between for the T1w/T2w profile skewness between females in their low vs. high progesterone group and between females in their low vs. high estrogen group was not significant at an FDR-corrected threshold (for not-corrected maps, see **supplementary Figure 7**).*

[...]

Within NC-female contrasts for the microstructural gradient were not significant.”

Discussion:

“We furthermore demonstrate that there is a particularly big difference in cortical microstructure between females who take OC and NC females who report menstruation within 28 days of the scan, as supported by significant within-females effects between these groups. Areas in which we observe these variations largely overlapped with regions that had previously been named as key regions for volumetric menstrual cycle differences (hippocampus, cingulate cortex, insula, inferior parietal lobule, prefrontal

cortex ⁴⁷), or gray matter volume differences due to oral contraceptive use (prefrontal cortex ⁸¹ and the cingulate cortex ⁴⁶). Importantly, our findings do not extend to significant differences within cycle phases for any microstructural measure. Together, adding to previous observations of the effect of sex hormones on macro-level brain structure, our results demonstrate microstructural variability as a function of exogenous and endogenous sex hormones in females in the long and medium term. “

[...]

“Even though we observed more local variations in the sex-difference effect-size by hormonal subgroup comparison in the collapse microstructural measure, these were not strong enough to show in a within-female comparison after correction for multiple comparisons. We thus conclude that sex differences in average cortical microstructure are at least partly dependent on long-term OC use; but that here, we did not find robust evidence for short-term cycle dependent variations within the female subgroups.”

8. The authors have added a thought-provoking discussion on the high progesterone vs. males comparison. This improves the impact and interpretation of results.

Many thanks!

9. The claims made in the discussion have been appropriately softened to avoid lofty inferences.

Many thanks!

Reviewer #4 (Remarks to the Author):

I co-reviewed this article with [REDACTED]. I agree with all the responses and how the authors addressed our comments. Congratulations.

Many thanks for your comments and appreciation of the work.

Reviewer #3 (Remarks to the Author):

I appreciate the authors' rephrasing and clarification efforts in this re-revised manuscript.

Though the classification of cycle phases is not ideal, I accept the authors' response and disclosures added to the methods.

Unfortunately though, the contrasts included in Supplement 7 weaken the authors' approach to these classifications, as I would expect the within-group comparisons for cycle phases among the NC cohort to be significant if accurately representing such distinct phases in which brain dynamics are known to differ. However, the attempt to soften claims regarding comparisons within the NC group, the focus on OC vs NS results, and the added discussion of these supplementary comparisons are sufficient to address this limitation.

I encourage the authors to proofread their manuscript for grammatical errors, as I noticed a few in the quoted text within the reviewer response document.

I thank the authors for their thorough efforts in revising this manuscript and improving its impact. I look forward to seeing it published - Congratulations!